# Representational dynamics during extinction of fear memories in the human brain

Extinction learning—the suppression of a previously acquired fear response—is critical for adaptive behaviour and core for understanding the aetiology and treatment of anxiety disorders. Electrophysiological studies in rodents have revealed critical roles of theta (4–12 Hz) oscillations in amygdala and hippocampus during both fear learning and extinction, and engram research has shown that extinction relies on the formation of novel, highly context-dependent memory traces that suppress the initial fear memories. Whether similar processes occur in humans and how they relate to previously described neural mechanisms of episodic memory formation and retrieval remains unknown. Intracranial EEG recordings in epilepsy patients provide direct access to the deep brain structures of the fear and extinction network, while representational similarity analysis allows characterizing the memory traces of specific cues and contexts. Here we combined these methods to show that amygdala theta oscillations during extinction learning signal safety rather than threat and that extinction memory traces are characterized by stable and context-specific neural representations that are coordinated across the extinction network. We further demonstrate that context specificity during extinction learning predicts the reoccurrence of fear memory traces during a subsequent test period, while reoccurrence of extinction memory traces predicts safety responses. Our results reveal the neurophysiological mechanisms and representational characteristics of context-dependent extinction learning in the human brain. In addition, they show that the mutual competition of fear and extinction memory traces provides a mechanistic basis for clinically important phenomena such as fear renewal and extinction retrieval.

In a constantly changing world, the ability to update previously learned knowledge is essential for adaptive behaviour. The neural mechanisms that support this ability have been extensively studied within the domain of extinction learning. Extinction learning allows for the inhibition of previously acquired unconditioned responses (in particular, fear responses) that have become dysfunctional; accordingly, deficits of extinction learning are associated with numerous psychiatric conditions, most prominently anxiety disorders and depression, and are targeted during exposure therapy[1–4].

While early accounts of extinction learning suggested that it weakens the association between a conditioned stimulus (CS) and an unconditioned stimulus (US)[5], more recent theories propose that it comprises the formation of a new inhibitory memory trace that suppresses the initial fear response[3,6–8]. This new memory trace is highly context-dependent, as demonstrated by the phenomenon of renewal, where fear responses re-emerge when tested either in the initial (acquisition) context or in a new context[3,6,9–12].

Human neuroimaging studies have provided a detailed understanding of the circuits involved in fear learning and extinction, identifying a core network that includes amygdala (AMY), hippocampus (HPC) and regions of prefrontal cortex (PFC) (see refs. 13,14). Among these regions, PFC and HPC appear particularly relevant for the

✉e-mail: daniel.pacheco@uab.cat; qi.chen27@gmail.com; nikolai.axmacher@rub.de

context specificity of extinction learning[3,6,8,10,15–17]. Indeed, the PFC seems to be more involved in the inhibition of fear responses[3,17] and in the contextual regulation of fear during and after extinction[10] than in the acquisition of novel fear memories. Furthermore, recent studies indicate that the lateral PFC (lPFC) supports flexible, task-dependent updating of memory traces[18], consistent with its role in adaptive cognitive control[19,20].

Research in rodents has provided important insights into the neural mechanisms of fear acquisition, expression and extinction. Notably, several studies have shown that the amplitude of theta (4–12 Hz) oscillations in the prelimbic cortex, dorsal anterior cingulate cortex (dACC) and/or amygdala increases during fear acquisition[21] and expression[22–29]. Similarly, human EEG studies have reported heightened frontocentral theta oscillations, particularly during fear conditioning (for example, refs. 30,31) and fear recall (for example, refs. 32–34; but see ref. 35). In addition, studies in rodents have shown that long-range theta (2–12 Hz) connectivity across regions of the extinction network, including AMY, PFC and HPC, is associated with the discrimination of aversive and safe cues[26], and with the extinction of fear associations[25,36,37].

In humans, the electrophysiological mechanisms underlying extinction learning in AMY and HPC are largely unknown because these regions are difficult to investigate non-invasively. Intracranial EEG (iEEG) recordings in epilepsy patients provide invaluable data about neurophysiological processes in deep brain areas at the highest temporal and spatial resolution possible[38,39]. Indeed, iEEG research in the domain of episodic memory has provided important insights into the mesoscopic neural mechanisms underlying memory formation and retrieval[40–47], including evidence for a prominent role of theta oscillations[48]. Other recent iEEG studies described the neurophysiological mechanisms in AMY and HPC underlying the impact of aversive emotions on episodic memory formation and retrieval[49–53].

Despite this progress in understanding episodic memory and its modulation by aversive emotions, surprisingly few iEEG studies explored the neurophysiological mechanisms of fear learning and extinction; thus, it is unclear whether and to which extent they rely on the same neural machinery that supports episodic memory. A notable exception is ref. 30, which reported increased theta oscillations in AMY and dorsomedial PFC following aversive stimuli during fear learning. Whether AMY and PFC theta frequency oscillations play a role during extinction learning remains unknown.

In addition to neural oscillations, rodent research has described how memory traces of specific experiences—engrams—are built in AMY, HPC and other areas during fear learning and how they are suppressed or modified during extinction[54,55]. In humans, representational similarity analysis (RSA; see refs. 56,57) is increasingly used to identify stimulus-specific memory traces, that is, the neural representations of unique episodes and events, at a meso- and macroscopic level[58–63]. Applied to iEEG data, this approach typically relies on patterns of the power of neural oscillations across frequencies and electrodes[64–66]. RSA can then either be used to calculate the similarity of representations between encoding and retrieval (reinstatement, or encoding–retrieval similarity), or to measure similarities between representations of different items during either encoding or retrieval.

Episodic memory studies using RSA revealed fundamental differences in the 'representational formats' of memory traces of item–context associations as compared with single items: while memories of individual items depend on broad frequency ranges and on sensory regions—for visual stimuli, in the lateral temporal cortex (TMP)—memory traces of item–context associations exhibit a strong reliance on theta frequency oscillations and the HPC[61,62]. Because of the strong context dependency of extinction learning, one may expect similar representational signatures as for episodic memory traces of item–context associations. Furthermore, RSA studies using functional magnetic resonance imaging (fMRI) demonstrated that the 'stability' of item

representations, that is, the similarity of neural activity patterns across repeated exposures, correlates with successful memory formation[67,68]. Similar effects were observed during fear acquisition, where higher levels of item stability predicted learning success[69].

Here we investigated the neurophysiological mechanisms and representational characteristics of fear and extinction memory traces and their impact on subsequent renewal. We developed a novel fear and extinction learning paradigm that we conducted with epilepsy patients ($N = 49$) who were implanted with iEEG electrodes across the fear and extinction network (Fig. 1a). During acquisition, patients were exposed to images of three electric devices, two of which were paired with an aversive stimulus (CS+) while the third one was not (CS−). During extinction, the contingency of one CS+ cue changed, resulting in three different types of cue: CS+$_{acquisition}$/CS+$_{extinction}$ (CS++), CS+$_{acquisition}$/CS−$_{extinction}$ (CS+−) and CS−$_{acquisition}$/CS−$_{extinction}$ (CS−−; Fig. 1a, top). During the final test phase, none of the items was paired with a US. Critically, to study the specificity of context representations, the CS items were presented within four different thematically related context videos in each experimental phase (for example, four snow landscape videos during acquisition, four videos during extinction and four videos during test), corresponding to an 'ABC' paradigm (Fig. 1a, top). This paradigm allowed us to track the memory traces of items and contexts during fear acquisition, extinction and renewal across key regions of the extinction network including AMY, HPC and PFC regions, as well as sensory processing areas in TMP (Fig. 1b,c).

## Behavioural results

We first analysed the behavioural responses to the different cue types across experimental phases. A two-way analysis of variance (ANOVA) with 'cue type' (CS++, CS+−, CS−−) and 'experimental phase' (acquisition, extinction, test) as repeated measures revealed significant main effects of both cue type ($F_{(2, 94)} = 19.02$, $p < 0.001$, $\eta p^2 = 0.29$, CI: [0.14, 0.42]) and experimental phase ($F_{(2, 94)} = 9.03$, $p < 0.001$, $\eta p^2 = 0.16$, CI: [0.04, 0.29]) as well as a significant interaction ($F_{(4, 188)} = 6.96$, $p < 0.001$, $\eta p^2 = 0.29$, CI: [0.04, 0.21]). Importantly, cues with changing contingency (CS+−) were perceived as significantly less threatening during extinction than during acquisition ($t_{(47)} = 5.67$, $p < 0.001$; Cohen's $d = 0.75$, CI: [0.46, 1.09]; Bonferroni corrected), confirming that participants correctly adjusted their expectations of the US.

We evaluated the progressive learning of trial contingencies during acquisition and extinction. We sorted trials according to their temporal position (that is, trial number) separately in the three conditions (CS++, CS+− and CS−−) and compared responses at every position using paired Wilcoxon signed-rank tests, followed by cluster-based permutation statistics to correct for multiple comparisons (Methods). During acquisition, ratings of CS++/CS+− vs CS−− cues started to differ at trials 10 and 13, respectively (both $p = 0.001$; thick brown and grey lines in Fig. 1d), while ratings did not differ between CS++ and CS+− items, which were both threatening during acquisition (red and yellow lines, all $p_{corr} > 0.05$). During extinction, ratings started to differ between CS++ and CS+− cues at trial 9 ($p = 0.001$), indicating that participants gradually learned the new contingencies, while they differed immediately between CS++ vs CS−− cues ($p = 0.001$; Fig. 1d, thick orange and brown lines). Ratings of CS+− and CS−− cues did not differ significantly (all $p_{corr} > 0.05$). During the test phase, ratings of CS++ and CS−− differed in the first 3 trials, while the remaining contrasts did not show any significant differences. However, comparisons of average ratings during the test revealed significant differences between CS++ and CS+− trials ($t_{(47)} = 2.51$, $p = 0.015$; Cohen's $d = 0.44$, CI: [0.08, 0.82]) and between CS++ and CS−− trials ($t_{(47)} = 3.1$, $p = 0.003$; Cohen's $d = 0.63$, CI: [0.21, 1.08]), whereas no significant difference was observed between CS+− and CS−− trials ($t_{(47)} = 1.13$, $p = 0.26$; Cohen's $d = 0.21$, CI: [−0.16, 0.6]). Taken together, these results show that participants accurately learned the task contingencies and their changes throughout the experiment. In addition, while differences in ratings diminished during

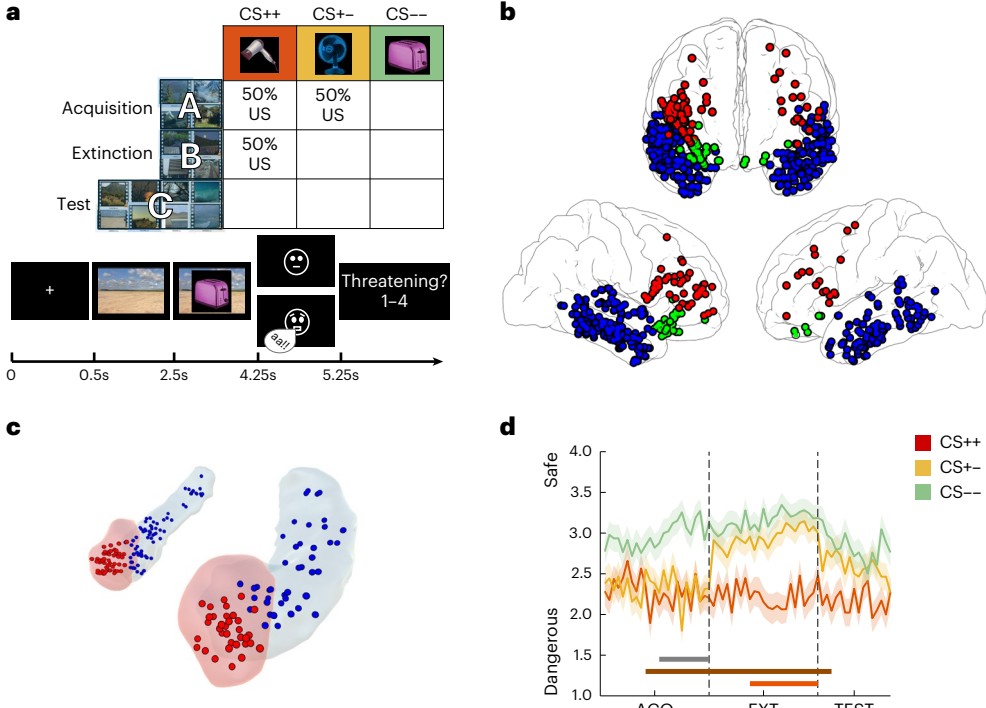

**Fig. 1 | Experimental paradigm, electrode implantation and behavioural results. a**, Experimental design. Top: contingencies of consistently threatening CS++ cues (red), consistently safe CS−− cues (green) and cues with changing contingencies (yellow) across the three experimental phases (acquisition, extinction and test). Cues were presented together with a set of context videos that were unrelated to the contingency in a given trial and that changed between phases (ABC paradigm). Bottom: trial structure. Contexts were presented first but remained on screen during CS presentation. The US consisted of either a neutral face (CS− trials) or a fearful face paired with a loud scream (CS+ trials). Participants rated the perceived cue contingencies on each trial. The cartoon faces shown are for illustrative purposes only and do not depict the actual stimuli used in the experiment. **b**, Electrode implantation. Selected contacts across our group of participants are overlaid on an average brain surface in MNI coordinates. Neocortical regions of interest selected for analysis included electrodes implanted bilaterally in temporal (blue), prefrontal (red) and orbitofrontal (green) cortices. **c**, Electrode locations in the amygdala (red) and the hippocampus (blue). **d**, Behavioural performance. Wilcoxon signed-rank tests were conducted at every trial position to investigate the progressive learning of task contingencies across experimental phases. Regions where significant effects were observed after correction for multiple comparisons are indicated at the bottom of the figure (cluster-based permutation statistics with 1,000 permutations; grey: CS+− vs CS−−; brown: CS++ vs CS−−; orange: CS+− vs CS++). Shaded lines depict group average ratings at each trial position ±s.e.m.

the test phase, participants still rated CS++ items on average as more threatening than CS−− or CS+− items, indicating that fear responses did not completely disappear.

## Amygdala theta oscillations signal safety during extinction

Previous literature has established a key role of theta (4–12 Hz) oscillations in AMY, HPC and PFC for fear learning and extinction in both rodents[28] and humans[30]. We thus focused on this frequency band in our initial analysis and compared differences in theta power between CS+ and CS− cues. Following previous studies (for example, ref. 30), we specifically assessed the time period from cue onset until US presentation (in reinforced CS+ trials, or corresponding time point in the other trials) during both acquisition and extinction.

During acquisition, we found no significant power differences between CS+ and CS− cues in any of our regions of interest (ROIs; all $p_{corr} > 0.05$; see Fig. 2a left for the results in the AMY). During extinction, theta power was significantly higher for CS− than for CS+ trials in the AMY ($p_{corr} = 0.005$; Fig. 2a, right), while no significant power differences were observed in any other ROI (all $p_{corr} > 0.05$). This AMY effect occurred at a relatively late time period preceding the US presentation (that is, from 1.18 to 1.75 s) and was confined to the theta (3–10 Hz) frequency range (Fig. 2a, right). Follow-up analyses revealed that theta power was significantly lower in CS++ trials than in both CS+− ($p = 0.005$; Fig. 2b, top) and CS−− trials ($p = 0.026$; Fig. 2b,

middle), with effects occurring in overlapping time frequency periods to those observed in the 'current valence' contrast in Fig. 2a, right. No significant differences were observed between CS+− and CS−− trials ($p = 0.38$; Fig. 2b, bottom). To determine whether the difference between CS+ and CS− trials was specific to the theta band, we analysed all frequencies across the 1–100 Hz range. Results confirmed that differences between conditions were localized to the same cluster observed in Fig. 2a, right, while no significant effects were observed in other time–frequency bins ($p = 0.004$; Fig. 2c, top). Further analyses in this cluster revealed that effects were driven by both theta power increases above baseline for CS+− items ($t_{(30)} = 3.36$, $p = 0.002$; Cohen's $d = 0.59$, CI: [0.21, 0.95]) and theta power reductions below baseline for CS++ items ($t_{(30)} = -4.08$, $p = 0.0003$; Cohen's $d = -0.71$, CI: [−1.09, −0.32]; Fig. 2c, bottom). CS−− items did not differ from baseline ($t_{(30)} = 2.03$, $p = 0.051$; Cohen's $d = 0.36$, CI: [−0.001, 0.71]). As a baseline, we chose the average across all trials in all phases (see Methods). No differences between CS+ and CS− trials occurred in this time–frequency cluster during acquisition ($t_{(30)} = -0.69$, $p = 0.5$; Cohen's $d = -0.07$, CI: [−0.56, 0.42]). Please note that these latter analyses are not circular because the time–frequency cluster is defined by contrasting CS+ and CS− trials during extinction and not by comparing the three different trial types (CS++, CS+−, CS−−) to zero. Together, these results show increases in AMY theta power during extinction for CS− compared with CS+ trials. They also reveal that theta power in CS− trials during extinction was above baseline, which may reflect a safety signal.

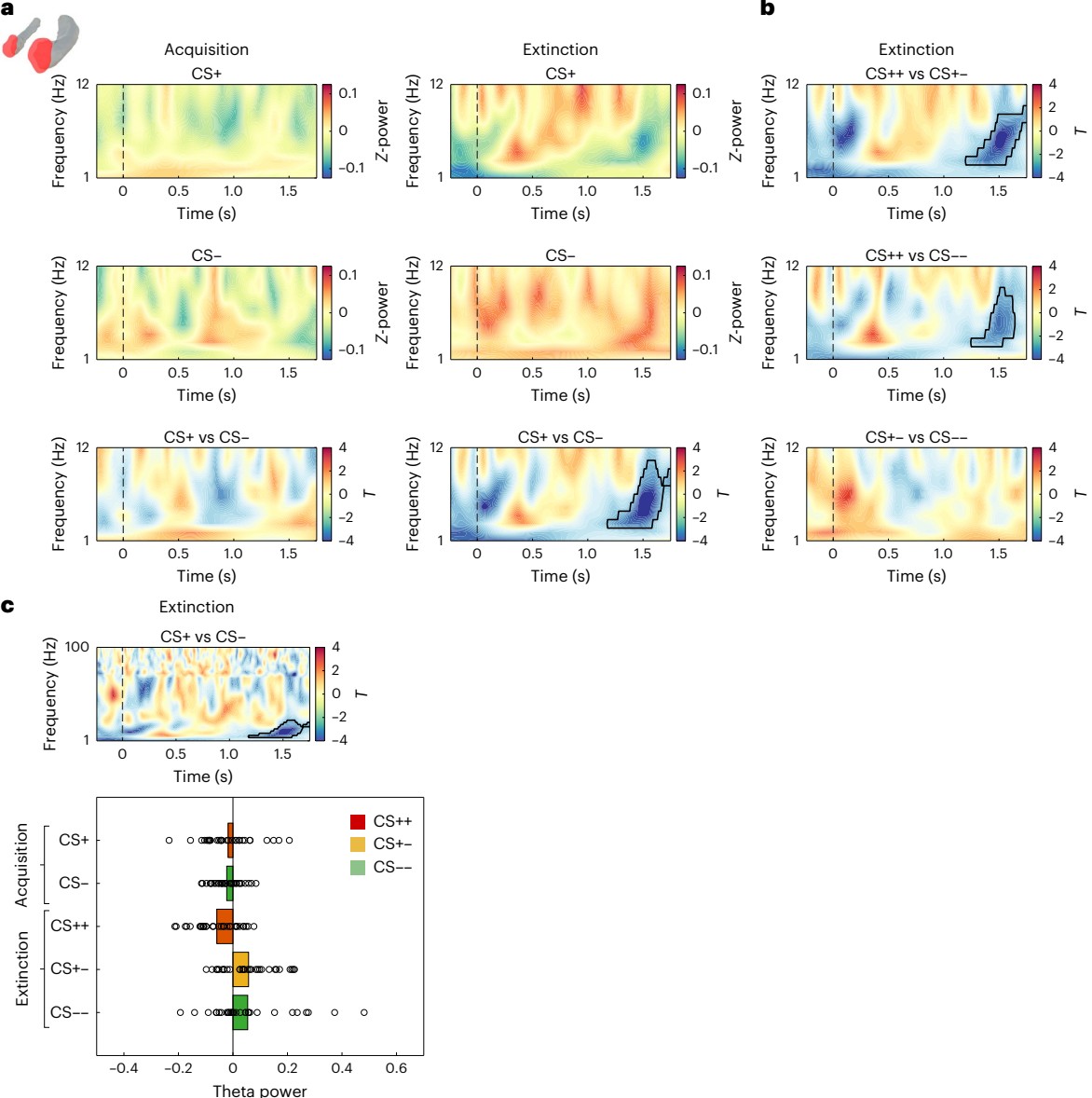

**Fig. 2 | Higher theta power in the amygdala for CS− than for CS+ trials during extinction. a**, Amygdala 1–12 Hz power. During acquisition (left), no significant differences in low-frequency power were observed between CS+ and CS− trials. Bottom row: result of the contrast between the two conditions. Top and middle rows: levels of Z-scored power for each condition. During extinction (right), oscillatory power was significantly higher for CS− trials in a late time period of cue acquisition, that is, from 1.18 to 1.75 s in the 3–10 Hz frequency range. Negative T values depicted in blue reflect higher theta power for CS− items.

**b**, T maps of the CS++ vs CS+− (top), CS++ vs CS−− (middle) and CS+− vs CS−− (bottom) contrasts. **c**, Top: CS+ vs CS− contrast in the frequency spectrum between 1 and 100 Hz. Bottom: average theta power in the cluster observed in **a**, bottom right, and **c**, top, for all CS types during the phases of acquisition and extinction. In **a**–**c**, two-sided paired t-tests were applied at each time–frequency bin (N = 32). Significant regions surviving correction for multiple comparisons using cluster-based permutation statistics are outlined in black in the time–frequency maps.

## Stable cue representations of CS+ items during extinction

Previous fMRI research has shown that the stability of neural representations across repetitions predicts both fear learning[69] and episodic memory formation[67,68]. We employed RSA to compare the stability of CS+ vs CS− cue representations during acquisition and extinction (Fig. 3a), again focusing on the time period of cue presentation. We built representational feature vectors based on the pattern of iEEG power values across frequencies (44 values in the 1–100 Hz range) and across electrodes in each ROI. These patterns were extracted in windows of 500 ms, incrementing in steps of 50 ms (90% overlap), and were compared across all pairs of trials using Spearman correlations (see 'Contrast-based RSA' in Methods).

During acquisition, item stability did not differ between CS+ and CS− items in any ROI (all $p_{corr} > 0.05$). During extinction, we observed significant increases in item stability for CS+ vs CS− cues in the AMY between 1.25 and 1.5 s ($p_{corr} = 0.018$), overlapping in time with the period of theta power increases for CS− items (Fig. 3b right). Follow-up analyses within this time period revealed that item stability was significantly higher for CS++ compared with both CS+− trials ($t_{(31)} = 2.30$, $p = 0.028$; Cohen's $d = 0.66$, CI: [0.07, 1.31]; Fig. 3c left) and CS−− trials ($t_{(31)} = 2.39$, $p = 0.023$; Cohen's $d = 0.62$. CI: [0.09, 1.22]; Fig. 3c middle). No differences were observed between CS+− and CS−− trials ($t_{(31)} = 0.02$, $p = 0.98$; Cohen's $d = 0.007$, CI: [−0.52, 0.54]; Fig. 3c right). In the TMP, item stability during extinction was significantly higher for CS+ compared with CS− trials between 0.65–1 s ($p_{corr} = 0.005$; Fig. 3d right). Within this time window,

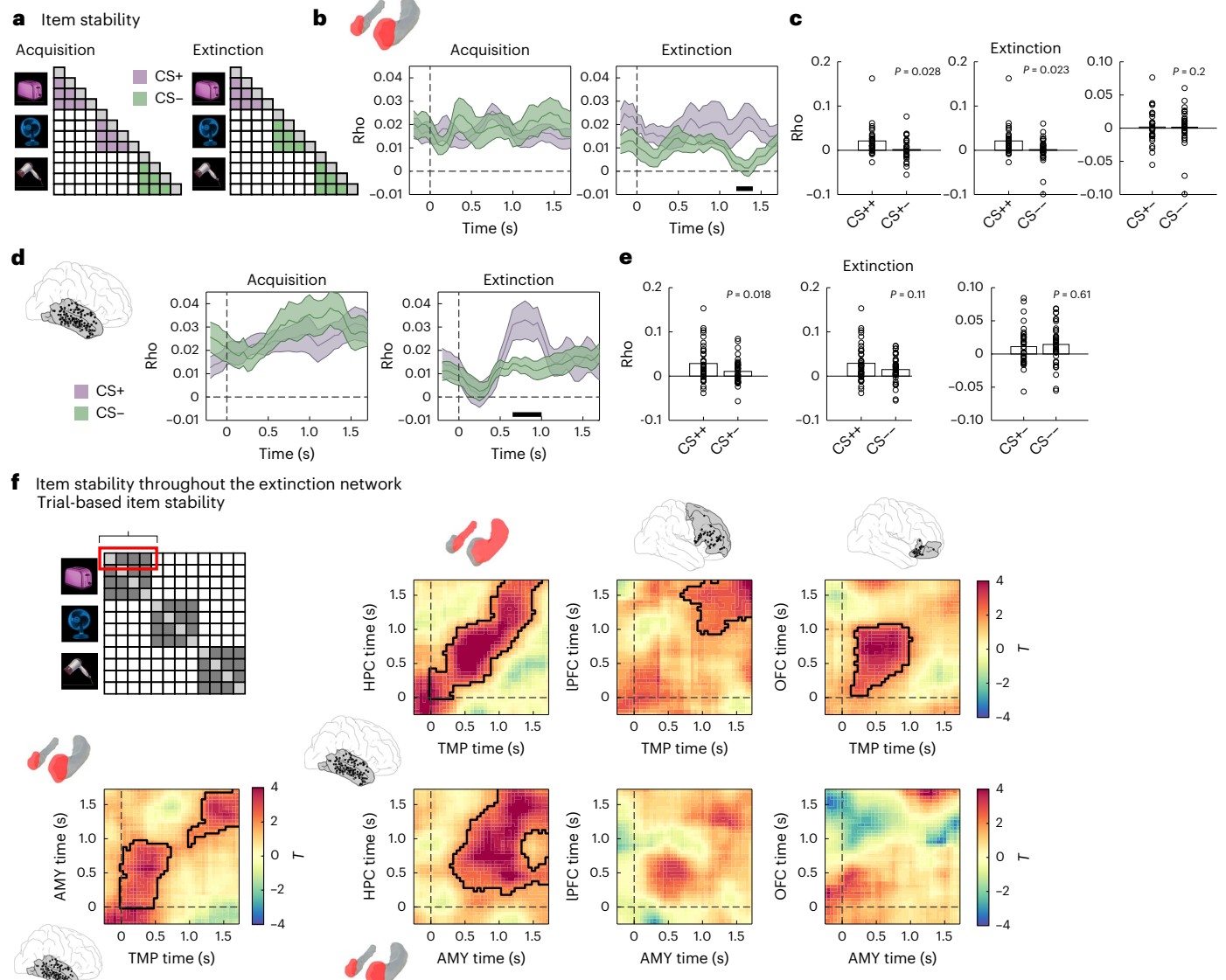

**Fig. 3 | Item stability in lateral temporal cortex, amygdala and across the extinction network. a**, Item stability, defined as the the average similarity of neural patterns representing individual items across repeated presentations, was assessed separately for CS+ and CS− trials during acquisition and extinction. **b**, Analysis of item stability in the AMY revealed no significant differences between CS+ and CS− trials during acquisition (left). During extinction (right), item stability was significantly higher for CS+ than for CS− items in a late time period during cue presentation (1.2–1.5 s after cue onset; $p_{corr}$ = 0.018). **c**, Item stability during extinction in the significant time period shown in **b** (right) for CS++ and CS+− (left; $p$ = 0.028), CS++ and CS−− (middle; $p$ = 0.023), and CS+− and CS−− trials (right; $p$ = 0.2) ($N$ = 32 in all analyses). **d**, Analysis of item stability in the TMP revealed no significant differences between CS+ and CS− trials during acquisition (left). During extinction (right), item stability was significantly higher for CS+ than for CS− items in the time period from 0.65–1 s after cue onset ($p_{corr}$ = 0.005). **e**, Item stability during extinction in the time period shown in **d** (right) for CS++ and CS+− (left; $p$ = 0.018), CS++ and CS−− (middle; $p$ = 0.11) and CS+− and CS−− trials (right; $p$ = 0.61) ($N$ = 41 in all analyses). **f**, Top left: a single-trial metric of item

stability was computed by assessing the similarity of each trial across repeated presentations (average of the comparisons highlighted with a red rectangle in the RSA matrix). Bottom left: fluctuations in item stability were coordinated across trials between the TMP and the AMY during the first second of cue presentation and towards the end of the cue presentation period. Right: item stability was computed in each ROI and correlated across trials with the values observed in TMP (top row) and AMY (bottom row). In **b** and **d**, thick black horizontal lines depict significant time periods after correction for multiple comparisons using cluster-based permutation statistics. Time zero indicates the onset of the CS, and shaded lines depict group average rho values ± s.e.m. Two-sided paired $t$-tests across conditions were applied at every time point (**b**,**d**) or every time-by-time point (**f**) to assess statistical significance. In **c** and **e**, two-sided $t$-tests against zero were conducted. In **f**, significant regions surviving multiple comparisons correction using cluster-based permutation statistics are outlined in black. Time zero indicates the onset of the CS in both time axes. AMY, amygdala; HPC, hippocampus; lPFC, lateral prefrontal cortex; OFC, orbitofrontal cortex.

TMP item stability was significantly higher for CS++ compared with CS+− trials ($t_{(40)}$ = 2.46, $p$ = 0.018; Cohen's $d$ = 0.5, CI: [0.08, 0.95]; Fig. 3e left), but no significant differences were observed between CS++ and CS−− trials ($t_{(40)}$ = 1.64, $p$ = 0.11; Cohen's $d$ = 0.39, CI: [−0.09, 0.91]; Fig. 3e middle) or between CS+− and CS−− trials ($t_{(40)}$ = −0.51, $p$ = 0.61; Cohen's $d$ = −0.12, CI: [−0.58, 0.34]; Fig. 3e right). No significant differences in item stability were observed in other ROIs during extinction (all $p_{corr}$ > 0.05).

We next investigated whether item stability was coordinated across brain regions within the extinction network, analogous to the coordination of item-specific representations between HPC and TMP during episodic memory retrieval[61]. Specifically, we tested whether trial-level metrics of item stability were correlated between AMY and TMP, where condition differences between CS+ and CS− items were observed, and between AMY or TMP and any other ROI. For each trial,

ROI and time point, we averaged the similarity of the representation of a particular item with the similarity of the same item in all other trials (Fig. 3f top left and Methods).

Item stability was significantly correlated between TMP and AMY (Fig. 3f bottom left): trials with high item stability in TMP also showed high item stability in AMY in two temporal clusters at the beginning of the cue period in both regions (0–0.95 s, $p_{corr}$ = 0.005) and shortly before the presentation of the US (0.9–1.75 s, $p_{corr}$ = 0.01). In addition, item stability was coordinated throughout the extinction network (Fig. 3f right): TMP item stability was correlated with item stability in HPC, lPFC and OFC (all $p_{corr}$ = 0.005). AMY item stability was correlated with item stability in HPC ($p_{corr}$ = 0.005), but not in prefrontal regions (lPFC: $p_{corr}$ = 0.375; OFC: $p$ = 0.25). Notably, most effects occurred around the diagonal in the temporal generalization map, demonstrating that item stability was coordinated at matching time periods across regions. However, the time windows at which item stability was coordinated varied across regions: while TMP–OFC correlations occurred early after cue onset, TMP–HPC and AMY–HPC correlations occurred across the whole time period of cue presentation, and TMP–lPFC correlations occurred close to the presentation of the US. In supplementary analyses, we tested whether item stability differed between CS+ and CS− trials, and between CS++ and CS−− trials during extinction, and observed no significant differences between these conditions (Supplementary Note 1). All relationships between all metrics are shown in the summary Fig. 8 (see below).

## Context-specific representations in HPC and lateral PFC

Rodent studies have described the crucial role of PFC, in coordination with HPC and AMY, in mediating the context dependency of extinction learning[3,10,17]. We investigated context-specific representations in these regions and across the extinction network by comparing the similarity of representations of same versus different contexts (Fig. 4a and 'Contrast-based RSA' in Methods). We analysed both the time period when only the context was shown and the subsequent time period when context and cue were shown conjointly, during both acquisition and extinction (Fig. 4).

During the time period when only the context was shown, we observed significant context-specific representations in the HPC during acquisition (0–500 ms; $p_{corr}$ = 0.005; Fig. 4b top left) and in the lPFC during extinction (50–300 ms, $p_{corr}$ = 0.025; Fig. 4b bottom right). No significant context-specific representations were observed in the other ROIs during acquisition or extinction (all $p_{corr}$ > 0.05).

During the subsequent period when both contexts and cues were shown, we did not observe any significant context-specific representations during acquisition in any ROI (all $p_{corr}$ > 0.05). During extinction, however, we observed significant context-specific representations in lPFC (0.8–1.15 s; $p_{corr}$ = 0.012), partially overlapping with the time period showing item stability in TMP (Fig. 4c middle). lPFC context specificity was significantly higher during extinction than during acquisition ($p$ = 0.001; Fig. 4c right). No other ROI showed context specificity in this time window during extinction (all $p_{corr}$ > 0.05).

To investigate whether context specificity was coordinated across the extinction network, we again performed a trial-level correlation analysis between the HPC, the lPFC and other ROIs, consistent with our analysis of item stability coordination (Fig. 3). Specifically, we computed a trial-level metric of context specificity by calculating the average difference in correlations between a given trial and other trials with the same context vs other trials with a different context (Fig. 4d left and Methods). We focused on the regions and time periods where we observed significant context-specific effects in our previous analysis: the HPC when only the video was shown during acquisition, and the lPFC when both video and cue were shown during extinction. We correlated the values observed in these two regions with those found in our other ROIs.

During acquisition, context specificity in the HPC was not coordinated with any other ROI (all $p_{corr}$ > 0.05). During extinction, we observed a significant coordination of context specificity in lPFC and TMP during an early time period (0–1.3 s) and between lPFC and AMY during a late time period (0.5–1.55 s; all $p_{corr}$ = 0.005; Fig. 4d right). Both periods overlapped with the time period showing significant context specificity in lPFC, and the time period of lPFC–AMY coordination also overlapped with the AMY theta power effect. Correlations between lPFC and other ROIs were not significant (all $p_{corr}$ > 0.05). We did not observe significant differences between conditions in the coordination analysis of context specificity (Supplementary Note 1).

Taken together, we found significant context-specific representations during both acquisition and extinction, but in different brain regions (HPC during acquisition, lPFC during extinction) and at different time periods (during acquisition: when the context was presented alone; during extinction: when the context was shown with the cue). During extinction, the magnitude of context specificity was coordinated across the extinction network between the lPFC and regions showing theta power differences (AMY) and/or differential item stability between CS+ and CS− trials (AMY and TMP). In a next step, we evaluated whether the different representational metrics were related.

## Links between theta power, item stability and context specificity

Our results so far indicate that, during the extinction phase, both sensory TMP regions and AMY represent CS+ cues in a more stable way than CS− cues. They also show that the lPFC exhibits context-specific representations in an overlapping time period. To uncover the relationship of AMY theta power ($AMY_{THETA}$), TMP/AMY item stability ($TMP_{ITEM}$ and $AMY_{ITEM}$) and lPFC context specificity ($lPFC_{CONTEXT}$), we investigated their correlation across trials and extended this analysis to item stability and context specificity effects in other ROIs during extinction (Fig. 5a). We extracted a single-trial metric of $AMY_{THETA}$ on the basis of the cluster of significant differences between CS+ and CS− items observed during extinction (Fig. 2; see schematic depiction in Fig. 5a left). We separately $z$-scored this metric for CS+ and CS− trials to avoid any spurious correlation of single-trial values driven by main condition differences. We first focused on the time period of significant AMY theta power effects and averaged item stability and context specificity across this time period in each trial and each ROI.

We observed that across trials, $AMY_{THETA}$ was correlated with both $AMY_{ITEM}$ ($t_{(31)}$ = −9.08, $p_{corr}$ < 0.001; Cohen's $d$ = −1.67, CI: [−2.2, −1.13]; Fig. 5b left) and $HPC_{ITEM}$ ($t_{(23)}$ = −5.61, $p_{corr}$ < 0.001; Cohen's $d$ = −1.13, CI: [−1.63, −0.61]; Fig. 5b right), but not with item stability in any other ROI (all $p_{corr}$ > 0.05). Notably, $AMY_{THETA}$ was also correlated with $lPFC_{CONTEXT}$ ($t_{(13)}$ = −4.21, $p_{corr}$ = 0.005; Cohen's $d$ = −1.06, CI: [−1.69, −0.41]; Fig. 5c), but not with context specificity in any other ROI (all $p_{corr}$ > 0.05). To assess the temporal specificity of these effects, we performed the same analysis across the entire cue presentation period, focusing on the ROIs where we observed significant effects (that is, AMY and HPC for item stability analysis and lPFC for context specificity). Notably, we observed significant correlations during the whole time period of cue presentation in AMY, which were most pronounced when the $AMY_{THETA}$ condition difference effects were observed (that is, from 1.18–1.75 s after cue presentation; $p_{corr}$ = 0.005; Fig. 5d left). Correlations between $AMY_{THETA}$ and $HPC_{ITEM}$ reached significance from 600 ms to 1.65 s ($p_{corr}$ = 0.005; Fig. 5d right), and correlations of $AMY_{THETA}$ with $lPFC_{CONTEXT}$ were observed in an intersecting time period (from 900 ms to 1.5 s after cue onset; $p_{corr}$ = 0.005; Fig. 5e). In supplementary analyses, we tested whether correlations of theta power and item stability or context specificity differed between CS+ and CS− trials, and between CS++ and CS−− trials during extinction, focusing on the regions where we observed significant effects in the analysis including all trials. Our results revealed no statistical differences between conditions (Supplementary Note 2).

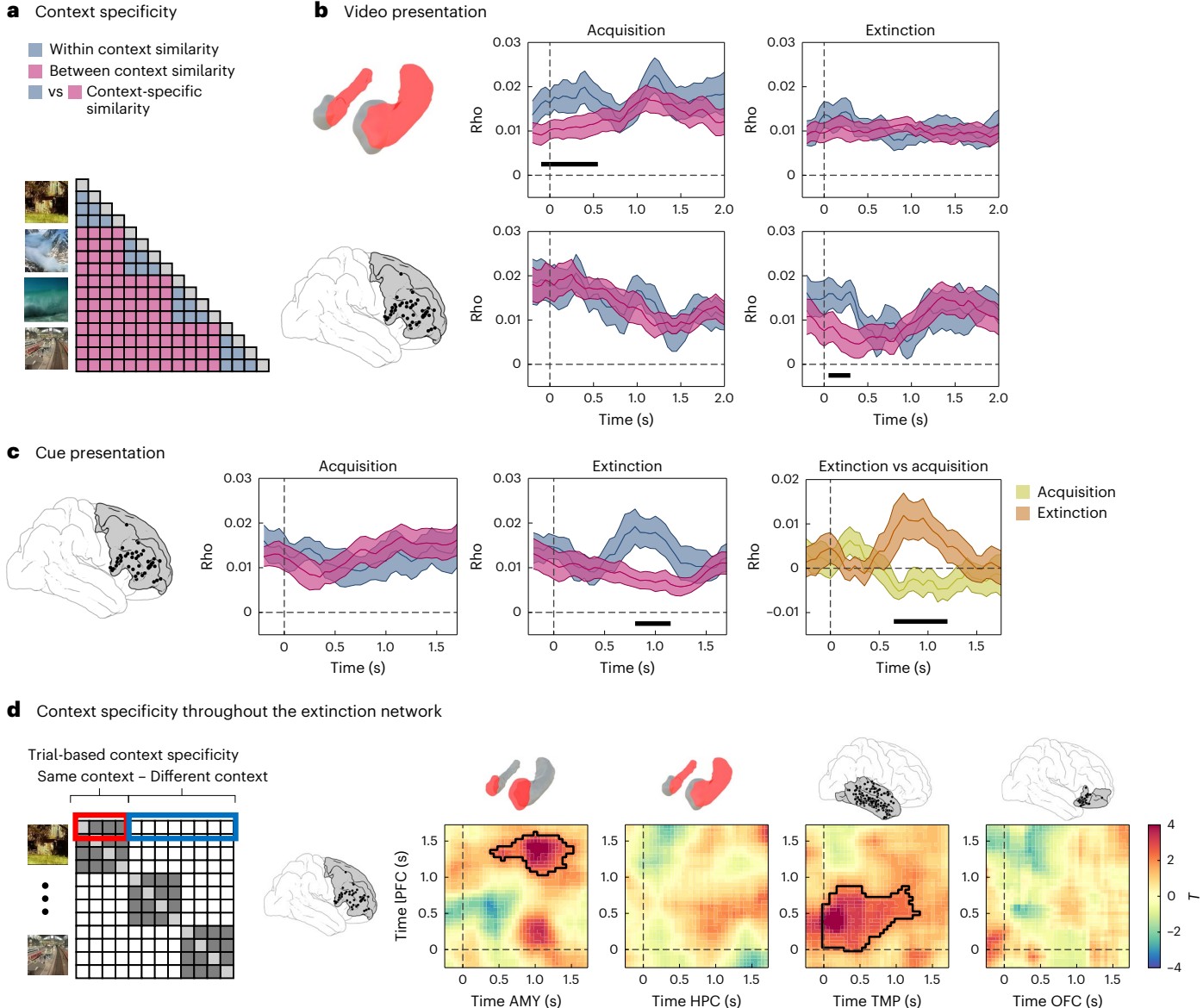

**Fig. 4 | Context specificity in HPC, lPFC and throughout the extinction network. a**, Context specificity analysis. Representational patterns corresponding to the presentation of individual context videos were compared across same contexts (blue) and different contexts (pink) during acquisition and extinction. Periods of context specificity were defined as those where same-context correlations were significantly higher than different-context correlations. **b**, Context specificity during video presentation in HPC (top) and lPFC (bottom). During acquisition (left), context-specific representations were observed in the HPC locked to the onset of the video ($p_{corr} = 0.005$). During extinction (right), a similar effect was observed in the lPFC ($p_{corr} = 0.025$). **c**, Context specificity during cue presentation. Left: during acquisition, no significant differences were observed between same and different contexts. Middle: during extinction, context-specific representations were observed

at the time period from 0.8 to 1.15 s ($p_{corr} = 0.012$). Right: context specificity during extinction (brown) was significantly higher than during acquisition (green; $p = 0.001$). **d**, Left: trial-specific values of context specificity, defined as average same minus different-context correlations in each individual trial, were computed in each of our ROIs. Right: focusing on the lPFC, we correlated context specificity values between regions and across trials. Significant regions surviving multiple comparison corrections using cluster-based permutation statistics are outlined in black. In **b** and **c**, black horizontal lines at the bottom of each panel depict significant time periods after correction for multiple comparisons using cluster-based permutation statistics. Time zero indicates the onset of the CS, and shaded lines depict group average rho values ± s.e.m. Two-sided paired *t*-tests were applied at every time point (**b** and **c**) or every time-by-time point (**d**).

Taken together, these results demonstrate that AMY$_{THETA}$ was correlated with both AMY and HPC item stability and lPFC context specificity during a late time period of cue presentation.

## LPFC context specificity predicts reinstatement of fear memory traces

Both rodent studies[3,17] and clinical studies in humans[70–73] suggest that high context specificity during extinction predicts renewal, that is, a reoccurrence of fear memories in acquisition contexts or novel

contexts. We thus quantified the degree to which the representations of the three cue types during acquisition and extinction reappeared during the test phase. Specifically, we computed the similarity of every trial during acquisition (or extinction) with all trials in which the same item was shown during the test phase. Averaging these values across trials separately for each cue type (CS++, CS+−, CS−−) and subtracting acquisition reinstatement and extinction reinstatement yielded a participant-specific metric of fear reinstatement (REINST = REINST-$_{ACQ}$ − REINST$_{EXT}$; Fig. 6a). We correlated these reinstatement values in

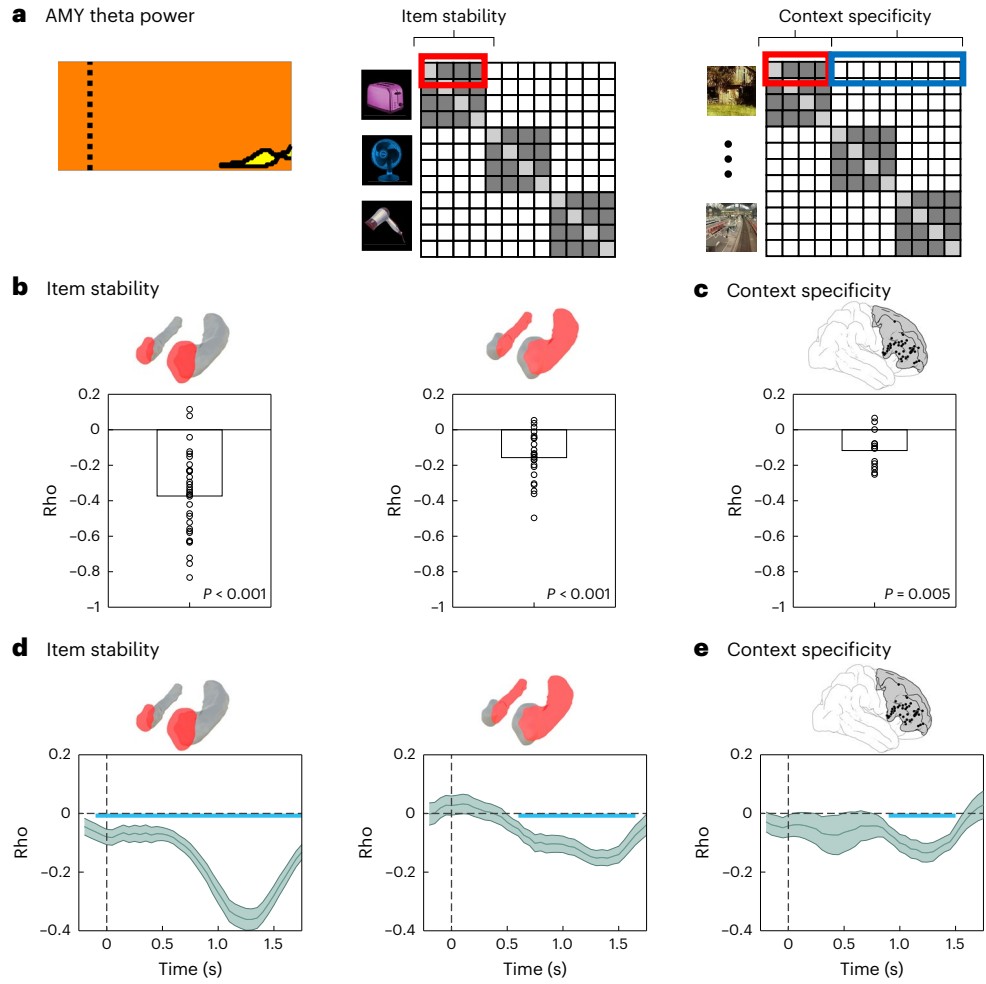

**Fig. 5 | Amygdala theta power correlates with item stability and context specificity across the extinction network. a**, Left: AMY theta power was averaged within the time–frequency cluster showing significant differences between CS+ and CS− items during extinction (highlighted in yellow). This average theta power was correlated across trials with two measures: item stability (middle) and context specificity (right). These metrics were either averaged during the same time period of significant AMY theta power effects (**b,c**), or in sliding windows across all time periods in each trial (**d,e**). Group-level rho values were contrasted against zero using two-sided *t*-tests in all analyses. **b**, AMY theta power was correlated across trials with item stability in both AMY (left; $p_{corr} < 0.001$; $N = 32$) and HPC (right; $p_{corr} < 0.001$; $N = 24$). **c**, AMY theta power was also correlated with lPFC context specificity across trials ($p_{corr} = 0.005$;

$N = 14$). **d**, Time-resolved analysis. Left: AMY theta power and item stability were correlated throughout the entire cue presentation period ($p_{corr} = 0.005$). Right: correlations between AMY theta power and HPC item stability were observed during a later time window of cue presentation (600 ms–1.65 s; $p_{corr} = 0.005$). **e**, AMY theta power was correlated with lPFC context specificity during a time period that overlapped with the significant condition differences in theta power in AMY (900 ms–1.5 s; $p_{corr} = 0.005$). In **d** and **e**, shaded lines depict group average rho values ± s.e.m., and time zero marks the onset of cue presentation during the extinction phase. Blue horizontal lines indicate time periods of statistical significance after correction for multiple comparisons using cluster-based permutation statistics. In **a**, the dotted vertical line represents cue onset (time zero).

AMY and TMP, where the main effects of item stability were observed, with participant-averaged metrics of context specificity in lPFC during the extinction phase (Fig. 6b). We focused on the time period when significant context-specific effects were observed.

As hypothesized, we observed a positive correlation between context specificity in lPFC and reinstatement in TMP, indicating more pronounced reinstatement of acquisition than extinction memory traces in participants with more pronounced context specificity during extinction. Interestingly, this effect occurred specifically for CS+− trials whose contingencies changed from acquisition to extinction (rho$_{(15)}$: 0.57, $p = 0.024$; Fig. 6c middle), but not for CS++ (rho$_{(14)} = −0.13$, $p = 0.3$; Fig. 6c left) or CS−− trials (rho$_{(15)} = 0.4$, $p = 0.13$; Fig. 6c right). lPFC context specificity did not correlate with AMY reinstatement for any cue type (CS++: rho$_{(13)} = −0.01$, $p = 0.98$; CS+−: rho$_{(13)} = 0.029$, $p = 0.92$; CS−−: rho$_{(13)} = 0.28$, $p = 0.32$). This effect was not observed when we correlated lPFC context specificity with either TMP REINST$_{ACQ}$ or REINST$_{EXT}$ separately for any of our three trial types, suggesting that it specifically

affects the balance (or competition) between reinstatement of acquisition vs extinction memory traces (see Discussion).

In complementary analyses, we evaluated whether overall levels of acquisition-to-test, extinction-to-test and differential (acquisition-to-test minus extinction-to-test) reinstatement differed across trials, and whether for each cue type, acquisition-to-test and extinction-to-test reinstatement differed in the TMP and the AMY. None of these analyses revealed significant effects (Supplementary Note 3 and Extended Data Fig. 1).

## Reinstatement of extinction memory traces predicts safety responses

In our final analysis, we evaluated whether reinstatement of extinction memory traces during the test phase predicted subjective ratings of safety. For every item during the test phase, similarity was calculated with all instances of this item during extinction, and these values were correlated across trials with the ratings during the test phase (Fig. 7a).

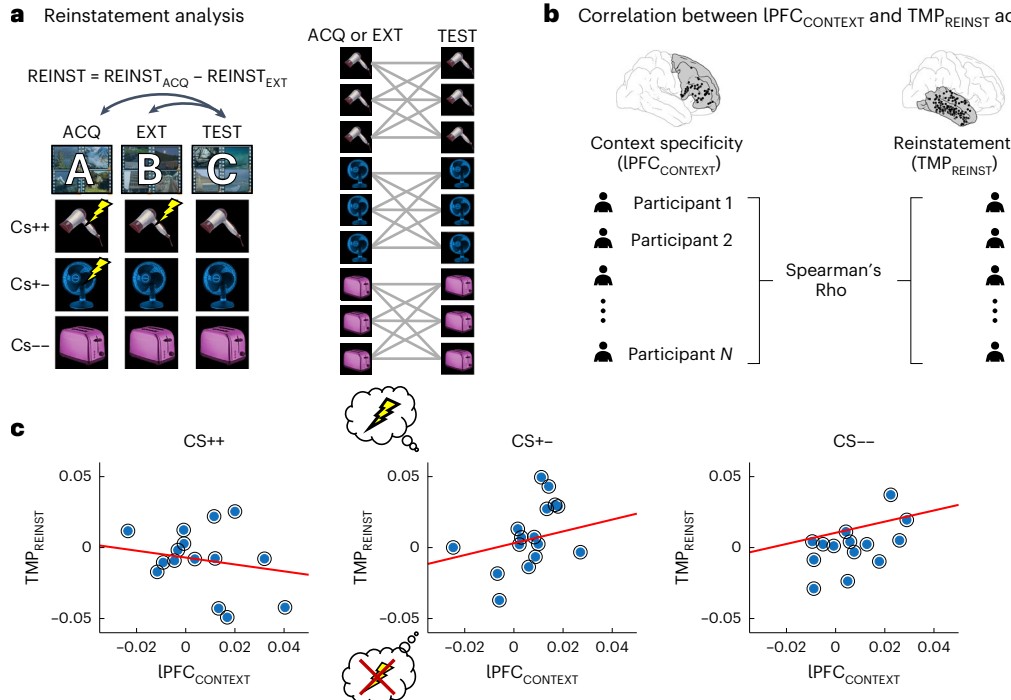

**Fig. 6 | Context specificity shifts the balance in reinstatement of fear vs extinction memory traces. a**, Reinstatement analysis. Reinstatement of acquisition and extinction activity patterns during the test was calculated separately for CS++, CS+− and CS−− items (left), by computing the similarity across repeated exposures to the same item across experimental phases (right). Acquisition and extinction reinstatement values were subtracted and averaged across trials, resulting in a single value of differential reinstatement in each participant and condition (REINST = REINST$_{ACQ}$ − REINST$_{EXT}$). **b**, Context

specificity in lPFC and reinstatement in TMP were correlated across our group of participants using Spearman correlations. **c**, lPFC context specificity and TMP reinstatement were correlated during the time period of significant context-specific effects in lPFC, specifically in CS+− trials ($p = 0.024$; $N = 16$). Note that for CS+− trials, positive values of TMP reinstatement indicate predominant reinstatement of fear memory traces, while negative values indicate predominant reinstatement of extinction memory traces.

We found that reinstatement of extinction memory traces was positively correlated with subjective safety ratings for CS+− cues ($t_{(18)} = 2.99$, $p = 0.0078$; Cohen's $d = 0.66$, CI: [0.17, 1.13]; Fig. 7b middle), while this was not the case for CS++ ($t_{(20)} = 0.36$, $p = 0.72$; Cohen's $d = 0.74$, CI: [−0.34, 0.49]; Fig. 7b left) or CS−− cues ($t_{(22)} = −0.16$, $p = 0.87$; Cohen's $d = −0.03$, CI: [−0.43, 0.36]).

## Discussion

Here we investigated the neurophysiological mechanisms and representational signatures of extinction learning in the human brain. CS− items during extinction were associated with increases in theta power in the AMY and lower levels of item stability in the TMP. Moreover, extinction learning showed higher context specificity in lPFC, which predicted the reinstatement of fear memory traces in TMP. Further analyses unravelled the distinct pattern and time course of coordinated representations across the fear and extinction network, which is summarized in Fig. 8.

### A safety signal during extinction in the AMY

While a previous human iEEG study reported increases in theta power for CS+ vs CS− items during acquisition[30], our results showed higher theta power for CS− vs CS+ items during extinction. This apparent discrepancy may be explained by the differing cognitive demands of acquisition and extinction, with the latter requiring pronounced coordination between AMY and other brain regions coding for contexts and cognitive control such as HPC and PFC. Indeed, increases in AMY theta power during extinction were correlated with higher levels of item stability in both AMY and HPC, and with higher levels of context specificity in lPFC. The lack of AMY theta power effects during acquisition in our study contrasts with findings in animals and humans, which

have typically reported theta increases during the formation of cue–US associations across several regions of the fear and extinction network (for example, refs. 30,31). Several factors may explain this discrepancy, including differences in recording methodologies (iEEG versus EEG), the distinct regions targeted (for example, different subregions of the PFC targeted in ref. 30), interspecies differences in theta oscillations[74], and the use of a more 'cognitive' paradigm in our study with a milder aversive stimulus.

In our paradigm, both CS+− and CS−− trials signal safety during extinction, either newly learned safety following extinction (for CS+− trials) or consistent safety throughout the experiment (for CS−− trials). Thus, the successful extinction of previous fear should result in neural activity and representations of CS+− items that match those of CS−− items, in line with our findings on both AMY theta power, and AMY and TMP item stability.

Notably, the increase in theta power for CS− items observed during extinction, coupled with the absence of such increases during acquisition, suggests that theta power does not simply signal safety but rather safety within the specific context of extinction. At least two key differences distinguish acquisition from extinction in our paradigm: (1) contingencies changed for one of the cues (CS+−) and (2) contexts changed from A to B. While both of these factors may have influenced theta responses, future research is needed to determine whether theta power during extinction signals safety specifically when both context and contingencies change in the context of a multicue paradigm such as ours, or whether it would also be observed in extinction learning paradigms in which contexts do not change (for example, AAB designs) or in those including only one cue (CS+−).

We note that in our AMY theta power analyses, some time–frequency bins observed early in the trial (~300–500 ms) showed

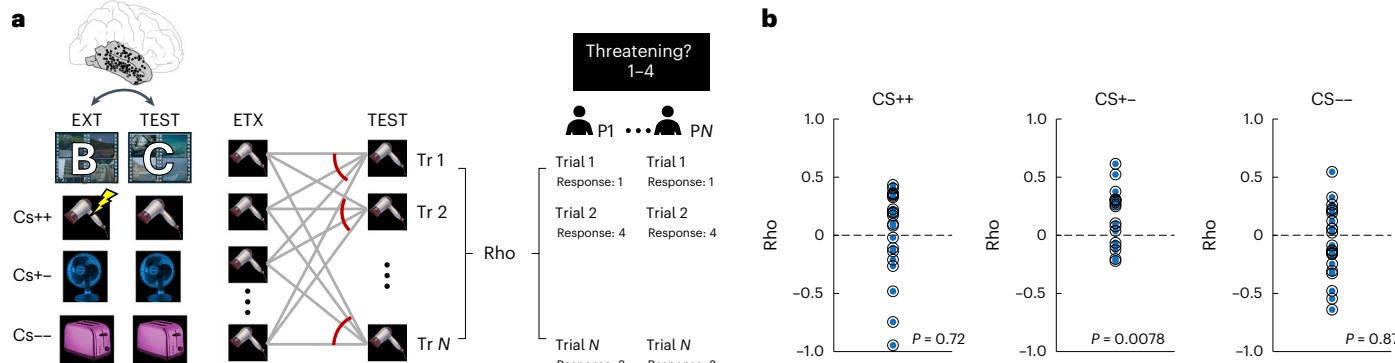

**Fig. 7 | Reinstatement of extinction memory traces in TMP predicts subjective safety ratings. a**, Extinction-to-test reinstatement analysis: for each trial type (CS++, CS+− and CS−−), the similarity between extinction and test trials was computed in TMP (left). This was done by averaging the similarity of one trial during the test across repeated presentations of this same item during the extinction phase (middle). Spearman correlations were then computed between extinction-to-test similarity and subjective ratings during the test across trials in each participant independently (right). **b**, Fisher z-transformed correlations were analysed at the group level to determine whether they significantly differed from zero using two-sided t-tests separately for CS++ (left; N = 21), CS+− (middle; N = 19) and CS−− (right; N = 23) trials.

numerical increases in theta power for CS+ items at an uncorrected level. This might seem surprising, as it contrasts with the main effect observed later in the trial (1.18–1.75 s) which reflects a decrease in theta power for threatening items. However, these numerical increases are within the range of what is expected by chance, as assessed by our cluster-based permutation analysis. While our primary analyses focused on theta frequency oscillations, humans are able to rapidly distinguish and respond to threats and safety cues[75]. This raises the possibility of a potential role of high-frequency neural activity in signalling threat and safety. Indeed, theoretical accounts suggest that fast-cycle gamma oscillations might promote fast information transfer between brain regions[76], facilitating fear learning by modulating the timing of neuronal firing inducing plasticity[77]. In particular, gamma cycles align with several biophysical factors regulating excitatory input integration in basolateral amygdala principal neurons[78], and experimental studies in rodents have shown that amygdala gamma oscillations contribute to fear memory formation through theta–gamma coupling and spike-field coherence[79,80]. While we did not observe any effect of gamma frequency oscillations in our paradigm, we believe further research will be needed to fully assess their possible role in signalling threat and safety, and modulating fear learning in humans, for example, via phase amplitude coupling[81]. Interestingly, recent findings suggest that hippocampal representational patterns linked to amygdala gamma bursts reinstate the content of emotional memories[50], providing evidence for a potential role in the representation of fear in the interaction with the hippocampus.

The high temporal resolution of iEEG recordings allowed us to shed light on the coordinated time courses of representational signals across regions of the extinction network. Our results reveal that the representation of specific contexts in the AMY and the lPFC are consistently correlated across trials. Notably, this coordinated representation coincides with the safety-related theta power increase in AMY. These results support previous research showing that distributed areas of the extinction network, including the PFC, coordinate the processing of contextual information in the AMY[82,83].

Notably, the correlation of lPFC and AMY context specificity also overlapped with the period of lPFC context specificity (from 0.8 to 1.15 s after cue onset). However, it also shows a delay between lPFC and AMY, with AMY context specificity emerging ~1 s earlier. This temporal lag seems to decrease during later trial periods. The main effect of lPFC context specificity occurs immediately before the time period when the strongest coordination is observed with the AMY (at −1.2 s). Although the AMY did not show a main effect of context specificity, possibly because the similarity of same-context representations was attenuated by repetition suppression, the observed temporal offset suggests that the effect observed in the lPFC drives the AMY representations. Our findings underscore the relevance of dynamic and distributed context representations throughout the extinction network during fear extinction. More specifically, they suggest that AMY safety signals and representations during extinction depend on the preceding representation of specific contexts in lPFC.

### The context dependency of extinction learning modulates fear renewal
Our findings also shed light on the functional role of lPFC context representations in modulating the reinstatement of fear vs extinction memories. Reinstatement of memory traces has previously been analysed via encoding–retrieval similarity[61–63]. In the context of fear extinction, previous fMRI studies investigated the reinstatement of fear and extinction memory traces using multivoxel pattern similarity analysis[84]. Following a similar approach, here we implemented a novel metric of reinstatement that directly compared the reoccurrence of fear vs extinction memory traces, assuming an inhibition of these two memories during test[85–88]. Using this metric, we observed that higher fear vs extinction memory reinstatement in TMP was correlated with context-specific representations in the lPFC, that is, in participants with elevated levels of context signalling in the lPFC during extinction, the balance between fear memory reinstatement and extinction memory reinstatement was shifted towards the former (Fig. 6). Importantly, this effect was selectively observed for CS+− cues, but not for CS++ or CS−− cues, which are unlikely to compete between acquisition and extinction. Our finding aligns with clinical observations that high levels of context dependency during extinction reduce the persistence of the safety responses learned during extinction and favour the recurrence of fear[86]. One may assume that if extinction contexts are represented more specifically, the newly formed extinction memories are considered as an exception, and the original fear memory is likely to come back. Our results further show that aversive memories are extinguished through an interaction between executive control (lPFC) and sensory regions (TMP) representing contexts and items, respectively. Possibly, more pronounced context representations in lPFC involve effortful inhibition processes similar to those during episodic memory control[89], which are difficult to maintain in the longer run.

### Fear learning, extinction and episodic memory
Our results highlight the crucial role of both HPC and lPFC in encoding contextual information during fear learning and extinction, consistent with findings in rodents[3,6,17]. Notably, the lPFC was selectively engaged

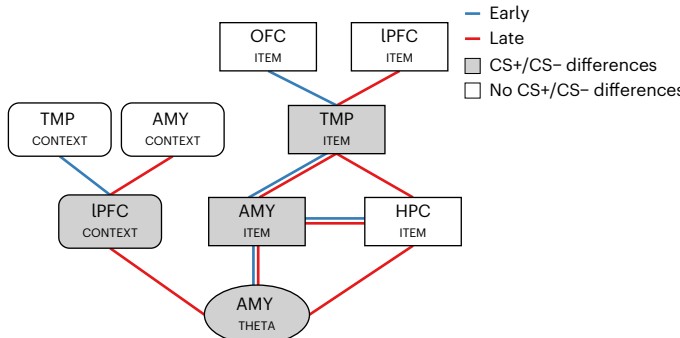

**Fig. 8 | Summary of the main results during extinction learning.** Summary of the main results obtained in the power analysis, the item stability analysis and the context specificity analysis during the extinction period across our five ROIs. During the time period of theta power condition differences in the amygdala (CS+ vs CS−; labelled 'AMY$_{THETA}$' in the figure), we also observed lPFC context-specific effects ('lPFC$_{CONTEXT}$') and item stability effects in AMY and lPFC ('AMY$_{ITEM}$', 'lPFC$_{ITEM}$'). Note that AMY$_{THETA}$ effects occurred during a late time period (indicated with red lines in the figure), but AMY$_{THETA}$ and AMY$_{ITEM}$ were also correlated earlier in the trial (indicated with blue lines in the figure). In turn, AMY$_{ITEM}$, TMP$_{ITEM}$ and HPC$_{ITEM}$ effects were observed during early and late time periods. AMY$_{CONTEXT}$ was correlated with lPFC$_{CONTEXT}$ during a late time period only, but lPFC$_{CONTEXT}$ was correlated with TMP$_{CONTEXT}$ during an earlier time period. Metrics showing main condition differences between CS+ and CS− items during extinction are highlighted in grey.

in the representation of context-specific activity during extinction and not during acquisition, again consistent with its involvement in the suppression of fear memories via context signalling. While we observed a context-specific representation during acquisition as well (in the HPC), this effect did not coincide with the time period of the cue presentation, suggesting that specific contexts are not neglected during acquisition but exert a less prominent influence on the memory traces of individual cues. While the lPFC effects align with the role of this region in inhibiting fear expression[3,17], the HPC responses are consistent with the greater sensitivity of this region to views of landscapes and scenes[90,91], and the well-acknowledged role of the HPC in rapidly encoding new memory traces during acquisition[92]. We note that several studies have shown that fear acquisition is characterized by an overgeneralized and decontextualized fear responses, while extinction engages PFC-dependent mechanisms more reliant on context representation (for reviews, see refs. 3,17). We previously found that overgeneralization of representations predicted subsequent intrusive memories[93], which are notorious for their decontextualized nature and the fact that they can be easily and involuntarily triggered by ubiquitous sensory cues.

Our findings suggest that extinction memory traces involve neurophysiological patterns and representational characteristics that differ from those formed during acquisition and are more reminiscent of episodic memory traces (see also ref. 84). First, CS− cues were associated with increased theta oscillations during extinction. Previous studies on the role of theta oscillations for episodic memory formation suggest that theta power increases particularly for associative, that is, context-dependent, memories, while the encoding of item memories is commonly associated with theta power reductions[48]. Second, the increased levels of lPFC context specificity during extinction are reminiscent of the formation of new episodic memories, which are defined, among other factors, by their dependence on context[94,95]. These findings support the notion of fear acquisition as a form of associative learning that lacks contextual specificity, while fear extinction, similar to episodic memory, is more strongly influenced by contextual representations. Third, memory traces of CS− items during extinction were less stable and, therefore, more trial-specific in sensory regions of the TMP, reminiscent of the unique and distinct representations

during individual trials observed in episodic memory paradigms. Indeed, while previous studies have linked the stability of neural representations to both episodic memory encoding[67,68] and the formation of fear associations[69], others have suggested that item stability is modulated by contextual factors and is disrupted when incongruent contextual information is presented[96]. To summarize, these findings point towards the flexible and malleable nature of safety memory traces during extinction; and indeed, higher level of context dependency (and thus putatively higher 'episodicity' of extinction memories) predicted their relative weakening compared with the more robust and generalized initial fear memory traces.

While both extinction learning and episodic memory are tightly linked to contextual information, these two forms of associative learning certainly differ in other respects. Episodic memory is specifically defined by its connection to 'mental time travel'—the ability to mentally project oneself into the past or future and to reconstruct specific experiences[94], a feature that is not evident in fear extinction paradigms. Moreover, fear learning and extinction typically occur across repeated trials, while episodic memory has been characterized as one-shot learning[92]. Despite these differences, our data show that extinction learning shares representational signatures with item–context associations in episodic memory[16,61].

## Methodological considerations

Cognitive impairments in implanted epilepsy patients are generally a concern when conducting cognitive experiments using iEEG data. To facilitate the interpretation of our findings, we provide in Extended Data Table 1 all available demographic and clinical information of our patients, including global intelligence quotient (IQ), memory quotient (MQ; assessed via the Wechsler Memory Scale, WMS-III) and seizure onset zone (SOZ). While the IQ scores were within the normal range, the patients in our study showed a lower-than-average MQ. Although our paradigm did not directly probe the specific memory functions assessed by the WMS, this limitation is inherent to intracranial EEG recordings in epilepsy patients and should be acknowledged. While we cannot entirely rule out the influence of memory impairments on our results, the simplicity of our task design (with only three cues and a single instruction for all trials) and the fact that learning curves align with expected behavioural patterns partially alleviate this concern. In addition, fear associations can be formed in the absence of conscious awareness of the CS–US contingencies[97,98], suggesting that explicit memory impairments may not necessarily prevent the formation and extinction of fear associations. Indeed, our trial-by-trial progressive learning analysis confirms that patients successfully understood the task instructions and appropriately learned the CS–US contingencies, adaptively adjusting their responses to contingency changes in CS+− trials (see Fig. 1d).

We note that, similar to other fear conditioning paradigms conducted with humans (for example refs. 99–106), we chose to rely on self-reported ratings of cue 'threat' and 'safety' as our behavioural measure of fear rather than employing implicit physiological recordings. While explicit ratings have limitations, we decided to employ them on the basis of both ethical and clinical considerations, particularly given the constraints of conducting research with implanted epilepsy patients. Specifically, the use of a mild aversive stimulus (a scream) rather than electric shocks in our study precluded the use of some traditional implicit measures, such as skin conductance or heart rate responses, which are more suited for capturing stronger autonomic reactions. Importantly, research indicates that explicit ratings are more stable than physiological responses across repeated measurements[107], and several reviews support their validity in fear conditioning research[108,109]. Explicit self-reports have been shown to correspond with both fear startle[99,102] and skin conductance responses[106,109,110]. Moreover, self-reports might better capture deliberative processes than implicit measures[111], which aligns with the more 'cognitive' nature

of our paradigm (see also refs. [101],[103]). Although explicit ratings are susceptible to demand effects or participant interpretation, such effects are unlikely to have played a major role in our results due to the simplicity of the task and the absence of explicit experimenter expectations beyond learning the correct CS–US associations. Crucially, participants were instructed to rate the 'safety' of the cue itself rather than their internal emotional state, minimizing subjective bias. Furthermore, apart from analysing the correlation of the ratings with reinstatement in the TMP (Fig. 7), we did not rely on the ratings to categorize stimuli or correlate them with the neural data in other analyses, which minimizes their influence in the overall interpretation of our findings. Finally, we highlight that the trial-by-trial learning effects we observed on the basis of the ratings align closely with what would be expected for skin conductance responses (see for example, ref. [112]).

Our study was not preregistered and we acknowledge this as a limitation, as preregistration enhances the transparency and reproducibility of research findings. However, the exploratory nature of our study, which investigates context-dependent extinction learning in humans via iEEG, made preregistration challenging. Our research aimed to provide a characterization of the representational dynamics underlying the context-dependent nature of extinction learning, which required the integration of methodologies from diverse domains and the development of novel metrics. For example, we adapted RSA, an established method in episodic memory research, to examine context specificity and item stability within a classical ABC context-dependent extinction paradigm. We also developed a new measure to quantify fear reinstatement in the context of classical conditioning (see Fig. 6). Despite the pioneering and exploratory nature of our study, we note that we formulated specific predictions in the introduction and results sections.

Notably, some of our results were not in line with our predictions. Specifically, we anticipated to observe increased theta power for CS+ as compared with CS− items during acquisition in AMY, which we did not find. Instead, we observed a relative decrease in power for these trials during extinction in this region (see section 'A safety signal during extinction in the AMY'). We also expected context-specific signals both in lPFC and HPC during extinction, on the basis of previous literature[3]. However, context specificity was only observed in the lPFC during extinction, while the involvement of the HPC was only observed during acquisition (see section 'Fear learning, extinction and episodic memory' above). Finally, we expected lPFC context specificity to drive context representations in AMY and HPC, but while lPFC and AMY context representations were indeed correlated across trials, we did not find a main effect of context specificity during extinction in either AMY or HPC. Despite its exploratory nature, our work represents an important first step in exploring the representations underlying context-dependent extinction learning in humans using iEEG, providing a foundation for more targeted, hypothesis-driven research in the future. In summary, our findings shed light on the key neurophysiological mechanisms, representational characteristics and functional relevance underlying context-dependence fear extinction in the human brain. They bridge the gap between animal and human studies, contribute evidence to a novel conceptual framework that integrates fear conditioning and episodic memory research, and provide a mechanistic basis for understanding the renewal of fear in clinical contexts.

## Methods

### Participants
Forty-nine patients (22 females, 28.2 ± 8.18 years) with medically intractable epilepsy participated in the study. Data were collected at the Pitié Salpêtrière Hospital, Paris, and the South China Normal University Hospital, Guangzhou, China. The study was conducted according to the latest version of the Declaration of Helsinki and approved by the corresponding ethical review boards (Paris: Institutional Review Board

at the Pitié Salpêtrière Hospital, C11–16, C19–55 National Institute of Health and Medical Research sponsor; Guangzhou: Ethics Committee of the School of Psychology at South China Normal University). All patients provided written informed consent and did not receive any monetary compensation for participating in the study, in line with ethical regulations at the hospitals where the experiment was conducted.

### Experimental design
Participants saw three different images of electrical devices (a hair dryer, a ventilator and a toaster) during acquisition, extinction and test. During acquisition, two out of these three images were associated with an aversive unconditioned stimulus (US) and labelled as 'CS+', while the remaining image was not associated with the US and labelled 'CS−' (Fig. 1a top). The association of the images with the US was counterbalanced across participants. Each image was presented 24 times during acquisition and extinction in a pseudo-randomized fashion to avoid patients anticipating the images and the US (total number of trials in a block: 72). Specifically, a maximum of three consecutive presentations of the same item was allowed, irrespective of their particular contingencies. Every cue was presented in one of many different contexts (that is, videos), which were not related to the type of trial and the occurrence of the US. Four unique context videos were presented during acquisition and another four were presented during extinction. Critically, during extinction, the CS type of one of the items was modified, while the other two remained the same. This led to the distinction among three different types of item: CS++, CS+− and CS−−. After the extinction period, participants performed a test in which they again saw the three items during 16 trials each (total number of trials during the test: 48), which were associated with 8 new experimental contexts (the number of contexts was doubled to match those presented during acquisition and extinction). In the test phase, no US was presented for any of the items. The association of the contexts and the cues was completely randomized in every participant so that contexts were not predictive of the CS. Please note that this approach differs from conventional fear and extinction learning paradigms. Our design was developed to leverage the power of RSA and iEEG to dynamically track the representation of contexts across key hubs of the extinction network. On the other hand, our paradigm aligns with conventional ABC paradigms as it contains three different context categories across experimental phases (see Supplementary Note 4). Similar to the cue–CS associations, the context videos were counterbalanced across participants and experimental phases. Following previous literature, the reinforcement rate of the CS+ items was set to 50% both during acquisition and extinction to avoid fast associative learning and extinction of the cue–CS associations[113].

At every trial, in each experimental phase, participants first saw a fixation cross for 0.5 s, and then a context video was presented for 2 s. After the video presentation, a cue was overlaid on top of the video for 1.75 s, and then the US was presented for 1 s. Finally, a question was presented about how much participants had expected a US, that is, how threatening or safe they had perceived the CS, and participants provided their responses on a scale from 1 (threatening) to 4 (safe).

The US consisted of an unpleasant scream, which was delivered via the computer speakers, paired with a negative facial expression (see Supplementary Note 4). The sound level of the scream was ~85 decibels, resulting in a highly aversive but not painful amplitude. The duration of the scream was 1 s. Given the 50% reinforcement rate of the US, the scream was presented a total of 36 times in the experiment (CS++: 24 times across acquisition and extinction; CS+− 12 times only during acquisition).

The experiment was programmed in Presentation (Neurobehavioral Systems) and was deployed on a laptop computer running Microsoft Windows. Patients performed the experiment while sitting in their hospital beds and responded to the memory test using the keyboard of the laptop computer.

Please note that our experimental design is grounded in theoretical considerations from the human fear learning literature and has been adapted to meet the specific requirements of conducting research in implanted epilepsy patients. While its specific combination of features is novel and has not been fully validated, we provide a detailed rationale for each of our design choices, highlighting their similarities to and differences from previous research in the field, in Supplementary Note 4.

## Behavioural analysis

We conducted two main behavioural analyses. In the first analysis, we examined average ratings across trials within each experimental phase to identify differences in the ratings between phases. A two-way repeated measures ANOVA was used, with trial type and experimental phase as factors. Post hoc tests with Bonferroni correction were applied across conditions after confirming that both a main effect and an interaction were observed (Fig. 1). Due to a technical issue during the test phase, one participant was excluded from this analysis, resulting in a sample size of 48 participants.

In the second analysis, we sorted trials according to their type (CS++, CS+− and CS−−) and compared the ratings for each trial type at every position. Since the comparisons were performed on single-trial data, group-level distributions were not normal. Thus, we employed the non-parametric Wilcoxon signed-rank test to assess whether conditions differed at every trial. This was done to evaluate the progressive learning of the trial contingencies during acquisition and extinction. In this analysis, we sorted trials by type (CS++, CS+− and CS−−) and performed a Wilcoxon signed-rank test across participants in each trial to compare performance across conditions. While sorting trials in this way disrupted the absolute trial positions in the experiment, the sequential order of trials within each condition was maintained. For a specific trial (for example, trial 9), the CS type could be different across participants. However, after sorting by condition, the first CS++ in all participants was always compared with the first CS+− and the first CS−− trial, the second with the second and so on. Thus, the paired Wilcoxon tests were performed across participants in individual trials corresponding to the same ordinal position in each condition. We corrected for multiple comparisons using cluster-based permutation statistics, shuffling the condition labels at every trial.

## Intracranial EEG recordings

In Paris, recordings were conducted using the ATLAS amplifier (Neuralynx) with a total of 160 channels and a sampling rate of 4,096 Hz (ref. 114). In Guangzhou, data were collected using a 256-channel Nihon Kohden Neurofax 1200A Digital System, sampled at 2,000 Hz. The number of electrodes and their implantation sites were determined exclusively by clinical requirements. For all analyses, data were downsampled to 1,000 Hz. Bipolar referencing was applied, where the activity at each contact point was subtracted from that of the nearest contact on the same electrode, resulting in a total of $N-1$ virtual channels for an electrode with $N$ channels after re-referencing.

## Channel localization

Stereotactically implanted electrodes varied in type, number of contacts and intercontact distances. In Paris, two types of electrode were used: standard macro electrodes and macro–micro electrodes. The standard macro electrodes were depth electrodes (AdTech Medical Instrument) consisting of 4–12 platinum contact points, each with a diameter of 1.12 mm and a length of 2.41 mm, connected via nickel–chromium wiring. The centre-to-centre distance between adjacent contacts was 5 mm. The macro–micro electrodes (also from AdTech) were of the Behnke–Fried type, consisting of 8 platinum contact points, each 1.28 mm in diameter and 1.57 mm in length, mounted on the surface of a polyurethane tube with a hollow lumen. In addition, eight 40 μm platinum–iridium microwires were embedded within the macroelectrode, extending 3–6 mm beyond the tip into the surrounding cerebral tissue (data from these microwires were not analysed in this study). The intercontact distance between contacts 1 and 2 was 3 mm, while the distance between subsequent contacts (for example, 2 and 3, 3 and 4 and so on) was 7 mm.

In Guangzhou, depth electrodes were obtained from Sinovation Medical Technology and consisted of platinum depth electrodes with 7–19 contacts each (0.8 mm diameter, 2 mm length and 1.5 mm spacing).

Electrode localization was conducted separately by our clinical teams in Paris and Guangzhou using very similar methodological approaches. At both sites, post-implantation computed tomography (CT) images were co-registered onto pre-implantation magnetic resonance imaging (MRI) data. Both sites used the same software including FreeSurfer (http://surfer.nmr.mgh.harvard.edu/), 3DSlicer (https://www.slicer.org) and Statistical Parametric Mapping (SPM; https://www.fil.ion.ucl.ac.uk/spm/).

In Paris, electrode coordinates were extracted using the EPILOC toolbox developed on the STIM platform (Stereotaxy, Techniques, Images, Models; http://pf-stim.cricm.upmc.fr). EPILOC automates several image processing steps using FreeSurfer, 3DSlicer and SPM, complemented by BrainVISA[115]. Anatomical models were generated from pre-operative MRI data using FreeSurfer. Post-operative CT data within the patient's native space were co-registered onto the pre-operative MRI data and then normalized to a Montreal Neurological Institute (MNI) template using SPM. Depth sEEG electrodes were automatically localized in post-operative CT images by segmenting electrode artefacts and classifying them on the basis of their proximity to trajectories planned by stereotactic guidance devices.

In Guangzhou, channel locations were again identified by co-registering the post-implantation CT images to pre-implantation MRIs acquired for each patient. Patient-specific anatomical surfaces were generated with FreeSurfer and normalized to MNI space using SPM. Electrode locations were determined using 3DSlicer.

A table containing the electrode coordinates in MNI space for all contacts included in the study is presented in Supplementary Table 1. We note that since the implantation scheme is solely determined by clinical criteria, density of sampling within and across brain regions is inherently variable in iEEG research. Several factors might affect the volume of neural tissue sampled, for example, different electrode configurations, individual implantation schemes and the selection of ROIs. We performed several control analyses to corroborate that the volume of neural tissue sampled did not differ between our two patient cohorts (see Supplementary Note 5 and Extended Data Fig. 2).

## ROI selection

We recorded activity from several brain regions pertaining to the fear learning and extinction network, including the amygdala, hippocampus, the lateral prefrontal cortex, the orbitofrontal cortex as well as various areas relevant for processing of sensory stimulus features in the temporal cortex (Fig. 1b). Electrodes from both left and right hemispheres were included in each of the ROIs. We employed the Desikan Killian atlas[116] for label extraction.

Electrodes located at the following locations were labelled as TMP electrodes: 'inferiortemporal', 'middletemporal', 'superiortemporal', 'transversetemporal', 'fusiform', 'temporalpole', 'bankssts', 'parahippocampal', 'entorhinal'. This resulted in a total number of 325 electrodes (46 participants) in TMP.

Electrodes located at the following FreeSurfer locations were labelled as lPFC electrodes: 'caudalmiddlefrontal', 'parsopercularis', 'parsorbitalis', 'superiorfrontal', 'parstriangularis', 'rostralmiddlefrontal', 'frontalpole'. This resulted in a total number of 74 electrodes (23 participants) in lPFC.

Since the ROIs lPFC and TMP cover relatively large brain areas, we provide information about electrode localizations in their specific subregions in Extended Data Fig. 3.

Electrodes located at the following FreeSurfer locations were labelled as OFC electrodes: 'lateralorbitofrontal', 'medialorbitofrontal'. This resulted in a total number of 35 electrodes (12 participants) in OFC.

In the AMY, we recorded from 82 electrodes in 32 participants. In the HPC, we recorded from 115 electrodes in 34 participants. A video depiction of AMY and HPC electrode locations is provided in Supplementary Video 1.

In all patients, we removed channels located in white matter, resulting in 631 clean channels across all patients (12.62 ± 6.66 channels per patient).

### Preprocessing

Preprocessing was performed on the entire raw data using EEGLAB[117], and included high-pass filtering at a frequency of 0.1 Hz and low-pass filtering at a frequency of 200 Hz. We also applied a band-stop (notch) filter with frequencies of 49–51 Hz, 99–101 Hz and 149–151 Hz. Artefact rejection was performed using a previously published method[118]. This automatic detection algorithm identifies epileptiform activity on the basis of three key criteria: (1) the amplitude of the time series, (2) the gradient (that is, the amplitude difference between adjacent time points) and (3) the amplitude of the data after applying a 250-Hz high-pass filter, which further enhances the detection of epileptogenic spikes. Each time point in the raw iEEG data was converted into a $z$-score on the basis of the participant-specific mean and standard deviation of these three measures across the whole experiment in each channel independently. A time point was marked as artefactual if it exceeded a $z$-score of 6 in any measure or met a conjunction threshold of an amplitude $z$-score of 4 combined with either a gradient or high-frequency $z$-score of 3.

We found that the combination of raw EEG amplitudes, EEG gradients and high-pass-filtered EEG amplitudes at 250 Hz provided high sensitivity for detecting epileptiform activity and other artefacts. This was further validated through manual identification of artefacts in randomly selected subsets of the data. To ensure that all epileptic activity was completely removed from our data, we not only excluded the detected time points but also removed the 1,000 ms preceding and following each detected artefact sample.

After identifying and marking the segments of the data contaminated with artefacts, we segmented the data into 7-s epochs (from −3 to 4 s) around the presentation of each CS cue. Any epoch containing artefact segments detected in the continuous (non-epoched) data was entirely removed from further analysis. In addition, we visually examined raw time-series plots and spectrograms to assess the presence of artefacts in the frequency domain. The number of epochs removed due to artefacts varied depending on the signal quality of each participant and was at 20.6 ± 19.2 epochs of cue presentation across participants and channels. The number of excluded trials in each condition of the experiment is presented in Extended Data Table 2.

### Time–frequency analysis

Using the FieldTrip toolbox[119], we decomposed the signal using complex Morlet wavelets with a variable number of cycles, that is, linearly increasing in 29 steps between 3 cycles (at 1 Hz) and 6 cycles (at 29 Hz) for the low-frequency range, and in 15 steps from 6 cycles (at 30 Hz) to 12 cycles (at 100 Hz) for the high-frequency range[61,62]. The resulting time series of frequency-specific power values were then $z$-scored by taking as a reference the mean activity across all trials in the experiment[120]. This type of normalization was applied to remove any common feature of the signal unrelated to the stability of the cue representations or the encoding of context-specific information.

We contrasted oscillatory power values in the frequency ranges of interest between CS+ and CS− trials during acquisition and extinction (Fig. 2a). Note that in the analysis during extinction, the data for the CS− condition during extinction include both CS+− and CS−− trials (that is, CS− trials that were either CS+ or CS− during acquisition). This was done to focus on the current valence of the CS during extinction—which

we assumed would be the primary factor influencing theta power—and to maximize statistical power by including all available trials in the CS− condition. The resulting time–frequency data (with the same decomposition parameters) were employed in our pattern similarity analyses (see below).

We only included participants with at least 8 trials in the group comparisons, resulting in a total of $N = 32$ participants in the analysis performed in the amygdala. The number of CS+ trials in this region was 38.72 ± 8.48 (mean ± s.d.) during acquisition and 19.46 ± 4.57 during extinction.

### Contrast-based RSA

We employed temporally resolved RSA to evaluate the dynamics of context and cue representations following previous work[61,121]. We assessed the stability of neural representations by computing the within-item similarity of each cue (that is, the average similarity for all repeated presentations of the same item, Fig. 3a). We compared this metric between CS+ and CS− trials, that is, tested whether the repeated visual presentation of a cue associated with a US (CS+ items) elicited more stable response patterns compared with the repeated presentation of non-predictive cues (CS− items; Fig. 3a). This was done separately during acquisition and extinction. The CS+ vs CS− comparisons were performed using paired $t$-tests after Fisher $z$-transforming the rho values obtained in each trial and condition in each participant independently. Note that the computation of item stability implies that representational patterns across repeated presentations of the same items cannot be averaged as is often done in RSA.

In addition, we investigated the presence of context-specific representations by comparing the similarity of same-contexts versus different-contexts in each of our ROIs. We assessed similarities for same-context and different-context item pairs and averaged them across all combinations of items in the same condition in each participant independently. The same-context and different-context correlations were then statistically compared at the group level using $t$-tests (rho values were Fisher $z$-transformed before contrasting them). This metric was also computed during acquisition and extinction. Similar to the item stability metric previously described, we did not average the representational patterns corresponding to the repeated presentation of the same context video but performed the similarity comparisons across all instances in which a particular video was presented. In some of our context-specific analysis (presented in Fig. 3), we locked the data to the presentation of the cue to facilitate the comparison with item stability results. Note that both cues and videos were presented simultaneously during the cue presentation (the cue was overlaid on top of the video), but the videos were additionally presented separately for 2 s before the cues.

Representational patterns were defined by specifying a 500-ms time window, in which averaged time courses of frequency-specific power values (all frequencies in the 1–100 Hz range) were included across all contacts in the respective ROI. A representational pattern was thus composed of activity of $N$ electrodes × 44 frequencies in each 500-ms window. Note that the number of channels included varied depending on the number of electrodes available for a particular participant/ROI. These two-dimensional arrays were concatenated into 1D vectors for similarity comparisons. In total, the number of electrodes included in each of our ROIs was as follows (mean ± s.d. across participants): TMP: 7.06 ± 4.23, AMY: 2.56 ± 1.39, HPC: 3.38 ± 1.82, OFC: 2.91 ± 0.9, lPFC: 3.21 ± 1.83.

Pairwise (Spearman) correlations among representational patterns were computed in sliding time windows (windows of 500 ms computed every 50 ms, that is, with a 90% overlap), resulting in a time series of neural representational similarity matrices for each of our ROIs. In the main analyses (Figs. 3 and 4), we assessed the correlation of these representational patterns across all matching time points, resulting in a time series of item stability or context specificity values for each condition. The time series were averaged in each condition for

each participant independently (after Fisher $z$-transforming them), and the resulting average time series were contrasted via paired $t$-tests across conditions at the group level. Multiple comparisons correction was performed using cluster-based permutation statistics (see below).

The number of trials for CS+ and CS− items was unbalanced during acquisition and extinction (two out of three items were labelled as CS+ during acquisition and only one as CS−, while the reverse was true during extinction). We thus computed within-item similarity separately for each of the two items with the same contingency in a given phase, and then averaged the results before contrasting the time series with the time series of the item corresponding to the opposite CS.

We only included participants with at least 8 trials in each condition, leading to the following number of participants for the item-stability analysis: AMY: 32, HPC: 34, OFC: 12, lPFC: 22 for both acquisition and extinction. In the TMP, the number of participants for the item-stability analysis was 43 during acquisition and 44 during extinction. The number of participants included in the context-specificity analysis was the same in both experimental phases in all regions: lPFC: 22, AMY: 32, HPC: 34, OFC: 12, TMP: 46.

In all pattern similarity plots, correlations corresponding to each 500-ms window were assigned to the time point at the centre of the respective window (for example, a time bin corresponding to activity from 0 to 500 ms was assigned to 250 ms).

## Coordination analyses

We computed a single-trial metric of item stability by assessing the similarity of the neural representation of an item presented in a trial with the representation of the same item in all other trials during a particular experimental phase. In addition, we computed a single-trial metric of context specificity by subtracting the similarity of each context video with all repeated presentations of the same video from the similarity with all different videos in a particular experimental phase. These metrics were computed in each of our ROIs and then correlated between different ROIs across trials for each participant. Correlations were calculated in sliding time windows (500-ms windows, overlapping by 95%) across all possible combinations of time points. This resulted in one correlation value for each participant in each time-bin × time-bin pair. We assessed potential temporal offsets because the latency and duration of representations may differ across brain regions, making it more likely that coordination is temporally shifted rather than simultaneous (see refs. 61,122). Notably, the relative timing of coordination offers insights into the temporal order of representations across brain regions during fear extinction. Statistical significance was assessed by contrasting these rho values against zero at the group level, and correction for multiple comparisons was performed using cluster-based permutation statistics (see below).

To confirm that the trial-level results were not driven by the main condition differences in the representational feature vectors representing CS+ and CS− items, we performed all trial-level correlation analyses after separately $z$-scoring the reinstatement values in CS+ and CS− trials. We obtained equivalent results in all reported effects. In additional analyses, we assessed connectivity between ROIs where we observed significant coordination effects using the weighted phase lag index. Results are presented in Supplementary Note 6.

The number of participants included in the trial-based RSA analyses depended on two criteria: (1) the number of trials in each condition, which was set to a minimum of 8 trials, consistent with the contrast-based RSA analyses (see above) (a participant was excluded from this analysis if the number of trials was below 8 in any of the conditions tested); and (2) the number of participants with electrodes in both ROIs, which was based solely on clinical needs. The number of participants included in the trial-based analyses comparing item stability between regions was as follows: TMP–AMY: 28, TMP–HPC: 31, TMP–lPFC: 18, TMP–OFC: 11, AMY–HPC: 24, AMY–lPFC: 14, AMY–OFC: 8. The number of participants included in the trial-based comparisons of

lPFC context specificity with context specificity in the other areas was as follows: lPFC–AMY: 14, lPFC–TMP: 18, lPFC–HPC: 15, lPFC–OFC: 11.

## Reinstatement analysis

To investigate whether lPFC context specificity during extinction predicted reinstatement in the TMP during the test phase, we conducted a participant-level correlation analysis. First, we calculated a metric of context specificity in the lPFC by averaging the difference between same-context and different-context correlations within the time period where significant context effects were observed in this region during extinction (from 0.8 to 1.15 s; Fig. 4c middle). These correlations were averaged both across trials and over time within this time window, resulting in a single value of lPFC context specificity for each participant. Second, we calculated reinstatement across experimental phases, comparing activity patterns during the test with those observed either during acquisition or during extinction (that is, acquisition–test reinstatement and extinction–test reinstatement). We specifically focused on the TMP, where the main condition differences were observed (that is, higher item stability for CS+ compared with CS− trials). We correlated each trial during acquisition and extinction with each trial during test in each participant independently. We then averaged these correlations across trials and across time (focusing on the same time window of significant lPFC context specificity) and subtracted acquisition–test reinstatement and extinction–test reinstatement, resulting in a single value of TMP reinstatement for each participant. Finally, we correlated context specificity and reinstatement across participants using Spearman correlations (Fig. 6b).

To further investigate whether the observed correlation between the lPFC and TMP was unique to the period of context specificity in the lPFC, we conducted a temporally resolved analysis. Employing sliding time windows of 500 ms with a 95% overlap, we again correlated same and different items during both acquisition and extinction, and quantified the time course of item-specific reinstatement during the test phase. In the lPFC, we selected context-specific representations in the same cluster used in the primary analysis (described above) and correlated these values across participants at every time point of the TMP reinstatement time course using Spearman correlations. This approach yielded a time course of rho and $p$ values, which we subsequently corrected for multiple comparisons using cluster-based permutation statistics (see below).

## Functional relevance of extinction reinstatement

To assess the functional relevance of reinstatement during the test phase, we computed a trial-level metric of reinstatement values. Specifically, we measured the similarity of each trial in the test phase to the same and different items from the extinction phase and subtracted these two values. This was conducted at matching time points within each trial, generating a time course of item reinstatement for every trial. We focused on the TMP, and specifically on the time period when significant differences in item stability between CS+ and CS− trials were detected. We averaged reinstatement values in this window, resulting in a trial-level metric of item-specific reinstatement for each participant. These reinstatement values were then correlated with the participants' subjective ratings using Spearman correlations (after Fisher $z$-transforming them). The resulting values were tested against zero at the group level to determine statistical significance (Fig. 7).

In our experiment, the average number of trials in which participants failed to provide a response was considerable ($12.65 \pm 15.47$ per participant), primarily because responding was not a prerequisite for progressing to the next trial in any experimental phase. Consequently, we applied a more liberal threshold for the minimum number of trials included in this analysis, setting it at 4 instead of 8 trials. We repeated the analysis with the stricter threshold of 8 trials and found qualitatively similar results (CS++: $t_{(14)} = 2.17$; $p = 0.048$; CS+−: $t_{(14)} = 2.29$; $p = 0.037$; CS−−: $t_{(13)} = -0.67$; $p = 0.51$).

To account for potential group-level biases introduced by outlier effects, we repeated our analysis after excluding trials with TMP reinstatement values exceeding 2 standard deviations from the mean for each participant. On average, $7.2 \pm 4.9$ trials were excluded per participant, resulting in a final sample of 17 participants. The results remained largely consistent (CS++: $t_{(17)} = 0.849$, $p = 0.408$; CS+−: $t_{(16)} = 2.51$, $p = 0.0231$; CS−−: $t_{(18)} = -0.0126$, $p = 0.99$).

## Multiple comparisons correction
We performed cluster-based permutation statistics to correct for multiple comparisons in the oscillatory power analyses (Fig. 2), the contrast-based pattern similarity analyses (Figs. 3b,d and 4b,c) and the trial-based pattern similarity analysis (Figs. 3f and 4d).

In the oscillatory power analyses, we contrasted CS+ and CS− time–frequency maps using paired $t$-tests after shuffling the trial labels 1,000 times for each participant independently. We considered a time point significant if the difference between these surrogate conditions was significant at $p < 0.05$ (two-tailed tests were employed). For each permutation, we computed clusters of significant values defined as contiguous regions in time–frequency space where significant correlations were observed. We took the largest cluster at each permutation to build the null distribution. We only considered significant those contiguous time pairs in the empirical (non-shuffled) data whose summed $t$ values exceeded the summed $t$-value of 95% of the surrogate clusters (corresponding to a corrected $p < 0.05$; see ref. 123).

Since the contrast-based pattern similarity analyses were based on matching time points, we contrasted time series of different conditions (CS+ vs CS− items or same vs different contexts) at different time points using $t$-tests after shuffling the trial labels of each condition 1,000 times for each participant independently. Adjacent regions of significant differences were formed along one dimension only (time), resulting in a distribution of surrogate $t$ values under the assumption of the null hypothesis. Similar to the procedure employed in the oscillatory power analysis, we only considered significant those contiguous time pairs in the empirical data whose summed $t$ values were above the summed $t$-value of 95% of the distribution of surrogate clusters[123].

In the trial-based pattern similarity analysis, a temporal generalization approach was employed, and therefore clusters of significant regions were formed in 2 dimensions (time × time). Item stability or context specificity was correlated across trials between brain regions at the group level using paired $t$-tests as in the original data. $t$ values of adjacent regions where significant differences were observed were summed. For each permutation, we shuffled the trial labels in one of the regions and performed the analysis again, storing the cluster with the highest absolute $t$-value. This resulted in a distribution of surrogate $t$ values under the null hypothesis of no condition differences. Similar to the procedure we employed in the contrast-based analyses, we then computed $p$ values by ranking the observed $t$ values of the biggest cluster in the empirical data with this surrogate distribution. Time periods with contiguous significant regions exceeding 95% of surrogate clusters were considered significant[123].

In addition to cluster-based permutations, we also corrected our results for multiple comparisons using the Bonferroni method. Given that we included 5 ROIs, the alpha level for significance was set to $p = 0.05/5 = 0.01$.

## Reporting summary
Further information on research design is available in the Nature Portfolio Reporting Summary linked to this article.

## Data availability
Anonymized intracranial EEG data supporting the findings of this study are available on the Open Science Framework at https://osf.io/xfprt/ (ref. 124). Source data are provided with this paper.

## Code availability
Custom-written Matlab code supporting the findings of this study is available on GitHub at https://github.com/dpachec/extinction and via Zenodo at https://doi.org/10.5281/zenodo.15926577 (ref. 125).

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

## Acknowledgements

This work was supported by the Deutsche Forschungsgemeinschaft (DFG, German Research Foundation) through grant SFB 1280 (projects A01 and A02) project number 316803389. D.P.-E. received funding from the Spanish Ministry of Science and Innovation (RYC2023-045254-I). The funders had no role in study design, data collection and analysis, decision to publish or preparation of the manuscript.

## Author contributions

M.-C.F., D.P.-E. and N.A. conceptualized the project. D.P.-E., A.B., G.J. and N.A. designed the methodology. M.-C.F., K.L., V.L., V.F. and J.Y. collected data. D.P.-E. and N.A. analysed data. D.P.-E. wrote the original manuscript draft. D.P.-E. and N.A. reviewed and edited the manuscript. N.A. and O.G. acquired funding. V.N., O.G., L.S., B.H. and Q.C. acquired resources. N.A. supervised the project.

## Funding

## Competing interests

The authors declare no competing interests.

## Additional information

**Extended data** is available for this paper at https://doi.org/10.1038/s41562-025-02268-5.

**Correspondence and requests for materials** should be addressed to Daniel Pacheco-Estefan, Qi Chen or Nikolai Axmacher.

Daniel Pacheco-Estefan ⓘ [1,2] ✉, Antoine Bouyeure[2], George Jacob ⓘ [2], Marie-Christin Fellner[2], Katia Lehongre ⓘ [3], Virginie Lambrecq[3,4,5], Valerio Frazzini[3,4,5], Vincent Navarro[3,4,5,6], Onur Güntürkün ⓘ [7], Lu Shen[8,9,10], Jing Yang[8,9,10], Biao Han[8,9,10], Qi Chen[8,9,10] ✉ & Nikolai Axmacher ⓘ [2] ✉

[1]Department of Basic, Developmental and Educational Psychology, Faculty of Psychology, Autonomous University of Barcelona, Barcelona, Spain. [2]Department of Neuropsychology, Institute of Cognitive Neuroscience, Faculty of Psychology, Ruhr University Bochum, Bochum, Germany. [3]Sorbonne Université, Paris Brain Institute – Institut du Cerveau, ICM, INSERM, CNRS, APHP, Pitié-Salpêtrière Hospital, Paris, France. [4]AP-HP, Département de Neurophysiologie, Hôpital Pitié-Salpêtrière, DMU Neurosciences, Paris, France. [5]AP-HP, Epilepsy Unit, Pitié-Salpêtrière Hospital, DMU Neurosciences, Paris, France. [6]AP-HP, Center of Reference for Rare Epilepsies, Pitié-Salpêtrière Hospital, Paris, France. [7]Department of Biopsychology, Institute of Cognitive Neuroscience, Faculty of Psychology, Ruhr University Bochum, Bochum, Germany. [8]Center for Studies of Psychological Application, and Guangdong Key Laboratory of Mental Health and Cognitive Science, South China Normal University, Guangzhou, China. [9]School of Psychology, South China Normal University, Guangzhou, China. [10]Key Laboratory of Brain, Cognition and Education Sciences, Ministry of Education, Beijing, China. ✉e-mail: daniel.pacheco@uab.cat; qi.chen27@gmail.com; nikolai.axmacher@rub.de

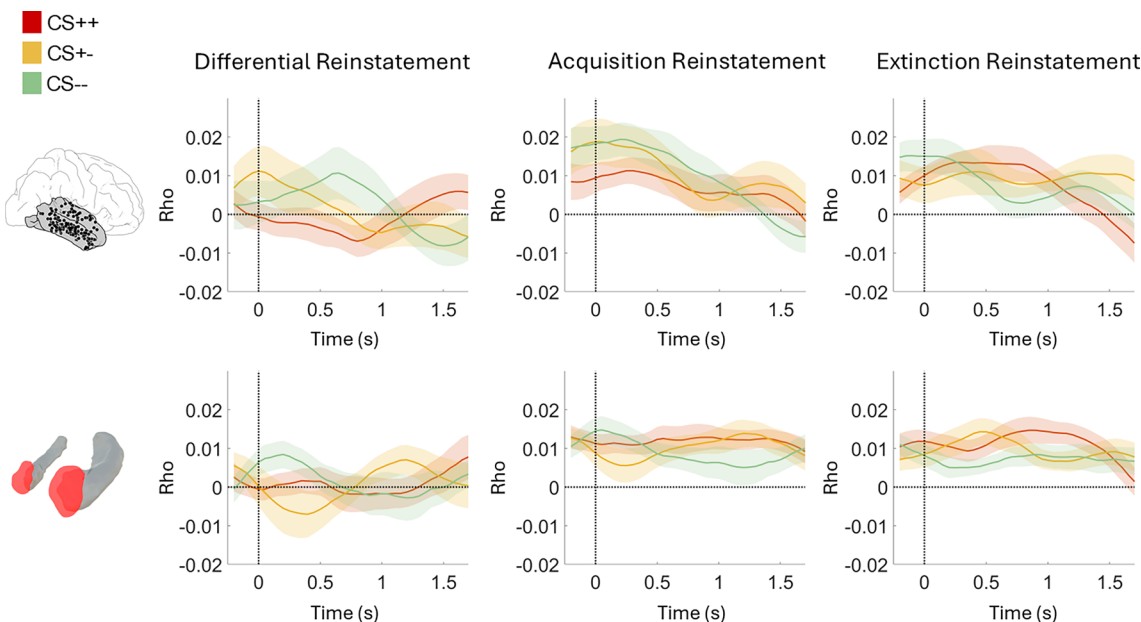

**Extended Data Fig. 1 | Reinstatement of acquisition and extinction memory traces do not differ across trial types.** Acquisition-to-test (left), extinction-to-test and differential reinstatement (acquisition minus extinction-to-test) did not differ between our three trial types (CS++, CS+−, CS−−) in the TMP (top row; N = 26) or in the AMY (bottom row (N = 31).

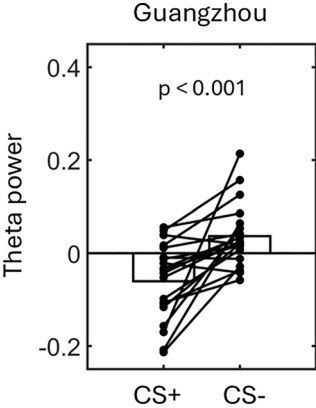
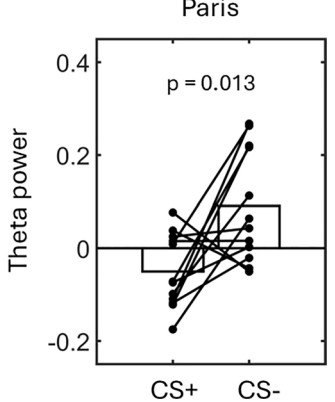

**Extended Data Fig. 2 | Theta power in the AMY during extinction was significantly higher for CS− as compared to CS+ trials in both the Guangzhou and the Paris patient cohorts.** Figure shows mean AMY power within the time-frequency cluster identified in the main analysis during extinction (Fig. 2a, bottom left) for the Guangzhou (left; N = 20) and the Paris (right; N = 12) patient groups. In both cohorts, AMY theta power was significantly higher for CS− as compared to CS+ trials, consistent with the findings observed when combining the data of the two groups. These results demonstrate that our AMY theta power findings are robust and not influenced by methodological differences between the epilepsy centers where the data was collected (for example, variations in electrode configurations or recording devices in Paris and Guangzhou).

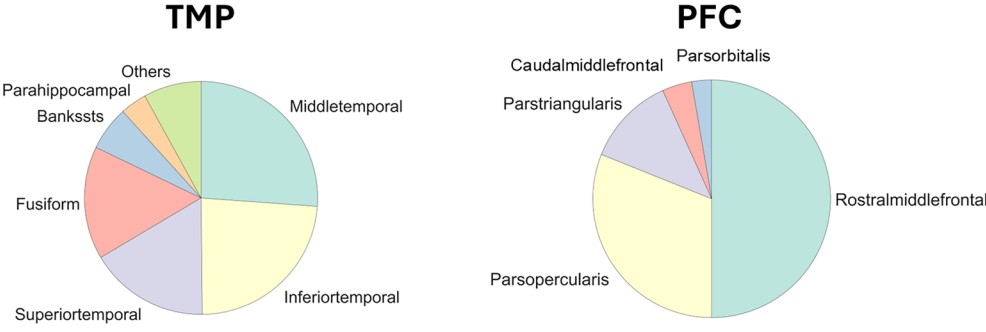

| Brain Region | Electrodes | Subjects |
|---|---|---|
| Bankssts | 20 | 11 |
| Entorhinal | 8 | 6 |
| Fusiform | 51 | 25 |
| Inferiortemporal | 77 | 33 |
| Middletemporal | 85 | 36 |
| Parahippocampal | 12 | 8 |
| Superiortemporal | 54 | 22 |
| Temporalpole | 10 | 5 |
| Transversetemporal | 8 | 5 |

| Brain Region | Electrodes | Subjects |
|---|---|---|
| Caudalmiddlefrontal | 3 | 3 |
| Parsopercularis | 23 | 11 |
| Parsorbitalis | 2 | 1 |
| Parstriangularis | 9 | 5 |
| Rostralmiddlefrontal | 37 | 19 |

**Extended Data Fig. 3 | Proportion of electrodes in TMP and PFC subregions.** Proportion of electrodes included in specific subregions of the TMP (left) and the PFC (right; Desikan Killian atlas; Desikan et al., 2006) across all subjects. A summary table for each ROI, including the number of electrodes and the number of subjects in each subregion, is presented below the pie plots.

**Extended Data Table 1 | Patient demographic and clinical information**

| Participant | Gender | Age | Global IQ | Seizure Onset Zone | MQ |
|---|---|---|---|---|---|
| DBX | M | 22 | 115 | Right hemisphere | 106 |
| LSY | F | 16 | 88 | Right hemisphere | 39 |
| JJL | F | 25 | 86 | Left posterior head (temporal and occipital lobes) | 73 |
| HWL | M | 19 | 104 | Left frontal lobe | 80 |
| ZHX | M | 31 | 87 | Left temporal lobe | <51 |
| WXY | M | 24 | 108 | Left temporal lobe | 75 |
| LQM | F | 23 | 96 | Right temporal insular cortex | 81 |
| CLF | F | 21 | 94 | Bilateral limbic system | 52 |
| HLL | F | 26 | 95 | Right hemisphere (TPO area) | 70 |
| DHZ | F | 16 | 83 | Right limbic system | 77 |
| CYY | M | 35 | 83 | Right temporal lobe | 51 |
| HLY | M | 26 | 107 | Multifocal | 95 |
| YJH | M | 22 | 92 | Right temporal lobe | 63 |
| GKS | M | 30 | 88 | Limbic system | <51 |
| ZH | M | 21 | 98 | Left orbital gyrus, anterior temporal lobe, parietal lobe | 81 |
| KYP | M | 27 | 98 | Right limbic system | 73 |
| LYT | M | 18 | 91 | Right temporal lobe | 89 |
| LC | M | 36 | 93 | Left posterior temporal lobe | 84 |
| ZJJ | M | 21 | 93 | Right temporal lobe | 108 |
| ZXF | M | 32 | 124 | right fronto-temporal areas | 120 |
| ZJH | M | 27 | 84 | Right posterior temporal areas | 52 |
| LHD | M | 33 | - | Left limbic system | - |
| C | M | 23 | 82 | Right medial temporal lobe | 66 |
| YLL | F | 33 | 75 | Right limbic system | 68 |
| WTT | F | 24 | - | Right temporal insular cortex | - |
| MAY | M | 40 | - | Right insular cortex | - |
| LHM | M | 25 | 97 | Left temporal lobe | 54 |
| LSW | M | 21 | 82 | Left frontal lobe | 55 |
| CCL | M | 22 | - | Left temporal insular cortex | - |
| LDN | F | 21 | 109 | Left anterior frontal areas | 103 |
| pat_02651_1127 | F | 27 | - | Left temporal lobe | - |
| pat_02660_1136 | F | 34 | - | Bilateral tempo-mesial | - |
| pat_02680_1158 | F | 52 | - | Left temporal lobe | - |
| pat_02689_1168 | F | 31 | 75 | Right temporo-parietal | - |
| pat_02711_1193 | F | 18 | 118 | Right temporal lobe | - |
| pat_02718_1201 | F | 26 | - | Left temporal lobe | - |
| pat_02757_1244 | M | 28 | - | Left frontal lobe | - |
| pat_02985_1405 | M | 31 | - | undetermined focus | - |
| pat_03012_1444 | M | 26 | - | Temporo-polar | - |
| pat_03046_1482 | F | 39 | 93 | Right temporal | - |
| pat_03083_1527 | M | 25 | 101 | Bilateral temporo-basal | - |
| pat_03092_1538 | F | 35 | 81 | Bilateral temporo-mesial | - |
| pat_03105_1551 | F | 32 | - | Bilateral temporo-mesial | - |
| pat_03128_1591 | F | 28 | 99 | Bilateral temporo-mesial | - |
| pat_03138_1601 | F | 43 | - | Left temporo-mesial | - |
| pat_03146_1608 | M | 37 | 88 | Bilateral temporo-mesial | - |
| pat_03174_1634 | M | 25 | - | Polar and baso medial region of the right temopral lobe | - |
| pat_03249_1711 | F | 52 | - | Left temporo-mesial | - |
| pat_03266_1729 | M | 39 | - | Bilateral temporo-mesial | - |
| pat_03300_1766 | F | 25 | - | Right temporo-mesial | - |

Details about patients' gender, age, Intelligence Quotient (IQ), Seizure Onset Zone (SOZ), and Memory Quotient (MQ) are provided.

**Extended Data Table 2 | Number of excluded trials after artifact rejection in every ROI across all task stages**

| ROI | CS++ | CS+- | CS-- |
|---|---|---|---|
| AMY | 38.41 ± 8.01 | 37.18 ± 8.84 | 37.78 ± 9.31 |
| HPC | 37.44 ± 8.32 | 36.53 ± 8.86 | 36.47 ± 8.72 |
| OFC | 39.17 ± 9.55 | 40.33 ± 9.57 | 39.83 ± 9.75 |
| IPFC | 40.78 ± 8.72 | 40.74 ± 9.67 | 41.26 ± 9.43 |
| TMP | 26.54 ±12.72 | 27.89 ± 12.41 | 26.85 ± 12.43 |

Please note that since some of the RSA analyses rely on the comparison of many items of the same type (for example, correlations are computed between all pairs of CS+ items during a specific task stage), the statistical power of these analyses is much higher. The number of trials in the table applies to the power analysis (Fig. 2) and to the trial-level RSA analysis (Figs. 3f and 4d).

                           Nikolai Axmacher
                           Qi Chen

# Reporting Summary

## Statistics

For all statistical analyses, confirm that the following items are present in the figure legend, table legend, main text, or Methods section.

| n/a | Confirmed | |
|---|---|---|
| ☐ | ☒ | The exact sample size (*n*) for each experimental group/condition, given as a discrete number and unit of measurement |
| ☐ | ☒ | A statement on whether measurements were taken from distinct samples or whether the same sample was measured repeatedly |
| ☐ | ☒ | The statistical test(s) used AND whether they are one- or two-sided<br>*Only common tests should be described solely by name; describe more complex techniques in the Methods section.* |
| ☐ | ☒ | A description of all covariates tested |
| ☐ | ☒ | A description of any assumptions or corrections, such as tests of normality and adjustment for multiple comparisons |
| ☐ | ☒ | A full description of the statistical parameters including central tendency (e.g. means) or other basic estimates (e.g. regression coefficient) AND variation (e.g. standard deviation) or associated estimates of uncertainty (e.g. confidence intervals) |
| ☐ | ☒ | For null hypothesis testing, the test statistic (e.g. *F*, *t*, *r*) with confidence intervals, effect sizes, degrees of freedom and *P* value noted<br>*Give P values as exact values whenever suitable.* |
| ☒ | ☐ | For Bayesian analysis, information on the choice of priors and Markov chain Monte Carlo settings |
| ☒ | ☐ | For hierarchical and complex designs, identification of the appropriate level for tests and full reporting of outcomes |
| ☒ | ☐ | Estimates of effect sizes (e.g. Cohen's *d*, Pearson's *r*), indicating how they were calculated |

*Our web collection on statistics for biologists contains articles on many of the points above.*

## Software and code

Policy information about availability of computer code

| Data collection | The experiment was programmed in Presentation (Neurobehavioral Systems, California, USA), and was deployed on Laptop computers running Microsoft Windows. |
|---|---|
| Data analysis | Behavioral and intracranial EEG data was analyzed using custom-written Matlab (R2024B) code, deposited in GitHub (https://github.com/dpachec/extinction). Several Matlab toolboxes were employed, including EEGLAB (v2021.0; https://sccn.ucsd.edu/eeglab/) and Fieldtrip (08-11-2024; https://www.fieldtriptoolbox.org/). Software employed for electrode localization included FreeSurfer (v7.4.1; https://surfer.nmr.mgh.harvard.edu/), 3D Slicer (v5.6.2; https://www.slicer.org/), SPM (v12; https://www.fil.ion.ucl.ac.uk/spm/) and BrainVisa (v5.1; https://brainvisa.info/web/). |

For manuscripts utilizing custom algorithms or software that are central to the research but not yet described in published literature, software must be made available to editors and reviewers. We strongly encourage code deposition in a community repository (e.g. GitHub). See the Nature Portfolio guidelines for submitting code & software for further information.

## Data

Policy information about availability of data

All manuscripts must include a data availability statement. This statement should provide the following information, where applicable:
- Accession codes, unique identifiers, or web links for publicly available datasets
- A description of any restrictions on data availability
- For clinical datasets or third party data, please ensure that the statement adheres to our policy

Anonymized intracranial EEG data supporting the findings of this study are available on the Open Science Framework at https://osf.io/xfprt/

## Research involving human participants, their data, or biological material

Policy information about studies with human participants or human data. See also policy information about sex, gender (identity/presentation), and sexual orientation and race, ethnicity and racism.

| | |
|---|---|
| Reporting on sex and gender | We collected data from both sexes in this study (self-reported), and did not evaluate differences related to sex for performing the task. |
| Reporting on race, ethnicity, or other socially relevant groupings | No distinctions were made regarding race or ethnicity. The experiment was conducted with different patient populations in France and China, and data from the two patient populations was aggregated in all analyses. |
| Population characteristics | Forty-nine patients (22 females, 28.2 ± 8.18 years) with medically intractable epilepsy participated in the study. |
| Recruitment | Data were collected at the Pitié Salpêtrière hospital, Paris, and the South China Normal University Hospital, Guangzhou, China. All patients were implanted for clinical purposes and participation in the study was voluntary. The study was conducted with all available patients, with no selection criteria. |
| Ethics oversight | The study was conducted according to the latest version of the Declaration of Helsinki and approved by the institutional review board at the Pitié Salpêtrière hospital (C11-16, C19-55 National Institute of Health and Medical Research sponsor), and by the ethics committee at the South China Normal University Hospital, Guangzhou. All patients provided written informed consent. |

Note that full information on the approval of the study protocol must also be provided in the manuscript.

# Field-specific reporting

Please select the one below that is the best fit for your research. If you are not sure, read the appropriate sections before making your selection.

☒ Life sciences ☐ Behavioural & social sciences ☐ Ecological, evolutionary & environmental sciences

For a reference copy of the document with all sections, see nature.com/documents/nr-reporting-summary-flat.pdf

# Life sciences study design

All studies must disclose on these points even when the disclosure is negative.

| | |
|---|---|
| Sample size | Since no prior iEEG research has specifically examined fear extinction, we based our sample size on previous studies investigating related processes, such as fear acquisition (e.g., Chen et al., Sci. Adv., 2021) and emotional memory formation (e.g., Costa et al., Nat. Commun., 2022; Zhang et al., Nat. Commun., 2024). These studies typically involved small samples (Chen et al., N = 13; Zhang et al., N = 7; Costa et al., N = 19). Our cohort of N = 49 patients substantially exceeds the sample sizes employed in these and other comparable studies. |
| Data exclusions | From a total of 50 subjects, one subject had to be excluded because he did not provided enough answers during the acquisition and extinction periods (12.5% and 7.8% respectively), leading to a total number of N=49 subjects. |
| Replication | Our study was not replicated, primarily due to the challenges of accessing iEEG patient populations. However, our relatively large sample size compared to most iEEG studies (N=49) enhances the statistical robustness of our findings. |
| Randomization | All participants took part in all experimental conditions in our study following a within subjects design. |
| Blinding | Investigators were blind to experimental conditions during data collection and during the preprocessing of the data (artifact rejection) as explained in the manuscript (Methods section). |

# Reporting for specific materials, systems and methods

We require information from authors about some types of materials, experimental systems and methods used in many studies. Here, indicate whether each material, system or method listed is relevant to your study. If you are not sure if a list item applies to your research, read the appropriate section before selecting a response.

## Materials & experimental systems

| n/a | Involved in the study |
|-----|----------------------|
| ☒ ☐ | Antibodies |
| ☒ ☐ | Eukaryotic cell lines |
| ☒ ☐ | Palaeontology and archaeology |
| ☒ ☐ | Animals and other organisms |
| ☒ ☐ | Clinical data |
| ☒ ☐ | Dual use research of concern |
| ☒ ☐ | Plants |

## Methods

| n/a | Involved in the study |
|-----|----------------------|
| ☒ ☐ | ChIP-seq |
| ☒ ☐ | Flow cytometry |
| ☒ ☐ | MRI-based neuroimaging |

## Plants

| | |
|---|---|
| Seed stocks | *Report on the source of all seed stocks or other plant material used. If applicable, state the seed stock centre and catalogue number. If plant specimens were collected from the field, describe the collection location, date and sampling procedures.* |
| Novel plant genotypes | *Describe the methods by which all novel plant genotypes were produced. This includes those generated by transgenic approaches, gene editing, chemical/radiation-based mutagenesis and hybridization. For transgenic lines, describe the transformation method, the number of independent lines analyzed and the generation upon which experiments were performed. For gene-edited lines, describe the editor used, the endogenous sequence targeted for editing, the targeting guide RNA sequence (if applicable) and how the editor was applied.* |
| Authentication | *Describe any authentication procedures for each seed stock used or novel genotype generated. Describe any experiments used to assess the effect of a mutation and, where applicable, how potential secondary effects (e.g. second site T-DNA insertions, mosiacism, off-target gene editing) were examined.* |

