## [Peer Review File · Nature Human Behaviour]

Representational dynamics during extinction of fear memories in the human brain

Corresponding Author: Dr Daniel Pacheco Estefan

Version 0:

Decision Letter:

17th December 2024

Dear Dr Pacheco Estefan,

Thank you once again for your manuscript, entitled "Representational dynamics during extinction of fear memories in the human brain", and for your patience during the peer review process.

Your Article has now been evaluated by 3 referees. You will see from their comments copied below that, although they find your work of potential interest, they have raised quite substantial concerns. In light of these comments, we cannot accept the manuscript for publication, but would be interested in considering a revised version if you are willing and able to fully address reviewer and editorial concerns.

We hope you will find the referees' comments useful as you decide how to proceed. If you wish to submit a substantially revised manuscript, please bear in mind that we will be reluctant to approach the referees again in the absence of major revisions. We are committed to providing a fair and constructive peer-review process. Do not hesitate to contact us if there are specific requests from the reviewers that you believe are technically impossible or unlikely to yield a meaningful outcome.

If you wish to submit a suitably revised manuscript, we would hope to receive it within 4 months. I would be grateful if you could contact us as soon as possible if you foresee difficulties with meeting this target resubmission date.

- Include a "Response to the editors and reviewers" document detailing, point-by-point, how you addressed each editor and referee comment. If no action was taken to address a point, you must provide a compelling argument. When formatting this document, please respond to each reviewer comment individually, including the full text of the reviewer comment verbatim followed by your response to the individual point. This response will be used by the editors to evaluate your revision and sent back to the reviewers along with the revised manuscript.
- Highlight all changes made to your manuscript or provide us with a version that tracks changes.

Link Redacted

Thank you for the opportunity to review your work. Please do not hesitate to contact me if you have any questions or would like to discuss the required revisions further.

Sincerely,

Nature Human Behaviour

Reviewer expertise:

Reviewer #1: Fear extinction, episodic memory, iEEG

Reviewer #2: Fear extinction, episodic memory, iEEG

Reviewer #3: Fear extinction, episodic memory, iEEG

REVIEWER COMMENTS:

Reviewer #1 (Remarks to the Author):

The reviewed study, Representational dynamics during extinction of fear memories in the human brain, aimed to examine the neurophysiological and representational characteristics of contextual fear conditioning and subsequent extinction learning. For this purpose, the authors conducted an elaborate multilab, multinational intracranial EEG (iEEG) study. The time-frequency characteristics of the iEEG signal were analyzed using classical Morlet wavelet analysis, extended with representational similarity analysis (RSA). The employed methods appear robust and suitable for addressing the stated research questions. However, I should note that I lack firsthand experience with RSA.

Overall, the study addresses a highly relevant topic. A better understanding of the neurophysiological underpinnings of threat processing could prove highly valuable for a better understanding of conditions such as anxiety disorders. By leveraging the high spatial resolution of iEEG, the current study records EEG activity from subcortical structures like the amygdala and hippocampus, which are key nodes in the neural fear and extinction circuitry. This approach also helps to bridge the translational gap between human and non-human studies. In sum, the study is well-written and engaging, features high-quality figures to illustrate the observed data, and should be of great interest to the neuroscientific and clinical community. However, before I can recommend it for publication, I would like the authors to address the following points:

1. The authors mention in lines 78-80, that rodent studies provide evidence that theta oscillations in the amygdala are associated with fear extinction. However, most of the cited rodent studies demonstrated that amygdala theta oscillations are increased during fear expression or during the presentation of the threat predictive CS+ (e.g., Pare & Colins, 2000, Seidenbecher et al., 2003, Likhik et al., 2004). Moreover, there is growing evidence from both animal and human research that theta oscillations within the neural fear circuitry (i.e., amygdala, hippocampus, and parts of the mPFC) play a significant role in fear expression. For instance, previous animal studies have especially demonstrated increased theta oscillations in the prelimbic cortex/dACC and/or amygdala during states of fear expression (e.g., Seidenbecher et al., 2003; Courtin et al., 2014; Fenton et al., 2014; Likhik et al., 2014; Lesting et al., 2011 and 2013; Taub et al., 2018; Rahman et al., 2018, Karalis et al., 2016). Additionally, human EEG studies have also reported heightened frontocentral theta oscillations, particularly during fear conditioning (e.g., Pirazzini et al., 2023, Chen et al., 2021) and fear recall (e.g., Mueller et al., 2014; Sperl et al., 2019; Bierwirth et al., 2021; but see Bierwirth et al., 2023), with some evidence suggesting a correlation between these frontal theta oscillations and the amygdala BOLD response (Sperl et al., 2018). Given this body of evidence, it is surprising that the current study did not observe increased theta oscillations in the PFC and amygdala. It would be valuable if the authors included this evidence in the introduction and acknowledged and discussed this absence of effect more thoroughly in the discussion.
2. The authors relied on “threat” and “safety” ratings to validate the fear conditioning paradigm. Typically, most fear conditioning studies utilize physiological measures such as skin conductance responses, fear-potentiated startle, or evoked heart rate responses to assess fear responses. While this might not have been feasible in the current study due to its cross-cultural, multilab nature, solely relying on rating data—highly susceptible to demand effects—is a limitation that should be discussed. This is particularly important given the use of a complex contextual fear conditioning paradigm, which is not widely employed in the fear conditioning literature.
3. In line 533, the authors mention that they used an aversive scream as the unconditioned stimulus (US). However, crucial details about the US are missing. Could the authors provide more information about the US? Specifically, at what decibel level was the US presented, and how was its level of aversiveness determined?
4. How many iEEG artifacts occurred for each stimulus? Were the number of artifacts comparable across conditions?
5. Was the study preregistered?

Literature:

Bierwirth et al. (2021) <https://doi.org/10.1016/j.bpsc.2021.02.011>
Bierwirth et al. (2023) <https://psycnet.apa.org/doi/10.1111/psyp.14283>
Karalis et al. (2016) 10.1038/nn.4251
Lesting et al. (2011) <https://doi.org/10.1371/journal.pone.0021714>
Lesting et al. (2013) <https://doi.org/10.1371/journal.pone.0077707>
Mueller et al. (2014) <https://doi.org/10.1523/JNEUROSCI.3427-13.2014>
Rahman et al. (2018) <https://doi.org/10.7554/eLife.35450>
Sperl et al. (2019) <https://doi.org/10.1093/cercor/bhx353>
Taub et al. (2018) <https://doi.org/10.1016/j.neuron.2017.11.042>

Reviewer #2 (Remarks to the Author):

In rodents, theta (4-8Hz) oscillations in amygdala and hippocampus play critical roles during both fear learning and extinction. Also extinction relies on the formation of novel, highly context-dependent memory traces that suppress the initial fear memories.

However, it remains unknown whether similar processes occur in humans and how they relate to previously described neural mechanisms of episodic memory formation and retrieval.

The authors analyzed Intracranial EEG (iEEG) recordings in epilepsy patients in the deep brain structures of the fear and extinction network, and they employed representational similarity analysis (RSA), which allows characterizing the memory traces of specific cues and contexts. In this study, they found that amygdala theta oscillations during extinction learning signal safety rather than threat and that extinction memory traces are characterized by stable and context-specific neural representations that are coordinated across the extinction network. They further demonstrate that context specificity during extinction learning predicts the reoccurrence of fear memory traces during a subsequent test period, while reoccurrence of extinction memory traces predicts safety responses.

The authors meticulously analyzed iEEG data recorded using the ABC paradigm with successfully drawing interesting results in the role of theta oscillations in amygdala and hippocampus. .

It is a well conducted and reasonably construed study.

Also their results look very persuasive.

However, I have several major concerns to accept their results.

Criterion 1 – Data & Methodology

Comments:

1. This study analyzed iEEG in 'medically intractable epilepsy participants'. Since antiepileptic drugs could potentially impair cognitive function, it would be helpful to clarify whether the participants had sufficient cognitive abilities to perform the experimental paradigm. Providing information on the participants' IQ, MQ, and seizure onset zone for each subject would allow for a more thorough assessment.
2. This study notes that the co-registration methods used in Paris and Guangzhou were different. It would be useful to explain whether this discrepancy is due to differences in institutional protocols or technical limitations. It would strengthen the study to demonstrate that the co-registration method does not affect the results. For example, showing that co-registration of Paris data using the Guangzhou method (or vice versa) does not lead to changes in labeling would help underscore the consistency of the results.
3. The experimental paradigm in this study references only Dunsmoor et al., 2014, which may not sufficiently support the design. Moreover, there appear to be notable differences between the two paradigms, such as the time intervals between the acquisition, extinction, and test phases. (For instance, Dunsmoor's paradigm was conducted over two days.) Could you clarify whether this experimental paradigm is identical to or a slight modification of those used in prior studies? Additionally, has this paradigm been validated in previously published research? If not, could you provide further justification for the validity of the design?
4. In Figures 2 and 3, does the CS- during the extinction phase include stimuli that were designated as either CS+ or CS- during the acquisition phase? In other words, does it encompass both CS+- and CS-- stimuli? I believe that CS+- reflects the extinction of fear learning, whereas CS-- serves as a safety signal that was not associated with fear learning. Could you clarify whether the results in the extinction phase distinguish between CS+- and CS--? If not, I would suggest separating these two categories in the results for clarity.
5. Regarding the CS++, I would expect the cumulative fear learning associated with this stimulus to result in very low amygdala (AMY) theta power, as it contrasts with the safety signal. However, in Figure 2, the CS+ (which presumably corresponds to the CS++ from the acquisition phase) appears to exhibit relatively high theta power. Could you clarify this apparent inconsistency?

Criterion 2 - Preregistration

Comments:

I have no comments.

Criterion 3 - Appropriate use of statistics and treatment of uncertainties

Comments:

The statistical methodology appears to have been appropriately applied; however, there seems to be a lack of discussion regarding the statistical uncertainty associated with the number of trials. How was the minimum number of trials, set at 8, determined? Additionally, a more detailed explanation of how varying the number of trials (e.g., fewer or more than 8) might influence the results would be valuable.

Furthermore, in Figure 1D, were there significant differences in behavioral performance among the three conditions during the test phase? The observed difference between CS++ and CS-- in the test phase seems similar to that observed during the extinction phase. Was the sample size (n) used for statistical analysis sufficient for each condition? Lastly, was normality assessed before conducting the paired t-tests?

Criterion 4 - Custom code

Comments:

I have no comments.

Criterion 5 - Conclusions

Comments:

I have no comments.

Criterion 6 - Suggested improvements

Comments:

1. The process of distinguishing and responding to threats and safety cues occurs rapidly in humans, as quick threat adaptation or escape is crucial for survival. In this context, it may be valuable to consider that, in addition to slower-cycle theta oscillations, faster-cycle neural activity could also play a role in safety responses. While the current manuscript focuses on theta oscillations, addressing this point in the discussion could enhance the interpretation of the results. Specifically, a brief mention of how faster neural oscillations might contribute to safety-related processes would provide a more comprehensive understanding of the neurophysiological mechanisms involved.

2. The experimental design specifies a video presentation interval of 2 seconds: "At every trial, in each experimental phase, participants first saw a fixation cross for 0.5s, and then a context video which was presented for 2s" (lines 557–558). However, in Figure 4B, the x-axis appears to span a cue presentation interval of 1.75 seconds. Could you clarify this apparent discrepancy?

3. In line 374, should the reference be to Figure 6D rather than Figure 6B? The description appears to correspond more closely to Figure 6D.

4. The iEEG data were acquired at two different sites, one in Paris and the other in Guangzhou, using different electrode configurations. Given that bipolar referencing was employed, differences in electrode characteristics (e.g., surface area, interelectrode distance, etc.) could influence the volume of neural tissue sampled. However, I could not find an explanation of how these differences were reconciled during the analysis. Could you provide clarification on how this issue was addressed?

5. While it is often necessary to aggregate electrode locations into broader regions, the TMP and IPFC regions described in the study appear to cover relatively large areas. Additionally, I could not find specific information on the exact locations of the electrodes within the HPC (e.g., CA sectors) and AMY (e.g., amygdala nuclei). Could you provide clarification or additional details about the precise electrode placement in these regions?

6. In the preprocessing section, it is unclear how seizure activity was excluded from the data. The manuscript only mentions the use of amplitude criteria. Could you clarify whether additional methods were employed to identify and exclude seizure activity, such as visual inspection or frequency-based criteria?

Criterion 7 - References

Comments:

I have no comments.

Criterion 8 - Clarity and context

Comments:

I have no comments.

Reviewer #3 (Remarks to the Author):

In this study, Pacheco and colleagues examine intracranial iEEG recordings to explore the neural signal that underlie extinction of fear memories in the human brain. Briefly, they employ a paradigm where subjects learn to associate individual items with an aversive stimulus during an acquisition phase. There are two categories of items, those that remain associated with an aversive stimulus during the extinction phase (CS++) and those with removal of the association with the aversive stimulus during extinction (CS+-). In this manner, they can examine how fear memories are extinguished. There is also a third category of items that are never paired with a negative aversive stimulus (CS--). Based on these conditions, the authors examine changes in theta power, previously associated with fear conditioning in the amygdala, as well as the stability of item representations. Finally, the authors introduce these items against a background of a video of different scenes, which serves as a context. This allows them to examine the stability and representations of context. Overall, this is an interesting study, and certainly leverages the rare ability to explore these neural signals with high spatial and temporal resolution. There are a number of conclusions that the authors draw from their results, but there are some suggestion that would perhaps strengthen the interpretability of their results and their conclusions.

For the behavioral analysis presented in figure 1, the authors use a paired t-test at every trial. Is This performed across subjects? It appears that is the case, but then raises the question as to how the trials may be matched across subjects, or how this is performed. For example, trial 9 may be a CS+- trial in one subject and a CS++ item in another. It would help to clarify how this is performed.

The main result regarding theta power in the amygdala that the authors present is that theta power is decreased in CS++ trials during extinction compared to CS+- trials. They interpret this to reflect a safety signal, yet it is unclear in that case why there would be no difference between CS+ and CS- trials during acquisition. Is this really reflecting safety, or some change in expectation with the trials that occurs after the CS+- items have changed their contingencies. This is especially the case since the authors present results based on CS+ versus CS- trials during extinction, so this should also be true during acquisition. If instead they are focusing on how these contingencies change, they should specifically focus on the change in contingencies, they should also examine CS++ versus CS+- trials.

Similarly, in the analysis of item stability, the authors make the primary comparison between CS+ and CS- trials. They show that CS+ trials increase their stability during extinction. However, again, this raises the question as to why they do not see these differences during acquisition. Again, if the focus here is on the change in contingencies for the CS+- items, then the comparison of item stability should also include analyses of CS++ versus CS+- items rather than all CS- items.

The authors examine coordination of item stability across brain regions by computing correlations between RSA similarity values. This is an interesting approach. Had the authors also considered simply examining standard metrics of connectivity between these brain regions during these times?

Many of the conclusions are based on the links that are thought to be established between items and context, and in particular between fear memories and context. This builds on an extensive prior literature. However, here, the role of context is unclear. As far as I can tell, it looks like the context video presented during each item is randomly chosen (from one of four videos per experimental stage), and is not directly linked with a specific item. It is certainly possible that my reading is incorrect. But if this were the case, it's not clear how context might serve a role in fear memory here, since context being randomly assigned becomes irrelevant. Instead, one would expect that a more direct analysis here would require that each item (CS+ or CS-) be linked with a specific context. Because of this, it is not clear how to interpret many of these data. For example, they show that there is greater context stability during the early stages of acquisition in HPC and during extinction only in PFC. Why only these time points and stages of the experiment, since context is presented on all trials. This makes it hard to interpret, especially since there is no direct association between context and each individual item or even contingency (CS+ versus CS-).

When looking at the correlations of context specificity, they find a significant correlation between PFC context stability and context stability in the Amygdala, but this is at a delayed time (the cluster is clearly above the diagonal in this plot). How should one interpret this, especially since there is no evidence of context stability in the amygdala.

For all analyses looking at correlations of item or context stability between brain regions, it is also not clear how these correlations may or may not be related to fear contingencies. They appear to perform these analyses by looking at all trials, but do these differ if we look at CS+ versus CS- trials, for example, or CS++ versus CS+-.

The authors also find a negative correlation between amygdala theta power and item stability in the amygdala and hippocampus. But again, it is hard to understand how to interpret this without specifically knowing how such item and context stability change between conditions, and also how theta power changes between CS++ and CS+- conditions.

Finally, the authors deploy a clever reinstatement analysis examining the similarity of items between acquisition and test and between extinction and test. They find a positive relation between context representations in the PFC and reinstatement in the TMP for CS+- trials. However, to make this analysis clearer, one would also like to know if there is a difference in reinstatement between CS++ and CS+- and CS-- trials at all. This seems like it would provide a more direct test regarding whether the item representations during acquisition and/or extinction are reinstated during test, with the hypothesis being that there will be significant differences in reinstatement between the different categories.

In several places in the text, the references to the specific figure and figure panels seem incorrect, e.g. pointing to the wrong figure panel.

Version 1:

Decision Letter:

Our ref: NATHUMBEHAV-24104353A

29th April 2025

Dear Dr Pacheco Estefan,

Thank you for submitting your revised manuscript "Representational dynamics during extinction of fear memories in the human brain" (NATHUMBEHAV-24104353A). It has now been seen by the original referees and their comments are below. As you can see, the reviewers find that the paper has improved in revision. We will therefore be happy in principle to publish it in Nature Human Behaviour, pending minor revisions to satisfy the referees' final requests and to comply with our editorial and formatting guidelines.

We are now performing detailed checks on your paper and will send you a checklist detailing our editorial and formatting requirements within two weeks. Please do not upload the final materials and make any revisions until you receive this additional information from us.

Sincerely,

Nature Human Behaviour

Reviewer #1 (Remarks to the Author):

I'd like to thank the authors for their thorough and thoughtful responses to all of my concerns. I have no further issues, and I am happy to recommend this excellent paper for publication. Congratulations on this outstanding work.

Reviewer #2 (Remarks to the Author):

Reviewer Comments on the Author's Rebuttal Letter

Criterion 1 – Data & Methodology

Comment on Response to Rebuttal 1 of Criterion 1.

The authors have adequately addressed the comment regarding the cognitive profiles of the epilepsy patients included in this study. The addition of Supplementary Table 1, containing detailed information on IQ, MQ, and seizure onset zones (SOZ) for the participants, improves the clarity and interpretability of the findings. Given that complete neuropsychological data were not available for all subjects due to clinical constraints, the authors have acknowledged this limitation and provided relevant available data.

The authors clarify that the participants' average IQ fell within the normal range, while also noting that the Memory Quotient (MQ) scores were below average. They have discussed how these cognitive characteristics might influence their findings and appropriately included this as a limitation in their discussion.

The limitation in the experimental design—specifically the limited number of visual cues and consistent instructions across trials—is reasonably defended, since it could mitigate the impact of potential cognitive impairments on task performance. Furthermore, the authors cite studies showing epilepsy patients can handle similarly complex tasks, supporting the study's context.

Overall, the authors' response satisfactorily addresses the initial concerns about cognitive factors.

Comment on Response to Rebuttal 2 of Criterion 1.

The authors have provided a clear and comprehensive response regarding the use of different co-registration methods across study sites. Their explanation—that these differences reflect institutional practices rather than technical limitations—is reasonable and aligns with common practices in multi-center iEEG research. The revision of the Methods section to highlight the shared core steps of electrode localization, including the use of standard tools such as FreeSurfer, SPM, and 3D Slicer, enhances transparency.

The inclusion of detailed descriptions of both pipelines—the EPILOC toolbox employed in Paris and the custom approach used in Guangzhou—helps establish that both follow validated and widely accepted procedures. The cross-validation analysis, in which the Paris team re-processed a subset of Guangzhou data and arrived at consistent electrode labels, provides additional empirical support for the reliability of their co-registration methods.

Moreover, the authors' clarification that most analyses were based on broader anatomical regions helps alleviate concerns about minor localization discrepancies.

Overall, the authors' response adequately addresses the concern and improves confidence in the methodological consistency and robustness of the study.

Comment on Response to Rebuttal 3 of Criterion 1.

The authors have provided a detailed and well-structured justification of their experimental design that addresses the concerns raised. They clarify that while the paradigm incorporates specific elements from prior studies—such as the 50% partial reinforcement rate used by Dunsmoor et al. (2014)—it was not directly based on that work. This distinction is now clearly stated in the revised manuscript.

The authors outline how each component of the paradigm is informed by existing literature, drawing on a broad range of studies from both human and animal fear conditioning research. The rationale for using a multi-cue structure, the ABC contextual framework, naturalistic stimuli and context videos, and the inclusion of a cognitive narrative is systematically presented and supported by relevant references. The use of multiple context videos per phase facilitates representational similarity analysis (RSA) and aligns with the methodological goals of the study.

The response also addresses practical considerations related to the clinical setting, such as the absence of individualized

calibration for stimulus aversiveness and the timing of testing sessions. These decisions are explained transparently and are consistent with approaches taken in related studies. The mention of a recent fMRI study by the same group using a similar design, while not central to the current work, supports the feasibility and applicability of the paradigm. Overall, the response clearly explained theoretical foundation of the study design. Particularly notable is that they included a comprehensive Supplementary Note.

Comment on Response to Rebuttal 4 of Criterion 1.

The authors have clearly explained the categorization of CS- trials during the extinction phase. They acknowledge that the original figures grouped both CS+- and CS-- trials under the CS- label and have revised the Methods section and corresponding figures (Figures 2 and 3) to reflect this distinction more accurately.

The updated figures, which include separate analyses for CS++, CS+-, and CS-- trials, improve the interpretability and specificity of the results. The observation that amygdala theta power differs between CS++ and both CS+- and CS-- trials—but not between CS+- and CS-- trials—supports the interpretation that neural responses during extinction are more strongly influenced by current stimulus valence than by prior reinforcement history. This pattern is also evident in the item stability analyses for both the amygdala and the temporal pole, where the most pronounced differences occur between CS++ and the two CS- conditions.

The interpretation that extinction leads to previously threatening (CS+-) cues being represented more similarly to consistently safe (CS--) cues is supported by the data. In addition, the discussion of residual fear memory reinstatement effects, as reflected in context representation dynamics (Figure 6), provides additional perspective on the neural processes underlying extinction.

Overall, the authors' clarification and additional analyses adequately address the concern.

Comment on Response to Rebuttal 5 of Criterion 1.

For my concern regarding the reported theta power levels for CS++ trials during extinction, they appropriately note that the early effects did not survive correction for multiple comparisons when applying a cluster-based permutation approach. This statistical rationale—emphasizing the increased likelihood of false positives given the large number of time-frequency bins analyzed—is consistent with current standards in time-frequency analysis.

The authors have also revised the Discussion section to explicitly clarify this point, thereby reducing the risk of misinterpretation. Their conclusion that the later significant effects (1.18–1.75 s) represent the only reliable indicators of reduced theta power in response to threatening stimuli is supported by the corrected analysis.

Overall, this response clarifies the statistical interpretation of the findings.

Criterion 3 - Appropriate use of statistics and treatment of uncertainties

Comment on Response to Rebuttal 1 of Criterion 3.

For the statistical concerns raised, they clearly explain the rationale for setting the minimum number of trials to eight, referencing Oehm et al. (2015) to support the view that fewer than five trials are generally insufficient for reliable hippocampal theta analyses in iEEG research. This reference offers a relevant justification for the chosen threshold. In addition, control analyses that vary the minimum number of trials (5, 8, and 10) indicate that the primary findings are robust across different inclusion criteria, supporting the validity of the reported effects.

Regarding the behavioral data, the authors include a follow-up analysis of the test phase using one-way ANOVAs and post hoc comparisons. These analyses indicate that CS++ items continued to be rated as more threatening than CS+- and CS-- items, consistent with partial fear renewal. The clarification that no progressive learning effects during the test phase survived correction for multiple comparisons helps to more accurately frame the behavioral findings.

The authors also address concerns about statistical power and sample size, noting the advantages of their within-subjects design and reporting effect size estimates (partial eta squared) that suggest the sample size was sufficient to detect key effects.

Finally, the authors examine the normality assumptions underlying their statistical tests. In addition to reporting the limitations of parametric analyses in certain trial-level behavioral measures, they provide results from non-parametric Wilcoxon and Friedman tests. The replication of core findings using these methods supports the robustness of the analyses. Overall, the response provides a clear and methodologically grounded justification of the statistical approach and supports the reliability of the study's conclusions.

Criterion 6 - Suggested improvements

Comment on Response to Rebuttal 1 of Criterion 6.

The authors have provided a detailed response to the previous comment concerning the role of faster-cycle neural oscillations in threat and safety responses. The additional paragraph in the Discussion section (lines 490–508) situates the current findings within a broader neurophysiological framework by considering gamma oscillations. The authors outline relevant theoretical perspectives and empirical evidence, including mechanisms such as theta-gamma coupling and spike-field coherence, that may underlie the functional role of gamma activity.

They also acknowledge the absence of significant gamma effects in their own dataset and suggest potential directions for future research, such as analyses of phase-amplitude coupling and interactions with hippocampal representational patterns. This addition provides a more comprehensive context for interpreting the current results and clarifies the potential involvement of high-frequency oscillations in signaling processes related to threat and safety.

Comment on Response to Rebuttal 2 of Criterion 6.

I have no comments.

Comment on Response to Rebuttal 3 of Criterion 6.

I have no comments.

Comment on Response to Rebuttal 4 of Criterion 6.

The authors have provided a detailed response to the concern regarding potential differences in electrode configurations between the Paris and Guangzhou cohorts. They outline how variables such as electrode type, contact spacing, and referencing schemes may influence the volume of neural tissue sampled, and describe the steps taken to account for these factors in their analyses.

Several strategies were used to assess and mitigate potential bias, including restricting analyses to gray matter contacts, averaging within subjects before conducting group-level comparisons, and evaluating sampling density using Delaunay triangulation. The analysis indicated no significant differences in electrode density between the two cohorts, supporting the comparability of the datasets.

Furthermore, the authors report that the key finding—amygdala theta power differences—was replicated independently in each cohort. This analysis, presented in Supplementary Figure 2, provides additional support for the robustness of the results across sites. The inclusion of this information, along with discussion of relevant considerations in multi-center iEEG research, contributes to the overall methodological transparency of the study.

These clarifications collectively address the concern and support the validity of the reported findings across different clinical settings.

Comment on Response to Rebuttal 5 of Criterion 6.

The authors have provided a clear response to the concern regarding the spatial specificity of electrode placement within large regions of interest (ROIs) such as the temporal pole (TMP), lateral prefrontal cortex (IPFC), hippocampus (HPC), and amygdala (AMY). The rationale for aggregating subregions—based on anatomical constraints, existing literature, and the need to maintain sufficient statistical power—is consistent with established practices in iEEG research.

The explanation of how subregions were grouped is supported by visual documentation (Supplementary Figure 3) and a comprehensive list of electrode coordinates (Supplementary Table 2), which contribute to transparency. The inclusion of Supplementary Video 1 further illustrates electrode placement in the AMY and HPC.

Although the study did not aim to differentiate subnuclei or subfields within the AMY or HPC, the authors provide a clear justification for this decision and acknowledge the methodological trade-offs associated with their representational analysis approach.

Overall, the response addresses the concern regarding the spatial extent of the TMP and IPFC regions by providing a rationale for the chosen level of aggregation and supporting it with visual and tabular documentation. While the study did not resolve electrode placement at the level of hippocampal subfields or amygdala nuclei, the authors offer a justification for this approach and acknowledge the associated limitations, thereby clarifying the level of anatomical specificity achieved in the current analysis.

Comment on Response to Rebuttal 6 of Criterion 6.

The authors have provided a detailed clarification regarding their approach to excluding seizure-related activity from the iEEG data. The description of the automatic artifact detection algorithm indicates that the procedure incorporates multiple features—such as amplitude, gradient, and high-frequency activity—based on established methods (Staresina et al., 2015). The use of z-scoring and conservative thresholding suggests that the approach is designed to be sensitive to epileptiform events.

In addition to automated detection, the authors confirm that all identified segments underwent visual inspection of both raw signals and time-frequency spectrograms. This combination of automated and manual review contributes to the reliability of the artifact rejection process. The additional step of excluding data 1,000 ms before and after each artifact point further reduces the risk of contamination by residual epileptic activity.

These clarifications in the revised manuscript address the initial ambiguity and provide a more complete account of how seizure activity was excluded, thereby increasing confidence in the reliability of the preprocessing methods.

Reviewer #2 (Remarks on code availability):

Reviewer comments regarding code reproducibility and syntax errors:

I prioritized reviewing the scripts listed in the provided README.md, specifically:

- behavior.m (for reproducing behavioral results, Fig. 1)
- power_analysis.m (for reproducing theta power results, Fig. 2)
- contrast_based_RSA.m (for reproducing item stability and specificity, Figs. 3-4)
- trial_level_analysis.m (for coordination analyses across ROIs, Figs. 3-5)

The following issues were identified during this review:

1. Critical reproducibility issue – Missing dependency: load_paths_EXT.m

All provided analysis scripts begin with:

```
paths = load_paths_EXT;
```

However, this essential function (load_paths_EXT.m) is missing from the repository and explicitly excluded via .gitignore.

There are also no instructions or templates provided for creating or configuring this file. Consequently, the scripts cannot currently be executed.

Required action:

- Include a default load_paths_EXT.m file, or
- Provide a clearly documented template for users, or
- Add detailed instructions in the README for configuring this file.

2. Missing data files (*_trinfo.mat)

The behavior.m script explicitly requires subject-specific trial information files (e.g., c_sub01_trinfo.mat). These files were not provided.

Required action:

- Supply the missing *_trinfo.mat files, or
- Provide instructions and scripts for generating these files from raw data.

3. Missing file and required structure (allS_AMY_C.mat)

The scripts depend on allS_AMY_C.mat located in paths.results.power, containing an ALLEEG cell array with EEG channel locations (EEG.chanlocs). Without this file, the analyses cannot be executed.

Required action:

- Provide the allS_AMY_C.mat file or clear instructions on generating it.

4. Missing additional data files

The correlation analysis requires additional files from the paths.results.trial_based directory:

- AMY_POW_1-44Hz_TR.mat
- trlSTA_HPC_CE_1-44_1_0_500-50.mat

These files must contain the specific structures (allPOWAMY, itstaTRALL, or ctxTRALL) used for trial-wise analyses.

Required action:

- Provide the listed .mat files or detailed instructions and scripts to generate them.

Additionally, I reviewed the connectivity.m script separately and found the following syntax error:

```
amyDF = eegfilt(amyD,1;;; f2u(1), f2u(end));
```

The syntax 1;;; is invalid in MATLAB. The corrected syntax should be:

```
amyDF = eegfilt(amyD, 1, f2u(1), f2u(end));
```

Please also verify the accuracy of using 1 as the sampling rate parameter (srate), as EEG analyses typically require precise sampling rate information.

Required action:

- Correct the syntax error.
- Confirm and correct the sampling rate parameter if necessary.

Reviewer #3 (Remarks to the Author):

The authors have provided a revision of their manuscript, and have comprehensively addressed all of the points raised in the initial review. This is an interesting study, and could be of interest to a number of researchers across different fields.

Reviewer #1 (Remarks to the Author)

The reviewed study, Representational dynamics during extinction of fear memories in the human brain, aimed to examine the neurophysiological and representational characteristics of contextual fear conditioning and subsequent extinction learning. For this purpose, the authors conducted an elaborate multilab, multinational intracranial EEG (iEEG) study. The time-frequency characteristics of the iEEG signal were analyzed using classical Morlet wavelet analysis, extended with representational similarity analysis (RSA). The employed methods appear robust and suitable for addressing the stated research questions. However, I should note that I lack firsthand experience with RSA.

Overall, the study addresses a highly relevant topic. A better understanding of the neurophysiological underpinnings of threat processing could prove highly valuable for a better understanding of conditions such as anxiety disorders. By leveraging the high spatial resolution of iEEG, the current study records EEG activity from subcortical structures like the amygdala and hippocampus, which are key nodes in the neural fear and extinction circuitry. This approach also helps to bridge the translational gap between human and non-human studies. In sum, the study is well-written and engaging, features high-quality figures to illustrate the observed data, and should be of great interest to the neuroscientific and clinical community. However, before I can recommend it for publication, I would like the authors to address the following points:

We thank the referee for their positive evaluation of our manuscript. Below, we provide our responses to all the points raised and indicate where changes have been made in the original manuscript to address their concerns. Our responses are highlighted in blue.

1. The authors mention in lines 78-80, that rodent studies provide evidence that theta oscillations in the amygdala are associated with fear extinction. However, most of the cited rodent studies demonstrated that amygdala theta oscillations are increased during fear expression or during the presentation of the threat predictive CS+ (e.g., Pare & Collins, 2000, Seidenbecher et al., 2003, Likhtik et al., 2014). Moreover, there is growing evidence from both animal and human research that theta oscillations within the neural fear circuitry (i.e., amygdala, hippocampus, and parts of the mPFC) play a significant role in fear expression. For instance, previous animal studies have especially demonstrated increased theta oscillations in the prelimbic cortex/dACC and/or amygdala during states of fear expression (e.g., Seidenbecher et al., 2003; Courtin et al., 2014; Fenton et al., 2014; Likhtik et al., 2014; Lesting et al., 2011 and 2013; Taub et al., 2018; Rahman et al., 2018, Karalis et al., 2016). Additionally, human EEG studies have also reported

heightened frontocentral theta oscillations, particularly during fear conditioning (e.g., Pirazzini et al., 2023, Chen et al., 2021) and fear recall (e.g., Mueller et al., 2014; Sperl et al., 2019; Bierwirth et al., 2021; but see Bierwirth et al., 2023), with some evidence suggesting a correlation between these frontal theta oscillations and the amygdala BOLD response (Sperl et al., 2018). Given this body of evidence, it is surprising that the current study did not observe increased theta oscillations in the PFC and amygdala. It would be valuable if the authors included this evidence in the introduction and acknowledged and discussed this absence of effect more thoroughly in the discussion.

We thank the reviewer for providing these relevant references. We have incorporated them in the introduction of the revised manuscript and modified the text to clarify and expand upon the role of amygdala theta oscillations in fear acquisition, expression and extinction. Additionally, we have highlighted that theta oscillations are a signature of conditioning across multiple areas of the fear and extinction network, including regions in the prefrontal cortex and the hippocampus. Finally, we have distinguished between studies that reported oscillatory power effects and those that examined connectivity effects in the theta frequency band. The revisions are reflected in lines 80-94 of the introduction, which now state:

“Research in rodents has provided important insights into the neural mechanisms of fear acquisition, expression and extinction. Notably, several studies have shown that the amplitude of theta oscillations in the prelimbic cortex, dorsal anterior cingulate cortex (dACC), and/or amygdala increases during fear acquisition (Paré & Collins, 2000) and expression (e.g., Courtin et al., 2014; Fenton et al., 2014; Karalis et al., 2016; Lesting et al., 2011, 2013; Likhtik et al., 2014; Rahman et al., 2018; Seidenbecher et al., 2003; Taub et al., 2018). Similarly, human EEG studies have reported heightened frontocentral theta oscillations, particularly during fear conditioning (e.g., Chen et al., 2021; Pirazzini et al., 2023) and fear recall (e.g., Bierwirth et al., 2021; Mueller et al., 2014; Sperl et al., 2019; but see Bierwirth et al., 2023). In addition, studies in rodents have shown that long range theta (2-12Hz) connectivity across regions of the extinction network, including AMY, PFC and HPC, has been associated with the discrimination of aversive and safe cues (Likhtik et al., 2014), and with the extinction of fear associations (Lesting et al., 2011, 2013; Nguyen et al., 2023).”

We also refer to a paper that specifically reported a role of theta oscillations in fear extinction (Nguyen et al., 2023). This study showed that fear extinction involves a safety signal encoded by synchronized AMY-HPC activity, with theta oscillations in the HPC playing a critical role. In addition, we highlight that in the studies by Lesting et al., (2011, 2013), mentioned by the reviewer, a role of theta oscillations in both fear expression and extinction is reported. The study of Lesting et al. (2011) demonstrated increased theta (2–12 Hz) coupling between the mPFC, HPC, and AMY during fear retrieval both during and after extinction. The study of Lesting et al. (2013) demonstrated a directional interaction

of prefrontal neuronal spiking and hippocampal and amygdala theta oscillations during extinction recall, confirming that connectivity in the theta frequency band is related to changes in appraisal of threatening stimuli during extinction.

We agree that the absence of an effect in theta power during fear acquisition in our study may appear unexpected given prior literature. Several factors might explain this fact: 1) **The use of different recording modalities.** Many previous studies in humans, such as those by Pirazzini et al. (2023), Mueller et al. (2014), Sperl et al. (2019), and Bierwirth et al. (2021), relied on scalp EEG recordings. EEG captures activity from wide-spread neocortical areas, which differs from the more local signals recorded with iEEG electrodes, in particular when a bipolar reference scheme is employed as in our study. While some simultaneous iEEG/EEG studies have shown that it is possible to localize scalp EEG activity to the HPC or the AMY (Dalal et al., 2011), this has not commonly been applied in fear learning and extinction studies. Thus, it remains a bit unclear whether previous studies using scalp EEG indeed captured specific activity from these deep brain structures. 2) **Differences in the exact localization of electrodes.** The study by Chen et al. (2021), using iEEG recordings, reported theta power increases during fear acquisition in the dorsomedial and ventral PFC (dmPFC and vmPFC). We did not target these regions due to limited electrode coverage in our group of patients and instead focused on the lateral prefrontal cortex (IPFC) and the orbitofrontal cortex (OFC). Notably, Chen et al. (2021) also observed AMY theta power effects during acquisition, a finding we could not replicate. Differences in the experimental paradigm employed—a more cognitive and contextual paradigm in our case vs. the use of electric shocks in Chen et al. (2021)—might partially explain these differences (see below, point 4). 3) **Species-specific distinctions in the neurophysiological properties of theta oscillations.** Theta oscillations in rodents have marked properties that differ from those of humans, e.g., they are more sustained during navigation (Jacobs, 2014). In addition, human hippocampal theta oscillations occur in two different frequency ranges – a lower range in the delta/theta band (2.5-5 Hz) and a higher range at the border between theta and alpha (5.5-10Hz) – that appear to have different and maybe even opposing cognitive functions (Herweg et al., 2020) and may relate to different theta generators in rodents (Buzsáki, 2002). It is possible that similar differences exist for fear learning and extinction. 4) **The use of a ‘cognitive’ and contextual paradigm with screams as aversive stimuli in our study.** The relatively mild nature of the US in our study contrasts with the stronger aversiveness of electric shocks typically applied in rodent studies, which elicit powerful autonomic, behavioral and neurophysiological responses in animals (e.g. theta oscillations related to freezing; Courtin et al., 2014; Fenton et al., 2014; Karalis et al., 2016; Rahman et al., 2018; Seidenbecher et al., 2003; Taub et al., 2018).

To address these points, we have expanded the discussion in lines 451-466, where we now write: “*While a previous human iEEG study reported increases in theta power for*

CS+ vs. CS- items during acquisition (Chen et al., 2021), our results showed higher theta power for CS- vs. CS+ items during extinction. This apparent discrepancy may be explained by the differing cognitive demands of acquisition and extinction, with the latter requiring a more pronounced coordination between AMY and other brain regions coding for contexts and cognitive control such as HPC and PFC. Indeed, increases of AMY theta power during extinction were correlated with higher levels of item stability in both AMY and HPC and with higher levels of context specificity in IPFC. The lack of AMY theta power effects during acquisition in our study contrasts with findings in animals and humans, which have typically reported theta increases during the formation of cue-US associations across several regions of the fear and extinction network (e.g., Chen et al., 2021; Pirazzini et al., 2023). Several factors may explain this discrepancy, including differences in recording methodologies (iEEG versus EEG), the distinct regions targeted (e.g., different subregions of the PFC targeted in [Chen et al., 2021]), interspecies differences in theta oscillations, and the use of a more “cognitive” paradigm in our study with a milder aversive stimulus.”

We hope these clarifications address the reviewer’s concerns and enhance the comprehensiveness of the manuscript.

2. The authors relied on “threat” and “safety” ratings to validate the fear conditioning paradigm. Typically, most fear conditioning studies utilize physiological measures such as skin conductance responses, fear-potentiated startle, or evoked heart rate responses to assess fear responses. While this might not have been feasible in the current study due to its cross-cultural, multilab nature, solely relying on rating data—highly susceptible to demand effects—is a limitation that should be discussed. This is particularly important given the use of a complex contextual fear conditioning paradigm, which is not widely employed in the fear conditioning literature.

Our decision to use explicit ratings rather than physiological recordings was guided by both ethical and practical considerations, particularly given the constraints of conducting research with implanted epilepsy patients. To minimize interference with clinical monitoring and ensure compliance, we chose to use a mild aversive stimulus (a scream) rather than electric shocks. This decision was essential for the viability of our study. However, it also limited the applicability of methods such as electrodermal activity (EDA) or heart rate recordings, which are better suited for capturing stronger autonomic reactions.

We acknowledge that self-reports can be influenced by demand effects or biased by participant interpretations and now mention this as a limitation of our study (see added text below). However, we would also like to note that such effects are unlikely to play a major role in our paradigm due to its simplicity and the absence of explicit experimenter

expectations beyond learning the correct CS-US associations. Moreover, participants were explicitly instructed to evaluate the ‘safety’ or ‘threat’ of the cue itself rather than their internal emotional state, minimizing subjective bias.

Importantly, in our paradigm, apart from the analysis of the correlation of the ratings with reinstatement in the TMP (Figure 7), we did not employ the ratings to sort the stimuli into categories or correlate them with the neural data in any other analysis. Notably, this restricts their relevance in interpreting most of our neural results. Nevertheless, we note that our ratings show learning curves that match what would be expected for skin conductance. For instance, the study of (Ma et al., 2025) showed that differences in skin conductance responses for CS+ and CS- items increase linearly until trial 9 during acquisition before plateauing, consistent with our behavioral results.

We would also like to mention that several other previous fear conditioning studies employed self-reports as well. Several studies have used explicit ratings of “shock expectancy” alone (Bandarian Balooch & Neumann, 2011; Neumann et al., 2007; Neumann & Kitlertsirivatana, 2010), while others have applied them in combination with physiological measures, in particular EDA (Shiban et al., 2013; Taschereau-Dumouchel et al., 2020), and fear startle responses (Balooch et al., 2012; Lissek et al., 2008). These studies demonstrated consistency between explicit reports and startle responses (Balooch et al., 2012; Lissek et al., 2008), as well as between fear reports and skin conductance (Constantinou et al., 2021; Shiban et al., 2013; Taschereau-Dumouchel et al., 2020). Together, these findings support the validity of assessing fear acquisition and extinction with expectancy ratings (see also Boddez et al., 2013).

Expectancy ratings also offer advantages over physiological measures, as they are more stable across repeated measurements (Corneille & Gawronski, 2024), facilitating comparisons with previous research (Balooch et al., 2012; Bandarian Balooch & Neumann, 2011; Kinner et al., 2016; Lissek et al., 2008; Lissek & Tegenthoff, 2024; Neumann et al., 2007; Neumann & Kitlertsirivatana, 2010; Shiban et al., 2013).

Finally, some research suggests that the conscious and deliberative perceptions of CS-US contingencies—directly measured by expectancy ratings—might affect acquisition and extinction. In that sense, the use of expectancy ratings fits well with the cognitive and contextual nature of our paradigm. The use of explicit ratings is thus consistent with previous “cognitive conditioning” setups that employed fictitious and “cognitively aversive” stimuli (Kinner et al., 2016; Lissek & Tegenthoff, 2024; see Methods and response to Reviewer 2, Criterion 1, *Data & Methodology*, Point 3).

We have modified the Discussion section of the manuscript (lines 641-672) to incorporate these considerations:

“We note that, similar to other fear conditioning paradigms conducted with humans (e.g., Balooch et al., 2012; Bandarian Balooch & Neumann, 2011; Kinner et al., 2016; Lissek et al., 2008; Lissek & Tegenthoff, 2024; Neumann et al., 2007; Neumann & Kitlertsirivatana, 2010; Shiban et al., 2013), we chose to rely on self-reported ratings of cue ‘threat’ and ‘safety’ as our behavioral measure of fear rather than employing implicit physiological recordings. While explicit ratings have limitations, we decided to employ them based on both ethical and clinical considerations, particularly given the constraints of conducting research with implanted epilepsy patients. Specifically, the use of a mild aversive stimulus (a scream) rather than electric shocks in our study precluded the use of some traditional implicit measures, such as skin conductance or heart rate responses, which are more suited for capturing stronger autonomic reactions. Importantly, research indicates that explicit ratings are more stable than physiological responses across repeated measurements (Corneille & Gawronski, 2024), and several reviews support their validity in fear conditioning research (Boddez et al., 2013; Constantinou et al., 2021). Explicit self-reports have been shown to correspond with both fear startle (Balooch et al., 2012; Lissek et al., 2008) and skin conductance responses (Constantinou et al., 2021; Shiban et al., 2013; Taschereau-Dumouchel et al., 2020). Moreover, self-reports might better capture deliberative processes than implicit measures (Lonsdorf et al., 2017), which aligns with the more “cognitive” nature of our paradigm (see also Kinner et al., 2016; Lissek & Tegenthoff, 2024). Although explicit ratings are susceptible to demand effects or participant interpretation, such effects are unlikely to have played a major role in our results due to the simplicity of the task and the absence of explicit experimenter expectations beyond learning the correct CS-US associations. Crucially, participants were instructed to rate the ‘safety’ of the cue itself rather than their internal emotional state, minimizing subjective bias. Furthermore, apart from analyzing the correlation of the ratings with reinstatement in the TMP (Figure 7), we did not rely on the ratings to categorize stimuli or correlate them with the neural data in other analyses, which minimizes their influence in the overall interpretation of our findings. Finally, we highlight that the trial-by-trial learning effects we observed based on the ratings align closely with what would be expected for skin conductance responses (see for example, Ma et al., 2025)”.

3. In line 533, the authors mention that they used an aversive scream as the unconditioned stimulus (US). However, crucial details about the US are missing. Could the authors provide more information about the US? Specifically, at what decibel level was the US presented, and how was its level of aversiveness determined?

We chose to use a milder aversive stimulus (a scream) rather than electric shocks to minimize interference with clinical monitoring of the implanted epilepsy patients. We did not conduct individualized aversiveness testing for the US due to time constraints and the

limited availability of patients in the clinical environments. Instead, as in previous studies (Schmitz & Grillon, 2012; Wehrli et al., 2022), we selected a notoriously aversive stimulus—a loud scream—and adjusted its intensity to highly unpleasant level. The sound was delivered via computer speakers for 1 second, at an intensity of ~85 decibels. We have now included this information in the Methods section of the revised manuscript (lines 776-781):

“The US consisted of an unpleasant scream, which was delivered via the computer speakers. The sound levels of the scream were at ~85 decibels, resulting in a highly aversive but not painful amplitude. The duration of the scream was 1s. Given the 50% reinforcement rate of the US, the scream was presented a total of 36 times in the experiment (CS++: 24 times across acquisition and extinction; CS+- 12 times only during acquisition)”.

4. How many iEEG artifacts occurred for each stimulus? Were the number of artifacts comparable across conditions?

We separately computed artifacts for each electrode using an automatic procedure followed by visual inspection of the raw amplitude time series and spectrograms, without any knowledge about the specific conditions of the experiment. Given that the automatic procedure was conducted for every channel separately, the number of trials differs slightly between the different ROIs.

We present the number of trials included in each of our ROIs in the new Supplementary Table 3 (shown below), which shows that the number of trials across conditions is comparable. We also highlight that since some of the RSA analyses rely on the comparison of many items of the same type (e.g., correlations are computed between all pairs of CS+ items during a specific task stage), the statistical power of these analyses is much higher. The number of trials below applies to the power analysis (Figure 2) and to the trial-level RSA analysis (Figure 3 and 4).

ROI	CS++	CS+-	CS--
AMY	38.41 ± 8.01	37.18 ± 8.84	37.78 ± 9.31
HPC	37.44 ± 8.32	36.53 ± 8.86	36.47 ± 8.72
OFC	39.17 ± 9.55	40.33 ± 9.57	39.83 ± 9.75
IPFC	40.78 ± 8.72	40.74 ± 9.67	41.26 ± 9.43
TMP	26.54 ± 12.72	27.89 ± 12.41	26.85 ± 12.43

Supplementary Table 3: Number of excluded trials after artifact rejection in every ROI across all task stages. Please Note that since some of the RSA analyses rely on the comparison of many items of the same type (e.g., correlations are computed between all pairs of CS+ items during a specific task stage),

the statistical power of these analyses is much higher. The number of trials in the table applies to the power analysis (Figure 2) and to the trial-level RSA analysis (Figure 3F and 4D).

5. Was the study preregistered?

Our study was not preregistered, and we recognize this as a limitation, as we acknowledge that preregistration enhances the integrity of hypothesis testing and promotes the reproducibility of findings. However, for several reasons that we outline below, preregistration was particularly challenging in our study.

First, our study is the first to investigate extinction learning in humans via iEEG, making it difficult to formulate precise hypotheses or define a comprehensive preregistered analysis plan. Second, our study involved the integration of methodological approaches from different research domains. We drew upon established metrics from episodic memory research, particularly the use of time-resolved RSA to investigate “context specificity” and “item stability”. The application of these metrics in the context of fear conditioning was unprecedented, making it challenging to anticipate all possible outcomes in advance. Finally, we developed novel metrics specifically tailored to our research questions, i.e., to assess the relationship of context specificity with fear reinstatement in the analysis presented in Figure 6. These novel methods introduced additional uncertainty.

Despite these challenges, we note that we articulated our main hypotheses in the introduction and the results section of our original manuscript. Specifically, we selected regions of interest (ROIs) based on prior studies that identified the fear learning and extinction network in animals and humans (lines 103-106). We also hypothesized that theta frequency oscillations would play a crucial role in the formation of fear and safety associations, building on our previous research into item-context associations in the hippocampus (lines 123–125). We predicted that item stability would be linked to fear memory formation (lines 127-130), drawing from prior findings during episodic memory formation (Lu et al., 2015; Xue et al., 2010) and fear conditioning (Visser et al., 2013). Finally, we anticipated that PFC context specificity would correlate with the reinstatement of fear rather than safety memory traces (lines 337-340), based on previous work in rodents (Gilmartin et al., 2014; Maren et al., 2013) and humans (Craske et al., 2014; Garfinkel et al., 2014; Milad & Quirk, 2012; Wang et al., 2024).

To ensure transparency, we have now included a paragraph in the discussion explicitly acknowledging the limitation of not preregistering our study (see below, lines 673-702). We also clarify which of our findings aligned with our initial hypotheses and which were unexpected. While further validation will be needed in future research, our study constitutes an important first step in understanding the representational dynamics of context-dependent extinction learning in humans.

“Our study was not preregistered, and we acknowledge this as a limitation, as preregistration enhances the transparency and reproducibility of research findings. However, the exploratory nature of our study, which is the first to investigate context-dependent extinction learning in humans via iEEG, made preregistration challenging. Our research aimed to provide a first characterization of the representational dynamics underlying the context-dependent nature of extinction learning—which required the integration of methodologies from diverse domains and the development of novel metrics. For example, we adapted RSA, an established method in episodic memory research, to examine context specificity and item stability within a classical ABC context-dependent extinction paradigm (Figure 2 and 3). We also developed a new measure to quantify fear reinstatement in the context of classical conditioning (see Figure 6). Despite the pioneering and exploratory nature of our study, we note that we formulated specific predictions in the introduction and results sections.

Notably, some of our results were not in line with our predictions. Specifically, we anticipated to observe increased theta power for CS+ as compared to CS- items during acquisition in AMY, which we did not find. Instead, we observed a relative decrease of power for these trials during extinction in this region (see section above: A safety signal in the AMY driven by context representations in the extinction network). We also expected context-specific signals both in IPFC and HPC during extinction based on previous literature (Maren et al. 2013). However, context specificity was only observed in the IPFC during extinction, while the involvement of the HPC was only observed during acquisition (see section Fear learning, extinction, and episodic memory above). Finally, we expected IPFC context specificity to drive context representations in AMY and HPC, but while IPFC and AMY context representations were indeed correlated across trials, we did not find a main effect of context specificity during extinction in either AMY or HPC (see section A safety signal in the AMY driven by context representations in the extinction network above). Despite its exploratory nature, our work represents an important first step in exploring the representations underlying context-dependent extinction learning in humans using iEEG, providing a foundation for more targeted, hypothesis-driven research in the future”.

Reviewer #2 (Remarks to the Author)

In rodents, theta (4-8Hz) oscillations in amygdala and hippocampus play critical roles during both fear learning and extinction. Also extinction relies on the formation of novel, highly context-dependent memory traces that suppress the initial fear memories. However, it remains unknown whether similar processes occur in humans and how they relate to previously described neural mechanisms of episodic memory formation and retrieval.

The authors analyzed Intracranial EEG (iEEG) recordings in epilepsy patients in the deep brain structures of the fear and extinction network, and they employed representational similarity analysis (RSA), which allows characterizing the memory traces of specific cues and contexts. In this study, they found that amygdala theta oscillations during extinction learning signal safety rather than threat and that extinction memory traces are characterized by stable and context-specific neural representations that are coordinated across the extinction network. They further demonstrate that context specificity during extinction learning predicts the reoccurrence of fear memory traces during a subsequent test period, while reoccurrence of extinction memory traces predicts safety responses.

The authors meticulously analyzed iEEG data recorded using the ABC paradigm with successfully drawing interesting results in the role of theta oscillations in amygdala and hippocampus.

It is a well conducted and reasonably construed study.

Also their results look very persuasive.

However, I have several major concerns to accept their results.

We thank the referee for their favorable assessment of our manuscript and the constructive suggestions for improvement. Below, we provide a point-by-point response to all the issues raised, including additional information and novel analyses that we believe have significantly improved our manuscript and effectively address the reviewer's concerns.

Criterion 1 – Data & Methodology

Comments:

1. This study analyzed iEEG in 'medically intractable epilepsy participants'. Since antiepileptic drugs could potentially impair cognitive function, it would be helpful to clarify whether the participants had sufficient cognitive abilities to perform the experimental paradigm. Providing information on the participants' IQ, MQ, and seizure onset zone for each subject would allow for a more thorough assessment.

We thank the reviewer for raising this important point. To provide a comprehensive assessment of our patients' cognitive and clinical profiles, we have included a new Supplementary Table 1 in our manuscript (see below), detailing patient's gender, age, global Intelligence quotient (IQ), Memory quotient (MQ, as assessed by the Wechsler Memory Scale), and Seizure onset zone (SOZ). While we were able to provide data for most of our participants, unfortunately not all information was collected for every subject in the hospital environments. For example, the IQ test was only performed for 33 out of 50 patients, and the MQ test was performed for 26 out of 50 subjects. Importantly, for the subset of patients with available IQ and MQ data, we can confirm that overall IQ levels fell within the normal range (94.15 ± 11.83). The memory quotient, based on the Wechsler memory scale (WMS-III), was lower than average, reflecting reduced memory in a substantial proportion of epilepsy patients ($MQ = 73.73 \pm 20.8$). While this is not a major concern for our study because we do not specifically probe the types of memory that the Wechsler scale addresses, we do acknowledge this possible limitation in the discussion section (see below).

We note, however, that the simplicity of our task minimizes the likelihood that potential cognitive impairments of the patients affected the results. Our task only contains one instruction that is applicable to all trials throughout the experiment, i.e., to evaluate whether the visual cues are safe or threatening. Only three cues are presented (a toaster, a hairdryer, and a fan), which are consistently associated with the same contingencies in CS++ and CS-- trials. Only in the CS+- trials the contingencies change between acquisition and extinction. The simplicity of the task and of the instructions minimize the risk of confusion. Indeed, as reported in the Results section (lines 165-196), our behavioral data shows that patients were able to learn cue-US associations during the acquisition phase and to adapt to the change in contingencies during extinction in the CS+- trials (Figure 1D).

Notably, as has been shown in previous studies by our group and others, implanted epilepsy patients can perform complex tasks that are substantially more demanding at levels comparable to healthy populations. For example, in a study we published recently (Pacheco Estefan et al., 2021), the task combined spatial navigation with a complex visual recognition memory test, and we could directly compare performance in patients and healthy controls, showing no significant differences in overall recognition memory performance. Other studies with implanted patients have employed complex perceptual and working memory tests, and challenging episodic memory tests (Y. Chen et al., 2021; Lopour et al., 2013; Miller et al., 2013). While we acknowledge that different groups of patients may show different levels of performance, the fact that similar patient populations were able to conduct even more complex tasks shows the general ability of implanted patients to perform cognitively demanding experiments.

Regarding the SOZ, we note that the affected brain regions in our patient cohort primarily involved unilateral and (less often) bilateral temporal regions, with some involvement of frontal and insular areas. In total, 32,65% of patients had a SOZ in the left hemisphere, 40,82% had a SOZ in the right hemisphere, and 26,53% had bilateral SOZs. A total of 83,67% of patients had a SOZ in temporal regions, while 8.16% of subjects had a SOZ in frontal regions, 24.49% had SOZ in the insula, and 6.12% had a SOZ in the limbic system. We applied a conservative automatic algorithm to reject any epileptic activity and visually inspected the data to remove any remaining contamination by epileptic artifacts (see also Methods, line 660-676 and our response to point 6 Criterion 6 *Suggested Improvements*). More generally, it is unlikely that epileptic activity biased our results, as most analyses involved comparisons across conditions, making it improbable for epileptic activity to disproportionately affect one condition.

We have added a dedicated paragraph in the Discussion section (lines 622-640) clarifying the potential impact of cognitive impairments in our group of subjects. Specifically, we write:

“Cognitive impairments in implanted epilepsy patients are generally a concern when conducting cognitive experiments using iEEG data. To facilitate the interpretation of our findings, we provide in Supplementary Table 1 all available demographic and clinical information of our patients, including global intelligence quotient (IQ), memory quotient (MQ; assessed via the Wechsler Memory Scale; WMS-III) and seizure onset zone (SOZ). While the IQ scores were within the normal range, the patients in our study showed a lower-than-average MQ. Although our paradigm did not directly probe the specific memory functions assessed by the WMS, this limitation is inherent to intracranial EEG recordings in epilepsy patients and should be acknowledged. While we cannot entirely rule out the influence of memory impairments on our results, the simplicity of our task design—with only three cues and a single instruction for all trials—, and the fact that learning curves align with expected behavioral patterns, partially alleviates this concern. Additionally, fear associations can be formed in the absence of conscious awareness of the CS-US contingencies (Knight et al., 2003, 2009), suggesting that explicit memory impairments may not necessarily prevent the formation and extinction of fear associations. Indeed, our trial-by-trial progressive learning analysis confirms that patients successfully understood the task instructions and appropriately learned the CS-US contingencies, adaptatively adjusting their responses to contingency changes in CS+ trials (see Figure 1D).”

Participant	Gender	Age	Global IQ	Seizure Onset Zone	MQ
DBX	M	22	115	Right temporal lobe, bilateral medial temporal lobes, insula, and right extra-orbital cortex	106
LSY	F	16	88	Right medial temporal lobe	39
JJL	F	25	86	Left temporo-occipital lobe	73
HWL	M	19	104	Left frontal lobe	80
ZHX	M	31	87	Left medial temporal lobe	<51
WXY	M	24	108	Left temporal lobe	75
LQM	F	23	96	Right temporal insular cortex	81
CLF	F	21	94	Bilateral medial temporo-occipital junction regions	52
HLL	F	26	95	Right temporo-occipital junction region	70
DHZ	F	16	83	Right limbic system	77
CYY	M	35	83	Right temporal lobe	51
HLY	M	26	107	Bilateral limbic system	95
YJH	M	22	92	Right temporo-insular region	63
GKS	M	30	88	Bilateral medial temporal lobes, right hippocampus, and right insula	<51
ZH	M	21	98	Left temporo-insular lobe, orbital frontal gyrus, precentral sulcus, and parietal lobe.	81
KYP	M	27	98	Medial temporal lobe combined with insula, opercula, and orbitofrontal region	73
LYT	M	18	91	Right temporo-parietal junction region	89
LC	M	36	93	Left posterior temporal lobe	84
ZJJ	M	21	93	Right hippocampus, right insula, right central operculum	108
ZXF	M	32	124	Bilateral parieto-occipital regions	120
ZJH	M	27	84	Right posterior temporal region and temporo-occipital junction cortex	52

LHD	M	33	-	Left temporo-insular lobe	-
C	M	23	82	Right anterior temporal lobe	66
YLL	F	33	75	Right limbic system	68
WTT	F	24	-	Right temporal insular cortex	-
MAY	M	40	-	Right insula, right posterior orbital region and right medial temporal lobe	-
LHM	M	25	97	Left medial temporal lobe and insula	54
LSW	M	21	82	Left prefrontal lobe	55
CCL	M	22	-	Left temporo-insular lobe	-
LDN	F	21	109	Left temporal lobe	103
pat_02651_1127	F	27	-	Left temporal lobe	-
pat_02660_1136	F	34	-	Bilateral temporo-mesial	-
pat_02680_1158	F	52	-	Left temporal lobe	-
pat_02689_1168	F	31	75	Right temporo-parietal	-
pat_02711_1193	F	18	118	Right temporal lobe	-
pat_02718_1201	F	26	-	Left temporal lobe	-
pat_02985_1405	M	31	-	undetermined focus	-
pat_03012_1444	M	26	-	Temporo-polar	-
pat_03046_1482	F	39	93	Right temporal	-
pat_03083_1527	M	25	101	Bilateral temporo-basal	-
pat_03092_1538	F	35	81	Bilateral temporo-mesial	-
pat_03105_1551	F	32	-	Bilateral temporo-mesial	-
pat_03128_1591	F	28	99	Bilateral temporo-mesial	-
pat_03138_1601	F	43	-	Left temporo-mesial	-
pat_03146_1608	M	37	88	Bilateral temporo-mesial	-
pat_03174_1634	M	25	-	Polar and baso-medial region of the right temporal lobe	-
pat_03249_1711	F	52	-	Left temporo-mesial	-
pat_03266_1729	M	39	-	Bilateral temporo-mesial	-
pat_03300_1766	F	25	-	Right temporo-mesial	-

Supplementary Table 1. Patient demographic and clinical information. Details about patients' gender, age, Intelligence Quotient (IQ), Seizure Onset Zone (SOZ), and Memory Quotient (MQ) are provided.

2. This study notes that the co-registration methods used in Paris and Guangzhou were different. It would be useful to explain whether this discrepancy is due to differences in institutional protocols or technical limitations. It would strengthen the study to demonstrate that the co-registration method does not affect the results. For example, showing that co-registration of Paris data using the Guangzhou method (or vice versa) does not lead to changes in labeling would help underscore the consistency of the results.

The extraction of electrode coordinates was performed by two independent clinical teams at the respective hospitals where the experiments took place. This approach is standard practice in multi-center iEEG studies (e.g., see Dimakopoulos et al., 2022). While the methods employed by each team differ in certain aspects, they largely converge on the most critical steps (see details below). Our decision to use both approaches was not due to any technical limitation preventing electrode processing in a single center, but rather practical considerations: electrode coordinate extraction and labeling are routine medical procedures in both hospitals. In addition, extracting electrode information locally avoids the transfer of Magnetic Resonance Images (MRIs) across countries and thereby reduces data protection concerns.

Notably, the team in Paris has developed an in-house software, EPILOC (Lehongre et al., 2022), to automate certain aspects of co-registration. The team in Guangzhou developed a custom-written pipeline to extract the electrodes using standard toolboxes for this purpose. The methodologies employed at both sites are largely similar. Specifically, both teams employ co-registration of preoperative MRI and postoperative CT scans, generate patient-specific anatomical brain surfaces using FreeSurfer (Fischl, 2012), normalize these surfaces into the Montreal Neurological Institute (MNI) coordinates using Statistical Parametric Mapping (SPM), and use 3D Slicer (Pieper et al., 2004) for visualization and electrode localization. Thus, both methods fundamentally follow the same steps and rely on the same third-party software. We acknowledge that this was not sufficiently emphasized in the original manuscript, and have revised the text in the Methods section to emphasize similarities and differences between the two approaches. In lines (850-872), we now write:

“Electrode localization was conducted separately by our clinical teams in Paris and Guangzhou using very similar methodological approaches. At both sites, post-implantation computed tomography (CT) images were coregistered onto preimplantation magnetic resonance imaging (MRI) data. Both sites used the same software including Freesurfer (<http://surfer.nmr.mgh.harvard.edu/>), 3DSlicer (<https://www.slicer.org/>), and Statistical Parametric Mapping (SPM; <https://www.fil.ion.ucl.ac.uk/spm/>).

In Paris, electrode coordinates were extracted using the EPILOC toolbox developed on the STIM platform (Stereotaxy, Techniques, Images, Models; <http://pf-stim.cricm.upmc.fr>). EPILOC automates several image processing steps using Freesurfer, 3DSlicer, and SPM,

complemented by Brainvisa (Rivière et al., 2011). Anatomical models were generated from preoperative MRI data using Freesurfer. Post-operative CT data within the patient's native space were coregistered onto the preoperative MRI data and then normalized to a Montreal Neurological Institute (MNI) template using SPM. Depth sEEG electrodes were automatically localized in postoperative CT images by segmenting electrode artifacts and classifying them based on their proximity to trajectories planned by stereotactic guidance devices.

In Guangzhou, channel locations were again identified by coregistering the post-implantation CT images to pre-implantation MRIs acquired for each patient. Patient specific anatomical surfaces were generated with Freesurfer and normalized to MNI space using SPM. Electrode locations were determined using 3DSlicer.

A table containing the electrode coordinates in MNI space for all contacts included in the study is presented in Supplementary Table 2”.

We recognize the importance of verifying that the different co-registration approaches do not introduce significant discrepancies in the results. As suggested by the reviewer, we requested our clinical team in Paris to perform the co-registration and labeling of electrode locations for a randomly chosen subset of the data collected in Guangzhou. The results indicate that the labeling remains consistent for the large majority of electrodes, confirming the reliability of our approach.

Additionally, we note that many of our analysis relied on relatively large brain areas. For example, in the TMP, electrodes located in the inferior, superior and middle temporal gyri—covering most of the temporal lobe—were included. This reduces the risk of obtaining substantially different results with the two methods for electrode localization (see also our response to Criterion 6, *Suggested Improvements*, Point 5).

3. The experimental paradigm in this study references only Dunsmoor et al., 2014, which may not sufficiently support the design. Moreover, there appear to be notable differences between the two paradigms, such as the time intervals between the acquisition, extinction, and test phases. (For instance, Dunsmoor's paradigm was conducted over two days.) Could you clarify whether this experimental paradigm is identical to or a slight modification of those used in prior studies? Additionally, has this paradigm been validated in previously published research? If not, could you provide further justification for the validity of the design?

We appreciate the opportunity to provide a more detailed description of our paradigm and its validity—as we acknowledge that this was not sufficiently addressed in the original manuscript. We first note that in the Methods section (line 556), we did not mean to suggest that our paradigm is directly based on the paradigm in Dunsmoor et al., 2014.

Instead, we cited their study to highlight that our choice of applying a 50% partial reinforcement rate —where CS+ items are paired with the US in 50% of trials—, was informed by their approach. We have now revised the text (lines 767-770) to clarify this distinction.

Importantly, while our experimental design is novel and its unique combination of features has not been validated in previous studies, each of the design decisions we made was based on practical and theoretical considerations derived from the fear conditioning literature. Below, we provide a rationale for each of these decisions.

First, we employed a multi-cue conditioning protocol in which one stimulus (CS+) predicts the unconditioned stimulus (US), while another (CS-) does not. This approach is the most commonly employed in human fear conditioning studies and differs from paradigms in which a single cue is first associated with a US and later extinguished (Lonsdorf et al., 2017).

Second, like previous research investigating the representational dynamics of fear learning (Visser et al., 2013), we used natural images as conditioned stimuli rather than auditory stimuli—more employed in animal research—or simplified shapes (S. Chen et al., 2021; Neumann et al., 2007; Schmitz & Grillon, 2012). Our approach is not only arguably more ecologically valid but also better suited to capture cue representations across several sensory areas including association cortices.

Third, our study follows an ABC design, in which acquisition occurs in one context, extinction in another, and testing in a novel third context. This structure is commonly employed in both rodent and human studies that examine the role of context in fear extinction (Balooch et al., 2012; Corcoran & Maren, 2001, 2004; Effting & Kindt, 2007; Hermann et al., 2016; Milad et al., 2005). However, it differs from other human studies that have used ABA (Krisch et al., 2018) or ABB (Balooch et al., 2012) paradigms to investigate context-dependent fear extinction. We employed an ABC paradigm because we aimed to investigate whether the re-occurrence of fear memories during the final test phase was determined by the similarities of internal (i.e., neural) representations of contexts during this phase to context representations during either the acquisition or the extinction phase. This differs from ABA and ABB paradigms in which the presented contexts during the test phase are either identical to the contexts during acquisition (ABA) or during extinction (ABB).

Notably, while our paradigm follows an ABC structure, our design includes the presentation of several distinct yet thematically related context videos in each experimental phase (e.g., videos of sea landscapes). This was done to facilitate the analysis of context-specific representations using Representational Similarity Analysis (RSA), which requires the presentation of multiple instances of a particular context. In particular, as stated in our hypothesis, we aimed to evaluate the generalization of context

representations during acquisition and their distinctiveness during extinction (see also our response to reviewer 3 point 5).

Fourth, we applied a partial reinforcement rate of 50% (Dunsmoor et al., 2007, 2012, 2014; Visser et al., 2013). While some studies have used slightly higher rates, such as 60% (Battaglia et al., 2018; Milad et al., 2007), 62.5% (Hermann et al., 2016), or 66% (Dunsmoor & Murphy, 2014), we opted for 50% to promote a more gradual acquisition and extinction of fear associations (note that both processes are affected by this choice). Slower rates also reduce the risk of habituation to the unconditioned stimulus (US), which was particularly important in our study given the relatively mild US we employed (see next point).

Fifth, in our paradigm, the US consisted of a human female face with a neutral expression following CS- items and a fearful expression following (50% of) CS+ items. Presentation of the fearful face following CS+ items was, in addition, paired with a loud scream. The choice of using emotional facial expressions was informed by studies suggesting that negative facial expressions increase resistance to extinction compared to neutral faces (Mineka & Öhman, 2002; Ney et al., 2022; Öhman & Öst, 1985). Similarly, pairing this face with a scream was grounded on previous fear conditioning research in sensitive populations such as infants and patients (Glenn, Klein, et al., 2012; Hamm et al., 1989). Moreover, the face-scream protocol has been shown to produce similar—though milder—responses in fear-potentiated startle and self-reported fear ratings in healthy participants as compared to electric shocks (Glenn, Lieberman, et al., 2012).

Sixth, to strengthen the association between the CS and US, we incorporated a narrative element, or "cognitive embedding". Previous research has shown that semantically or perceptually meaningful CS-US pairings are learned more effectively and are more resistant to extinction than arbitrary associations (Garcia & Koelling, 1966; Hamm et al., 1989). In our paradigm, we introduced the character of Nina, a backpacker who travels the world and stays in hotels where electric devices occasionally malfunction. Pictures of these electric devices were used as our CS (a fan, a hairdryer and a toaster). The negative emotional expression of "Nina's" face, paired with the scream, aligns with this narrative. We recognize that this aspect of our paradigm was not sufficiently explained in the previous version of our manuscript and thank the reviewer for the opportunity to provide a more detailed description.

Seventh, we used videos to represent contextual information rather than simpler stimuli such as colored backgrounds (Kalisch et al., 2006), images of scenes or rooms (Battaglia et al., 2018; Xia et al., 2023), or more complex VR environments (Baas et al., 2004; Glotzbach-Schoon et al., 2013). This was made to enhance realism and reinforce the narrative of a traveler visiting diverse regions with distinct landscapes and climates while avoiding the difficulties of conducting VR research in clinical populations. In addition, we

used videos since previous research has shown that their neural representations can be particularly well tracked in human electrophysiological data (Michelmann et al., 2016).

Eighth, as in various human fear conditioning studies that rely on self-reports rather than physiological measures, we assessed participants' expectancy ratings of safety and threat rather than collecting electrodermal activity data (Bandarian Balooch & Neumann, 2011; Boddez et al., 2013). This was guided by both ethical and practical considerations, particularly given the constraints of conducting research with implanted epilepsy patients, and to emphasize the role of conscious awareness in learning and extinction (see also our response to the second point of Reviewer 1).

Ninth, our testing phase occurred shortly after extinction without introducing additional delays. This decision was driven by the limited availability of patients in the clinical context. We acknowledge that some studies have implemented delays between extinction and testing to evaluate the stability of extinction in the long-term (Milad et al., 2005; Xia et al., 2023), but also highlight that several others have applied immediate testing (e.g., (Vansteenwegen et al., 2007; Visser et al., 2013).

Finally, due to the time constraints of conducting research with implanted patients, we did not conduct individualized aversiveness testing for the US. Instead, as in previous studies (Schmitz & Grillon, 2012; Wehrli et al., 2022), we selected a notoriously aversive stimulus—a loud scream—and adjusted its duration and intensity to a highly unpleasant level (1s, ~85dB).

The unique and novel combination of features in our paradigm—including the use of multiple, thematically related videos as contextual stimuli, the ABC structure, the 50% reinforcement rate, and the embedding of the experiment into a plausible narrative—has not been validated in previous research. However, as we have discussed, these methodological choices were carefully informed by the fear conditioning literature in humans. Notably, a very similar version of this paradigm, developed by our group for an fMRI study, includes the same stimuli, a very similar overall structure, and the same cover story but was conducted in healthy participants and employed a different US (electric shocks). This study has been recently published in bioRxiv and is currently under review (Bouyeure et al., 2024). While these data are beyond the current study, they provide an additional validation of our core paradigm choices.

We have added this comprehensive description in the new Supplementary Note 1. In addition, we included the following text in the Methods section (lines 787-792) to refer to it:

“Please note that our experimental design is grounded in theoretical considerations from the human fear learning literature and has been adapted to meet the specific requirements of conducting research in implanted epilepsy patients. While its specific

combination of features is novel and has not been fully validated, we provide a detailed rationale for each of our design choices, highlighting their similarities and differences to previous research in the field, in Supplementary Note 4”.

4. In Figures 2 and 3, does the CS- during the extinction phase include stimuli that were designated as either CS+ or CS- during the acquisition phase? In other words, does it encompass both CS+- and CS-- stimuli? I believe that CS+- reflects the extinction of fear learning, whereas CS-- serves as a safety signal that was not associated with fear learning. Could you clarify whether the results in the extinction phase distinguish between CS+- and CS--? If not, I would suggest separating these two categories in the results for clarity.

In Figures 2A and 3, the data for the CS- condition during extinction indeed includes both CS+- and CS-- trials (i.e., CS- trials that were either CS+ or CS- during acquisition). This was done to focus on the *current* valence of the CS during extinction—which we assumed would be the primary factor influencing theta power and item stability—and to maximize statistical power by including all available trials in the CS- condition. We have now clarified this in the Methods section of the revised manuscript (lines 951-957).

We acknowledge the importance of analyzing our three trial types separately (CS++, CS+-, and CS--). We note that in the original manuscript we provided the data of these different conditions in Figure 2B (bottom panel), focusing on the time-frequency cluster where we observed significant differences in the contrast including all trials (Figure 2B, top panel). In the revised version, we have modified Figure 2B to include the contrasts of CS++ vs. CS+-, CS++ vs. CS-- and CS+- vs. CS-- across all time frequency points during extinction. The results previously presented in Figure 2B are now presented in Figure 2C (see below the updated Figure 2).

We observed a very similar cluster of increased theta power for the two types of “current” CS- trials (i.e., CS+- and CS--) when compared to CS++ trials during extinction (CS+- vs. CS++: $p = 0.005$; CS++ vs. CS--: $p = 0.026$), while the CS+- vs. CS-- analysis revealed non-significant differences ($p = 0.38$). We included these results in the revised manuscript, results section (lines 213-217). Thus, theta power in amygdala follows current CS valence during extinction regardless of previous valence during acquisition.

We performed the same contrasts as in the power analysis in the item stability analysis (CS+- vs. CS++, CS-- vs. CS++, and CS+- vs. CS--; Figure 3). We specifically focused on the AMY and the TMP, in the time periods where we observed significant effects (AMY: 1.25-1.5s; TMP: 0.65-1s). We observed a qualitatively similar pattern of results in AMY and TMP. In the AMY, CS++ trials had significantly higher item stability than both CS+- ($t(31) = 2.30$; $p = 0.028$) and CS-- trials ($t(31) = 2.39$; $p = 0.023$), exactly matching the

pattern of results observed in the AMY theta power analyses. In the TMP, we observed higher item stability for CS++ vs CS+- trials ($t(40) = 2.46$, $p = 0.018$), while the difference between CS++ vs CS-- trials did not reach significance (even though it numerically went into the same direction; $t(40) = 1.64$, $p = 0.11$). The CS+- versus CS-- contrast also did not reveal significant differences in the TMP ($t(40) = 0.52$, $p = 0.61$). These results were included in the revised Figure 3 (panels B and C, see below), and we also describe them in the results section of the revised manuscript (lines 246-261).

As the Reviewer points out, the CS+- trials are those that require extinction of previous fear, while the CS-- trials convey a safety signal during both acquisition and extinction. However, following successful extinction learning of the CS+- stimuli, their valence matches the valence of the CS-- stimuli, and thus the valence representations of these two stimulus types should be identical. Our findings both regarding theta power and item stability in AMY showed a prominent effect of current valence (i.e., CS+ trials differed from both CS+- and CS-- trials), but no effect of previous valence (i.e., no difference between CS+- and CS-- trials). In the TMP, the results are qualitatively similar but not identical: while CS++ trials showed higher item stability than CS+- trials, they were not significantly higher than CS--. On the other hand, CS+- were also not different from CS--, and the more prominent differences were observed in the CS++ vs CS-- contrast, which followed the same direction of the CS++ vs CS+- contrast.

Taken together, these results suggest that theta power and item stability levels in both AMY and TMP reflect the fact that participants could successfully extinguish the previous valence representations of the CS+- items, because these previously threatening cues are now processed similar to previously safe stimuli.

Nevertheless, the differences in context representations during acquisition and extinction (with more generalized context representations during acquisition and more distinct context representations during extinction; see Figure 6) facilitate the subsequent reinstatement of fear memories, indicating that these fear memories are not entirely “unlearned” during extinction but only suppressed. We have added this discussion in the revised manuscript (lines 467-472).

“In our paradigm, both CS+- and CS-- trials signal safety during extinction – either newly learned safety following extinction (for CS+- trials) or consistent safety throughout the experiment (for CS-- trials). Thus, the successful extinction of previous fear should result in neural activity and representations of CS+- items that match those of CS-- items—in line with our findings on both AMY theta power and AMY and TMP item stability.”

Figure 2. Higher theta power in the amygdala for CS- as compared to CS+ trials during extinction

A) Amygdala 1-12Hz power. During acquisition (left), no significant differences in low frequency power were observed between CS+ and CS- trials. Bottom row shows the result of the contrast between the two conditions, while top and middle rows reflect levels of Z-scored power for each condition. During extinction (right), oscillatory power was significantly higher for CS- trials in a late time period of cue acquisition, i.e., from 1.18 to 1.75s in the 3-10Hz frequency range. Negative T-values depicted in blue reflect higher theta power for CS- items. B) T-maps of the CS++ vs C+- (top), CS++ vs CS-- (middle) and CS+- vs CS-- (bottom) contrasts. C) Top: CS+ vs CS- contrast in the frequency spectrum between 1-100Hz. Bottom: average theta power in the cluster observed in panel A, bottom right, and C, top, for all CS types during the phases of acquisition and extinction. In panels A, B and C, significant regions surviving correction for multiple comparisons using cluster-based permutation statistics are outlined in black in the time-frequency maps. *: $p < 0.05$; **: $p < 0.01$.

Figure 3. Item stability in lateral temporal cortex, amygdala, and across the extinction network

A) Item stability, defined as the average similarity of neural patterns representing individual items across repeated presentations, was assessed separately for CS+ and CS- trials during acquisition and extinction. B) Analysis of item stability in the AMY revealed no significant differences between CS+ and CS- trials during acquisition (left). During extinction (right), item stability was significantly higher for CS+ as compared to CS- items in a late time period during cue presentation (1.2-1.4s after cue onset). C) Item stability during extinction in the significant time period shown in panel B, right, for CS++ and CS+- (left panel), CS++ and CS-- (middle panel), and CS+- and CS-- trials (right panel). D) Analysis of item stability in the TMP revealed no significant differences between CS+ and CS- trials during acquisition (left). During extinction (right), item stability was significantly higher for CS+ as compared to CS- items in a time period from 0.65s-1s after cue onset. E) Item stability during extinction in the time period shown in panel D, right, for CS++ and CS+- (left), CS++ and CS-- (middle) and CS+- and CS-- (right) trials. F) Top left: A single-trial metric of item stability was computed by assessing the similarity of each trial across repeated presentations (average of the comparisons highlighted with a red rectangle in the RSA matrix). Bottom left: Fluctuations in item stability were coordinated across trials between the TMP and the AMY during the first second of cue presentation

and towards the end of the cue presentation period. Right: Item stability was computed in each ROI and correlated across trials with the values observed in TMP (upper row) and AMY (lower row). In panels B and D, black horizontal lines depict significant time periods after correction for multiple comparisons using cluster-based permutation statistics. Time zero indicates the onset of the CS, and shaded lines depict group average Rho values \pm S.E.M. Two-sided paired t-tests were applied at every time point (panels B and D) or every time-by-time point (panel F). In panel F, significant regions surviving multiple comparisons corrections using cluster-based permutation statistics are outlined in black. Time zero indicates the onset of the CS in both time axis. * $p < 0.05$; ** $p < 0.01$; *** $p < 0.001$. AMY: Amygdala; HPC: Hippocampus; IPFC: Lateral Prefrontal Cortex; OFC: Orbitofrontal Cortex.

5. Regarding the CS++, I would expect the cumulative fear learning associated with this stimulus to result in very low amygdala (AMY) theta power, as it contrasts with the safety signal. However, in Figure 2, the CS+ (which presumably corresponds to the CS++ from the acquisition phase) appears to exhibit relatively high theta power. Could you clarify this apparent inconsistency?

We agree that it may appear surprising, given the pattern of theta power results we observe, that the CS+ trials during extinction—which indeed correspond to CS++ items—appear to exhibit higher theta power than CS- trials during certain time periods of cue presentation (Figure 2A). This happens in particular at the middle of the trial, from ~300-500ms after stimulus onset and at around ~4-5Hz. However, we note that most of these numerical increases are not statistically significant even before correcting for multiple comparisons, and they do not survive correction for multiple comparisons. Specifically, only a small number of time-frequency bins show significantly higher theta power for CS+ compared to CS- trials at the uncorrected level, occurring at a very specific individual frequency (4Hz) and for a brief period of time (80ms, from 330 to 410ms after cue onset; cluster indicated with a red arrow in the Figure below, bottom panel). The highly transient and spectrally narrow nature of this cluster suggests it is driven by random fluctuations.

In detail, our time-frequency analyses involved conducting a t-test at each time-frequency bin, resulting in 12 (frequencies) x 175 (time points) = 2,100 tests in each contrast. Given the standard 5% false positive rate, a proportion of these tests are expected to yield significant results solely by chance. Cluster-based permutation correction accounts for this by preserving only clusters whose summed t-values exceed the 95th percentile of a null distribution constructed by shuffling the trial labels (Maris & Oostenveld, 2007). In our data, only the cluster at the end of the period (1.18-1.75s) survived this stringent correction (Figure 2A). The observed 4Hz cluster ranks 523rd out of 1,000 in this distribution ($p = 0.478$), confirming that it does not represent a significant effect.

We have explicitly addressed this apparent inconsistency in the Discussion section of the revised manuscript, reinforcing that the observed numerical increases in theta power for CS++ trials are unlikely to reflect any meaningful cognitive process. In lines (484-490), we write:

“We note that in our AMY theta power analyses, some time-frequency bins observed early in the trial (~300-500ms) showed numerical increases in theta power for CS++ items at an uncorrected level. This might seem surprising, as it contrasts with the main effect observed later in the trial (1.18-1.75s) which reflects a decrease in theta power for threatening items. However, these numerical increases are within the range of what is expected by chance, as assessed by our cluster-based permutation analysis”.

Uncorrected results in the AMY theta power analysis during extinction. Red arrow in the bottom panel indicates a small cluster of time-frequency bins showing significantly higher theta power for CS+ as compared to CS- trials at an uncorrected level. This transient and spectrally narrow cluster, however, does not survive correction for multiple comparisons using cluster-based permutation statistics (Maris & Oostenveld, 2007).

Criterion 2 – Preregistration

Comments:

I have no comments.

Criterion 3 - Appropriate use of statistics and treatment of uncertainties

Comments:

The statistical methodology appears to have been appropriately applied; however, there seems to be a lack of discussion regarding the statistical uncertainty associated with the number of trials. How was the minimum number of trials, set at 8, determined?

Additionally, a more detailed explanation of how varying the number of trials (e.g., fewer or more than 8) might influence the results would be valuable. Furthermore, in Figure 1D, were there significant differences in behavioral performance among the three conditions during the test phase? The observed difference between CS++ and CS-- in the test phase seems similar to that observed during the extinction phase. Was the sample size (n) used for statistical analysis sufficient for each condition? Lastly, was normality assessed before conducting the paired t-tests?

Please see below for specific answers.

Number of trials

Due to the challenges of accessing patient populations, iEEG studies are typically shorter than those conducted with other methodologies such as fMRI or EEG and rely on fewer trials. Additionally, epileptic activity can further reduce the number of usable trials. On the other hand, iEEG has a higher signal-to-noise ratio compared to other techniques, supporting the validity of using fewer trials (Axmacher, 2023; Parvizi & Kastner, 2018).

Since population-level analyses are essential in many cognitive experiments, including ours, we balanced the number of trials with the total number of participants. Previous research has indicated that hippocampal theta power analyses in iEEG should not be conducted with less than five trials (Oehr et al., 2015, see Figure Below). Analyses in this study showed that results between 5 and 8 trials often see a sharp decrease in variability, suggesting that >5 trials are required and 8 trials are appropriate as a minimum. For this reason, we set the minimum number of trials to 8 in all our analyses, as eight provided a reasonable trade-off between number of trials and number of subjects in our population level analyses.

Graphs depicting the variance of power estimates as a function of trial numbers in a previous study (Supplementary Figure 11 in Oehr et al., 2015). The plots show data for two subjects with the largest trial

count. Asterisks indicate significant differences of one trial bin to the bin with the maximum number of trials: * $p < 0.05$, ** $p < 0.01$, *** $p < 0.001$.

We agree with the reviewer on the importance of assessing how this decision impacts our results. To address this, we have now conducted additional analyses to illustrate how our findings change when setting the minimum number of trials to either 5 or 10 trials. Results are presented in the Table below.

N trials \ Analysis	5	8	10
AMY Power	<0.001 (N=32)	<0.001 (N=32)	<0.001 (N=31)
AMY Item stability	0.018 (N=32)	0.018 (N=32)	0.018 (N=32)
TMP Item stability	<0.001 (N=43)	<0.001 (N=43)	0.075 (N=42)
IPFC context specificity	0.012 (N=22)	0.012 (N=22)	0.012 (N=22)
AMY power – AMY STA	3.07e-10 (N=32)	3.07e-10 (N=32)	3.07e-10 (N=32)
IPFC context specificity - reinstatement correlation analysis (CS+- trials)	0.174 (N = 18)	0.024 (N = 16)	0.0407 (N=15)
TMP behavior (CS+- trials)	0.008 (N = 21)	0.032 (N = 16)	0.076 (N=15)

We note that for most of our analyses, changing the minimum number of trials did not affect the results. In particular, in the power analysis in the AMY (Figure 2), the IPFC context-specificity analysis (Figure 4), and the AMY analysis correlating power and item stability (Figure 5), results were not affected by the choice of different thresholds.

In the item stability analyses (Figure 2), the different number of trials did not affect the results in the AMY. In the TMP, increasing the minimum number of trials to 10 led to the exclusion of one subject, resulting in very similar trend-level results. Similarly, in the analysis correlating ratings and reinstatement in the TMP (Figure 7), setting the minimum number of trials to 10 excluded one subject but still produced trend-level results.

In the IPFC context specificity-reinstatement correlation analysis (Figure 6), the results were not significant with five trials but remained significant when the minimum number of trials was set to 10. Notably, the analysis with five trials involved the inclusion of two subjects whose data quality was in general the lowest—as these subjects were those in which more trials had to be removed. We propose maintaining the results with a minimum

number of eight trials, as this provides a more robust estimate of the different metrics in each subject. Importantly, the results with eight trials remain highly consistent with those using a minimum of 10 trials, showing equivalent significance or trend-level effects across all contrasts.

Behavioral results during the test phase

In our original two-way ANOVA model reported in the manuscript (lines 165-170), we observed a significant interaction of cue type and experimental phase ($F(4, 188) = 6.96$; $p = 3.02e-05$). We agree with the Reviewer's observation that the differences between CS++ and CS-- trials during the test phase appear similar to those observed in the extinction phase. To assess this, we conducted a one-way ANOVA with cue type as a factor in each experimental phase. A significant main effect was found in all phases (Acquisition: $F(2,45) = 20.18$, $p = 5.12e-08$; Extinction: $F(2,45) = 28.38$, $p = 2.27e-10$; Test: $F(2,45) = 6.12$, $p = 0.0032$). Post hoc comparisons in the test phase revealed significant differences between CS++ and CS+- trials ($t(47) = 2.51$, $p = 0.015$) and between CS++ and CS-- trials ($t(47) = 3.1$, $p = 0.003$), whereas no significant difference was observed between CS+- and CS-- trials ($t(47) = 1.13$, $p = 0.26$).

On the other hand, we did not observe significant differences in our trial-by-trial progressive learning analysis during the test phase after correcting for multiple comparisons (Figure 1D). We note that at the uncorrected level, some trials in the test period showed significant differences. However, the application of cluster-based permutation statistics eliminates any sequence of contiguous significant trials whose summed t-values fall below the 95th percentile of the null hypothesis distribution, which is constructed by shuffling trial labels.

Together, these results indicate that while participants exhibited reduced fear during the test—reflected in smaller differences in ratings between CS+ and CS- trials and the absence of significant effects in the trial-by-trial progressive learning analysis—they still rated CS++ items on average as more threatening than CS-- or CS+- items.

These findings suggest that participants did not completely unlearn the CS+ associations formed during acquisition and extinction. Despite the absence of a US, fear renewal occurred in some trials during the test, indicating a partial re-emergence of fear memories in new contexts, in line with our analysis on the reinstatement of fear memories (Figure 6).

We have included these results in the *Behavioral results* section of our manuscript (lines 188-195).

Was the sample size (n) used for statistical analysis sufficient for each condition?

In both the progressive learning analysis and the phase-specific behavioral analyses, our sample size included 48 subjects. This number exceeds that of most previous iEEG studies and is large enough to be comparable with previous behavioral studies. Notably, since we employed a within-subjects design, all participants were exposed to all experimental conditions, which increases the statistical power of our comparisons. To corroborate that the number of subjects included in our analyses was appropriate to assess the effects of interest, we assessed the effect size of the main ANOVA analysis by computing the partial eta squared of each of our factors and the interaction. Partial eta squared quantifies how much of the total variance in the dependent variable is explained by each of the independent factors and the interaction. We observed that both the main effect of trial type and the main effect of experimental phase had large effect sizes (Trial type: $\eta^2 = 0.29$; Experimental phase: $\eta^2 = 0.16$), while the interaction had a medium to large effect size ($\eta^2 = 0.129$).

Lastly, was normality assessed before conducting the paired t-tests?

We did not assess normality before conducting the paired t-tests. Instead, we applied non-parametric multiple comparisons correction using cluster-based permutation statistics (Maris & Oostenveld, 2007). In this approach, which we adopted from the analysis of time-frequency data, a null hypothesis distribution is generated by shuffling the condition labels 1,000 times. Summed t-values corresponding to clusters of contiguous trials showing significant differences between conditions in the observed data are compared to summed t-values in clusters formed under the assumption of the null hypothesis. The “null hypothesis distribution” is built by contrasting the shuffled conditions and storing the cluster with the highest summed t-value at each permutation. Only “large-enough” clusters in the real data, i.e., above the 95th percentile of those observed in the distribution constructed with shuffled data are considered significant. Thus, while a t-test is applied at each time-frequency bin, false positives are controlled for at the level of multiple comparisons correction in a non-parametric way, i.e., without relying on any assumption of normality.

We note that in our progressive learning analysis, since responses in each trial were reported on a scale from 1 (Threatening) to 4 (Safe), the population distributions at each trial were not normally distributed. We confirmed this by applying a Shapiro–Wilk normality test at each trial: none of the population distributions were normal—in any condition or trial.

We acknowledge the reviewer’s concern regarding the normality of the data. We thus performed an equivalent analysis using the Wilcoxon signed-rank test, the non-parametric

equivalent of the paired t-test to corroborate our results. We employed the standardized Wilcoxon signed-rank statistic (Z) as our metric of the strength of the effect of each test, and summed the Z value in all contiguous significant clusters. In addition, we assessed the validity of the approach by considering the size of the corresponding clusters instead of their summed Z-values, i.e., the length of the contiguous periods of trials showing a significant effect. We only considered significant sequences of consecutive significant trials whose summed Z value (or length) was above the 95th percentile of the null distribution computed by shuffling the condition labels 1,000 times.

Our results in both tests were identical to those observed in our original analysis. Specifically, the trials where conditions differed at the uncorrected level were exactly the same (see figure below), and the clusters that survived multiple comparisons correction were also identical, which resulted in the same p-values for the relevant clusters (all $p < 0.001$ in all three clusters; CS++ vs CS+-; CS++ vs CS-- and CS+- vs CS--).

We note that the approach followed in the progressive learning analysis is different from the two-way ANOVA where an average across trials in each experimental phase was taken within each participant before performing the group-level statistics (lines 165-170). In the ANOVA, approximately half of the conditions were normally distributed (Acquisition: CS++ and CS+-; Extinction: CS++ and CS+-; Test: none). Given that the non-parametric equivalent of a two-way repeated measures ANOVA is not straightforward to compute, we decided to stick with the parametric ANOVA for this analysis. However, when we implemented two one-way Friedman tests (the non-parametric equivalent of a one-way repeated measures ANOVA) for CS type and Experimental phase, we observed very consistent results: Trial type: Acquisition: $p = 1.28e-06$; Extinction: $p = 1.32e-09$; Test: $p = 0.016$; Experimental Phase: CS++: $p = 0.41$; CS+-: $p = 2.69177e-05$; CS--: $p = 0.0041$. These results confirm that the CS means are different within all experimental phases. They also reveal a dissociation between CS++ items, which are not significantly different across phases, and CS+- and CS-- items, which are significantly different across experimental phases. We note that the increases in safety ratings for the CS-- trials during the extinction phase and relative lower (more threatening) ratings during the acquisition and test phases might account for this difference.

We have included the non-parametric version of the progressive learning analysis in the revised manuscript (see changes in lines 803-819, *Methods* section, and 174-188, *Behavioral results* section).

Comparison of the progressive learning analysis conducted with parametric (left) and non-parametric (right) approaches. In the parametric method, group-level t-tests were applied at every trial, while in the non-parametric equivalent, i.e., the Wilcoxon signed rank test was conducted. Notably, both approaches provided identical results at the uncorrected level (top row), and also after correction for multiple comparisons using cluster-based permutation statistics (bottom row).

Criterion 4 - Custom code

Comments:

I have no comments.

Criterion 5 – Conclusions

Comments:

I have no comments.

Criterion 6 - Suggested improvements

Comments:

1. The process of distinguishing and responding to threats and safety cues occurs rapidly in humans, as quick threat adaptation or escape is crucial for survival. In this context, it may be valuable to consider that, in addition to slower-cycle theta oscillations, faster-cycle neural activity could also play a role in safety responses. While the current manuscript focuses on theta oscillations, addressing this point in the discussion could enhance the

interpretation of the results. Specifically, a brief mention of how faster neural oscillations might contribute to safety-related processes would provide a more comprehensive understanding of the neurophysiological mechanisms involved.

We thank the reviewer for this suggestion. We now include the following paragraph in the discussion to address this point (lines 490-508):

“While our primary analyses focused on theta frequency oscillations, humans are able to rapidly distinguish and respond to threats and safety cues (Méndez-Bértolo et al., 2016). This raises the possibility of a potential role of high-frequency neural activity in signaling threat and safety. Indeed, theoretical accounts suggest that fast cycle gamma oscillations might promote fast information transfer between brain regions (Fell & Axmacher, 2011), facilitating fear learning by modulating the timing of neuronal firing inducing plasticity (Cattani et al., 2024). In particular, gamma cycles align with several biophysical factors regulating excitatory input integration in basolateral amygdala principal neurons (Bocchio et al., 2017), and experimental studies in rodents have shown that amygdala gamma oscillations contribute to fear memory formation through theta-gamma coupling and spike-field coherence (Popescu et al., 2009; Stujenske et al., 2014). While we did not observe any effect of gamma frequency oscillations in our paradigm, we believe further research will be needed to fully assess their possibly role in signaling threat and safety and modulating fear learning in humans—for example, via phase amplitude coupling (Saint Amour di Chanaz et al., 2023). Interestingly, recent findings suggest that hippocampal representational patterns linked to amygdala gamma bursts reinstate the content of emotional memories (Costa et al., 2024), providing evidence for a potential role in the representation of fear in the interaction with the hippocampus”.

2. The experimental design specifies a video presentation interval of 2 seconds: "At every trial, in each experimental phase, participants first saw a fixation cross for 0.5s, and then a context video which was presented for 2s" (lines 557–558). However, in Figure 4B, the x-axis appears to span a cue presentation interval of 1.75 seconds. Could you clarify this apparent discrepancy?

We thank the reviewer for pointing out this inconsistency. We have corrected all figures in panel B of Figure 4 to show the time courses until the end of video presentation (i.e., until 2s).

3. In line 374, should the reference be to Figure 6D rather than Figure 6B? The description appears to correspond more closely to Figure 6D.

Thanks for noticing. We have included the correct reference to figure 6D in the text.

4. The iEEG data were acquired at two different sites, one in Paris and the other in Guangzhou, using different electrode configurations. Given that bipolar referencing was employed, differences in electrode characteristics (e.g., surface area, interelectrode distance, etc.) could influence the volume of neural tissue sampled. However, I could not find an explanation of how these differences were reconciled during the analysis. Could you provide clarification on how this issue was addressed?

In our study, data were collected using three types of electrodes: two different types in Paris (micro-macro: 21 electrodes; macro-only: 176 electrodes) and one from Guangzhou (434 electrodes). These electrodes have variable surface area and inter-contact distances (Paris micro-macro: 1.6/3-7mm; Paris macro-only: 1.7/5mm; Guangzhou: 2/3.5mm). Additionally, within a single micro-macro electrode in Paris, the first and second contacts are separated by 3 mm, while subsequent contacts (e.g., 2 to 3, 3 to 4) are spaced 7 mm apart.

We acknowledge that differences in electrode configurations, combined with our use of a bipolar referencing scheme, might have affected the total volume of neural tissue sampled in our two patient cohorts. However, it is unlikely that these differences affected our main results for several reasons. First, we only included electrodes located in gray matter, which removed a large number of contact points within each electrode (~50%). The removal of many electrode contacts introduced variability in sampling density, affecting both Paris and Guangzhou cohorts similarly. Second, a higher sampling density does not necessarily equate to more information, as closely spaced electrodes in iEEG often capture activity from the same sources and exhibit similar neurophysiological profiles. We note that most of our analyses involved averaging electrode activity in each subject independently before statistical comparisons at the group level, which further reduces the risk that sampling density differences systematically biased our findings. Third, many of our key findings were observed in the AMY, which had the lowest sampling density overall, with a mean number of electrodes per subject of 2.56 ± 1.39 . Notably, 10 out of 22 patients had only a single electrode in the AMY, making it unlikely that inter-electrode distances or contact surface area significantly influenced our results in this region. Fourth, the inter-contact spacing of the first two electrodes in the micro-macro arrays from Paris (3 mm) closely resembles that of the electrode configurations used in Guangzhou (3.5 mm), minimizing potential sampling density differences in these electrodes.

We note that since the implantation scheme is solely determined by clinical criteria, density of sampling within and across brain regions is inherently variable in iEEG research. In particular, two main factors might affect the volume of neural tissue sampled beyond electrode configurations: (1) implantation scheme, i.e., location and number of electrodes implanted in particular brain regions and (2) ROI selection, i.e., how electrodes are grouped in functional regions. To corroborate that the volume of neural tissue sampled

did not differ between our two patient cohorts, we computed the Delaunay triangulation of electrode configurations. This method quantifies electrode density by forming triangles between contact points and calculating the mean triangle area, providing a sensitive measure of electrode density. Focusing on our largest and more densely implanted ROI (the TMP) we evaluated triangulation in the left and right hemispheres in each subject independently. Only subjects with at least three electrodes in the TMP (N=36) were included in this analysis, as a minimum of three electrodes is required to form triangles. Statistical significance was assessed using a two-way ANOVA with "Recording site" (Guangzhou/Paris) and "Hemisphere" as factors. The results showed no significant difference in mean triangle area between hemispheres ($F(1,32) = 2.61, p = 0.11$) or between patient cohorts ($F(1,32) = 0.96, p = 0.33$). These findings indicate that although electrode spacing varied due to differences in electrode configurations, the overall sampling density remained consistent across our two patient groups.

To further corroborate that the different electrode configurations did not affect our results, we conducted our oscillatory power analyses in the AMY separately for Guangzhou and Paris patients. We specifically focused on the significant cluster observed in the main analysis contrasting "current" valence (CS+ vs CS-) during extinction. We observed the same pattern of results across our two cohorts of patients, with higher theta power for CS- as compared to CS+ items during extinction (Guangzhou: $t(19) = -4.32, p = 0.0004$; Paris: $t(11) = 2.93, p = 0.014$, see Figure below).

Supplementary Figure 2. Theta power in the AMY during extinction was significantly higher for CS- as compared to CS+ trials in both the Guangzhou and the Paris patient cohorts.

Figure shows mean AMY power within the time-frequency cluster identified in the main analysis during extinction (Figure 2A, bottom left) for the Guangzhou (left) and the Paris (right) patient groups. In both cohorts, AMY theta power was significantly higher for CS- as compared to CS+ trials, consistent with the findings observed when combining the data of the two groups. These results demonstrate that our AMY theta power findings are robust and not influenced by methodological differences between the epilepsy centers where the data was collected (e.g., variations in electrode configurations or recording devices in Paris and Guangzhou).

Taken together, the results of these analyses demonstrate that our two patient cohorts did not show prominent differences in the neural tissue sampled in the TMP, and that the AMY theta power results during extinction were consistent in our two patient groups.

Finally, we highlight that recent years have seen an increase of multi-center iEEG research, which involves collecting and analyzing data from multiple epilepsy centers to generate larger and more diverse datasets. Several studies have successfully integrated multi-center iEEG data despite variations in electrode configurations, including even combination of subdural strips and grids with depth electrodes, which differ in regard to sampling density even more than different types of depth electrodes do (Bernabei et al., 2023; Dimakopoulos et al., 2022; Henin et al., 2021). While there is currently no established consensus or guideline on how to address sampling differences in the literature, we believe this is an important topic of research in the future, as multi-center iEEG studies are becoming more prominent.

We have included a discussion of these considerations in our revised manuscript, lines 872-879 (Methods section), which now reads:

“We note that since the implantation scheme is solely determined by clinical criteria, density of sampling within and across brain regions is inherently variable in iEEG research. Several factors might affect the volume of neural tissue sampled, e.g., different electrode configurations, individual implantation schemes, and the selection of ROIs. We performed several control analyses to corroborate that the volume of neural tissue sampled did not differ between our two patient cohorts (see Supplementary Note 5 and Supplementary Figure 2).

In addition, we have included the new Supplementary Figure 2 and Supplementary Note 5 to incorporate our additional analyses.

Supplementary Note 5: Different electrode configurations did not affect volume of neural tissue sampled and main results

In our study, data were collected using three types of electrodes: two different types in Paris (micro-macro: 21 electrodes; macro-only: 176 electrodes) and one from Guangzhou (434 electrodes). These electrodes have variable surface area and inter-contact distances (Paris micro-macro: 1.6/3-7mm; Paris macro-only: 1.7/5mm; Guangzhou: 2/3.5mm). Additionally, within a single micro-macro electrode in Paris, the first and second contacts are separated by 3 mm, while subsequent contacts (e.g., 2 to 3, 3 to 4) are spaced 7 mm apart.

Differences in electrode configurations, combined with our use of a bipolar referencing scheme, might have affected the total volume of neural tissue sampled in our two patient groups in Paris and Guangzhou. However, it is unlikely that such differences affected our

main results for several reasons. First, we only included electrodes located in gray matter, which removed a large number of contact points within each electrode (~50%). The removal of many electrode contacts introduced variability in sampling density, affecting both patient cohorts similarly. Second, a higher sampling density does not necessarily equate to more information, as closely spaced electrodes in iEEG often capture activity from the same sources and exhibit similar neurophysiological profiles. We note that most of our analyses involved averaging electrode activity in each subject independently before performing statistical comparisons at the group level, which further reduces the risk that sampling density differences systematically biased our findings. Third, many of our key findings were observed in the AMY, which had the lowest sampling density overall, with a mean number of electrodes per subject of 2.56 ± 1.39 . Notably, 10 out of 22 patients had only a single electrode in the AMY, making it unlikely that inter-electrode distances or contact surface area significantly influenced our results in this region. Fourth, the inter-contact spacing of the first two electrodes in the micro-macro arrays from Paris (3 mm) closely resembles that of the electrode configurations used in Guangzhou (3.5 mm), minimizing potential sampling density differences in these electrodes.

We performed two control analyses to corroborate that different electrode configurations in our patient's cohort did not affect the density of neural tissue sampled. First, we computed the Delaunay triangulation of electrode configurations, a method that quantifies electrode density by forming triangles between contact points and calculating the mean triangle area. Focusing on our largest and more densely implanted ROI (the TMP) we evaluated triangulation in the left and right hemispheres in each subject independently. Only subjects with at least three electrodes in the TMP (N=36) were included in this analysis, as a minimum of three electrodes is required to form triangles. Statistical significance was assessed using a two-way ANOVA with "Recording site" (Guangzhou/Paris) and "Hemisphere" as factors. Results showed no significant difference in mean triangle area between hemispheres ($F(1,32) = 2.61, p = 0.11$) or between patient cohorts ($F(1,32) = 0.96, p = 0.33$). These findings indicate that although electrode spacing varied due to differences in electrode configurations, the overall sampling density remained consistent across our two patient groups.

To further corroborate that the different electrode configurations did not affect our results, we conducted our oscillatory power analyses in the AMY separately for Guangzhou and Paris patients. We specifically focused on the significant cluster observed in the main analysis contrasting "current" valence (CS+ vs CS-) during extinction. We observed the same pattern of results across our two cohorts of patients, with higher theta power for CS- as compared to CS+ items during extinction (Guangzhou: $t(19) = -4.32, p = 0.0004$; Paris: $t(11) = 2.93, p = 0.014$). Taken together, the results of these analyses suggest that our two patient cohorts did not show prominent differences in the neural tissue sampled

in the TMP, and that the AMY theta power results during extinction were consistent in our two patient groups (see Supplementary Figure 2).

Finally, we highlight that recent years have seen an increase of multi-center iEEG research, which involves collecting and analyzing data from multiple epilepsy centers to generate larger and more diverse datasets. Several studies have successfully integrated multi-center iEEG data despite variations in electrode configurations, including even combination of subdural strips and grids with depth electrodes, which differ in regard to sampling density even more than different types of depth electrodes do (Bernabei et al., 2023; Dimakopoulos et al., 2022; Henin et al., 2021). While there is currently no established consensus or guideline on how to address sampling differences in the literature, we believe this is an important topic of research in the future, as multi-center iEEG studies are becoming more prominent.

5. While it is often necessary to aggregate electrode locations into broader regions, the TMP and IPFC regions described in the study appear to cover relatively large areas. Additionally, I could not find specific information on the exact locations of the electrodes within the HPC (e.g., CA sectors) and AMY (e.g., amygdala nuclei). Could you provide clarification or additional details about the precise electrode placement in these regions?

We thank the reviewer for this question. The selection of our regions of interest (ROIs) was guided by three main criteria: (1) inclusion of regions implicated in fear conditioning based on prior literature, (2) selection of regions involved in representing visual and categorical properties, given our interest in performing RSA on cues and contexts, and (3) a minimum of 10 subjects with electrode coverage in each region. Based on these criteria, we initially focused on the prefrontal cortex (PFC), hippocampus (HPC), amygdala (AMY), temporal cortex (TMP), and occipital cortex (OCC).

Previous fMRI studies investigating extinction learning have also mentioned relevant roles of frontal regions such as the anterior cingulate cortex (ACC), dorsomedial PFC (dmPFC), and ventromedial PFC (vmPFC) / orbitofrontal cortex (OFC) (Fullana et al., 2016). However, electrode coverage in these regions was limited in our patient cohorts, except for the OFC. Therefore, we categorized frontal electrodes into two groups: the OFC and the prefrontal cortex (PFC). PFC electrodes included all frontal electrodes that were not classified as orbitofrontal in the Desikan-Killiany Atlas (Desikan et al., 2006). Specifically, these included the *caudal middle frontal, pars opercularis, pars orbitalis, superior frontal, pars triangularis, rostral middle frontal, and frontal pole*. As the majority of electrodes in these regions were located in lateral areas, we labeled this group as lateral prefrontal cortex (IPFC).

Additionally, we aimed to characterize cue and context representations in sensory regions, particularly the OCC and the TMP, due to their established roles in the representation and processing of item and categorical information (Cichy et al., 2014; Pacheco Estefan et al., 2019; Pacheco-Estefan et al., 2024). However, because the number of subjects with electrodes in the OCC was below our inclusion threshold of 10, we excluded this region from further analyses. In the TMP, we included the following bilateral regions: *inferior temporal, middle temporal, superior temporal, transverse temporal, fusiform, temporal pole, banks of the superior temporal sulcus (bankssts), parahippocampal, and entorhinal cortex*.

We aggregated data from all subregions in the IPFC and TMP, as we did not have a priori hypotheses regarding their specific contributions to fear extinction or stimulus representation. This grouping was also informed by prior iEEG studies employing RSA to examine visual feature representations in working memory (Pacheco-Estefan et al., 2024). We aimed to maximize the sensitivity of our analyses to extract context-specific activity. While our grouping included seven subregions in the IPFC and nine in the TMP, most of our electrodes were concentrated in the bilateral *rostral middle frontal* and *pars opercularis* in the IPFC, and in the *inferior, middle, and superior temporal gyri* in the TMP (with an approximately equal distribution across these three gyri, see Figure below).

Brain Region	Electrodes	Subjects	Brain Region	Electrodes	Subjects
Bankssts	20	11	Caudalmiddlefrontal	3	3
Entorhinal	8	6	Parsopercularis	23	11
Fusiform	51	25	Parsorbitalis	2	1
Inferiortemporal	77	33	Parstriangularis	9	5
Middletemporal	85	36	Rostralmiddlefrontal	37	19
Parahippocampal	12	8			
Superiortemporal	54	22			
Temporalpole	10	5			
Transversetemporal	8	5			

Supplementary Figure 3. Proportion of electrodes in TMP and PFC subregions.

Proportion of electrodes included in specific subregions of the TMP (left) and the PFC (right; Desikan Killian atlas; Desikan et al., 2006) across all subjects. A summary table for each ROI, including the number of electrodes and the number of subjects in each subregion, is presented below the pie plots.

Electrode placement in HPC and AMY subregions

To provide a comprehensive description of the locations of our HPC and AMY electrodes—and also of the electrodes included in our other ROIs—, we now provide the MNI coordinates of all electrodes in the new Supplementary Table 2. In addition, we included the Supplementary Video 1 with the new submission, which provides more detailed visualizations of AMY and HPC contacts.

In our study, we did not aim to obtain subregion-specific labels in our AMY and HPC contacts for several reasons. First, we aimed for a comprehensive and parsimonious analysis across various larger brain regions involved in fear learning and extinction, and the contribution of specific subregions in the AMY and the HPC was not among our primary objectives. Second, since the metrics of representational similarity we employed

capture representational patterns across electrodes, we needed as many electrodes as possible in each ROI to increase the sensitivity of our analysis. We acknowledge that such across-electrode representational analyses inherently limit the spatial resolution, but believe that this analysis choice is well suited to obtain robust measures in larger brain areas (see Rau et al., 2025 or Xie et al., 2020 for a similar approach). Third, even with our unusually large sample, the number of patients with electrodes in specific HPC subregions or AMY nuclei would be very limited. This is even more the case if subregions were divided across the longitudinal axis and across left and right hemispheres of the brain.

In the AMY, it is likely that most of our electrodes are located in the basolateral complex, as this is the largest subregion in AMY, covering ~75% of the total volume. However, we cannot exclude the possibility that our electrodes captured activity from other subregions (e.g., centromedial complex), or a combination of activity of these different sources.

While we did not apply methodological approaches to obtain subregion-specific labeling, we note that this is an excellent topic for future research, as research in animal models has shown distinct involvement of specific AMY and HPC subregions in fear extinction (Davis & Reijmers, 2018; Tonegawa et al., 2015; Zaki et al., 2022). Notably, this could be studied not only with iEEG, but also with other methods which provide excellent spatial resolution, such as 7T fMRI.

6. In the preprocessing section, it is unclear how seizure activity was excluded from the data. The manuscript only mentions the use of amplitude criteria. Could you clarify whether additional methods were employed to identify and exclude seizure activity, such as visual inspection or frequency-based criteria?

We thank the reviewer for the opportunity to clarify how seizure activity was excluded from our data. We systematically removed all epileptic activity using both an automatic detection algorithm specifically designed to identify epileptic spikes (which actually did not only include an amplitude criterion but also a gradient criterion and a high-frequency amplitude criterion, see below; Staresina et al., 2015) and thorough visual inspection of the data after epoching it. The visual inspection was performed using both the amplitude time-series and time-frequency decomposed data (spectrograms).

The automatic detection algorithm, originally proposed by (Staresina et al., 2015), identifies epileptiform activity based on three key criteria: (1) the amplitude of the time-series, (2) the gradient (i.e., the amplitude difference between adjacent time points), and (3) the amplitude of the data after applying a 250-Hz high-pass filter, which further enhances the detection of epileptogenic spikes. In our study, each time point in the raw iEEG data was converted into a z-score based on the participant-specific mean and standard deviation of these three measures across the whole experiment in each channel

independently. A time point was marked as artifactual if it exceeded a z-score of 6 in any measure or met a conjunction threshold of an amplitude z-score of 4 combined with either a gradient or high-frequency z-score of 3 (Staresina et al., 2015).

We found that the combination of raw EEG amplitudes, EEG gradients, and high-pass filtered EEG amplitudes at 250 Hz provided high sensitivity for detecting epileptiform activity and other artifacts. This was further validated through manual identification of artifacts in randomly selected subsets of the data. To ensure that all epileptic activity was completely removed from our data, we not only excluded the detected time points but also removed the 1,000ms preceding and following each detected artifact sample.

After identifying and marking the segments of the data contaminated with artifacts, we segmented the data into 7-second epochs (from -3 to 4 s) around the presentation of each CS cue. Any epoch containing artifact segments detected in the continuous (non-epoched) data was entirely removed from further analysis. Additionally, we visually examined raw time-series plots and spectrograms to assess the presence of artifacts in the frequency domain. As stated in the original manuscript, The number of epochs removed due to artifacts varied depending on the signal quality of each subject and was at 20.6 ± 19.2 epochs of cue presentation across subjects and channels.

In the revised version of the manuscript, we describe this procedure in greater detail in the Methods section (lines 911-928), where we now write:

“Artifact rejection was performed using a previously published method (Staresina et al., 2015). This automatic detection algorithm identifies epileptiform activity based on three key criteria: (1) the amplitude of the time-series, (2) the gradient (i.e., the amplitude difference between adjacent time points), and (3) the amplitude of the data after applying a 250-Hz high-pass filter, which further enhances the detection of epileptogenic spikes. Each time point in the raw iEEG data was converted into a z-score based on the participant-specific mean and standard deviation of these three measures across the whole experiment in each channel independently. A time point was marked as artifactual if it exceeded a z-score of 6 in any measure or met a conjunction threshold of an amplitude z-score of 4 combined with either a gradient or high-frequency z-score of 3.

We found that the combination of raw EEG amplitudes, EEG gradients, and high-pass filtered EEG amplitudes at 250 Hz provided high sensitivity for detecting epileptiform activity and other artifacts. This was further validated through manual identification of artifacts in randomly selected subsets of the data. To ensure that all epileptic activity was completely removed from our data, we not only excluded the detected time points but also removed the 1,000ms preceding and following each detected artifact sample.”

Criterion 7 – References

Comments:

I have no comments.

Criterion 8 - Clarity and context

Comments:

I have no comments.

Reviewer #3 (Remarks to the Author)

In this study, Pacheco and colleagues examine intracranial iEEG recordings to explore the neural signal that underlie extinction of fear memories in the human brain. Briefly, they employ a paradigm where subjects learn to associate individual items with an aversive stimulus during an acquisition phase. There are two categories of items, those that remain associated with an aversive stimulus during the extinction phase (CS++) and those with removal of the association with the aversive stimulus during extinction (CS+-). In this manner, they can examine how fear memories are extinguished. There is also a third category of items that are never paired with a negative aversive stimulus (CS--). Based on these conditions, the authors examine changes in theta power, previously associated with fear conditioning in the amygdala, as well as the stability of item representations. Finally, the authors introduce these items against a background of a video of different scenes, which serves as a context. This allows them to examine the stability and representations of context. Overall, this is an interesting study, and certainly leverages the rare ability to explore these neural signals with high spatial and temporal resolution. There are a number of conclusions that the authors draw from their results, but there are some suggestion that would perhaps strengthen the interpretability of their results and their conclusions.

We appreciate the referee's positive assessment of our manuscript and their constructive suggestions for improvement. Below, we provide a point-by-point response to all the issues raised, including several additional analyses that we believe will indeed enhance the interpretability of our findings.

1. For the behavioral analysis presented in figure 1, the authors use a paired t-test at every trial. Is this performed across subjects? It appears that is the case, but then raises the question as to how the trials may be matched across subjects, or how this is performed. For example, trial 9 may be a CS+- trial in one subject and a CS++ item in another. It would help to clarify how this is performed.

We thank the reviewer for the opportunity to clarify and elaborate on our trial-by-trial progressive learning analysis approach. In this analysis, we sorted trials by type (CS++, CS+-, and CS--) and performed a paired t-test across subjects at each trial to compare performance across conditions. Please note that sorting trials in this way disrupted the absolute trial positions in the experiment, but the sequential order of trials within each condition was maintained. For a specific trial (e.g., trial 9), the CS type could be different across subjects, as the reviewer correctly notes. However, after sorting by condition, the first CS++ in all subjects is always compared with the first CS+- and the first CS-- trial, the second with the second, and so on. Thus, the paired t-tests are performed across subjects in individual trials corresponding to the same ordinal position in each condition.

Below, we provide an example to illustrate how the sorting was performed with simulated data of two subjects. Note that in this example, the first CS++ trial corresponds to trial 1 for the first subject, but to trial 3 for the second subject. Since the responses were recorded on an ordinal scale from 1 (Threatening) to 4 (Safe), the group-level distributions at each trial were not normally distributed. We thus confirmed the statistical significance of our results using a Wilcoxon signed-rank test, a non-parametric equivalent of the paired t-test (see also our response to Reviewer 2, Criterion 3, *Appropriate use of statistics and treatment of uncertainties*).

Trial number during experiment	1	2	3	4	5	6	7	8	9	10	11	12	13	14	15
Trial type Subject 1	CS++	CS+-	CS++	CS--	CS++	CS--	CS--	CS+-	CS++	CS--	CS+-	CS--	CS+-	CS+-	CS++
Trial type Subject 2	CS+-	CS--	CS++	CS++	CS+-	CS--	CS--	CS+-	CS+-	CS++	CS--	CS++	CS+-	CS--	CS++

	Condition-specific trial number	1	2	3	4	5
Subject 1	CS++	1	3	5	9	15
	CS+-	2	8	11	13	14
	CS--	4	6	7	10	12
Subject 2	CS++	3	4	10	12	15
	CS+-	1	5	8	9	13
	CS--	2	6	7	11	14

This example illustrates how trial numbers in the experiment were transformed into condition-specific trial numbers. Values in the table correspond to the trial number during the experiment for each trial/condition for two hypothetical subjects. Note that this illustrative example does not reflect the actual number of trials and subjects in the experiment. After sorting trials by conditions independently in each subject, group-level paired t-tests were applied at every trial.

We clarified this point in the Methods section (lines 808-819), where we now write:

“[...] In this analysis, we sorted trials by type (CS++, CS+-, and CS--) and performed a Wilcoxon signed-rank test across subjects at each trial to compare performance across conditions. While sorting trials in this way disrupted the absolute trial positions in the experiment, the sequential order of trials within each condition was maintained. For a specific trial (e.g., trial 9), the CS type could be different across subjects. However, after sorting by condition, the first CS++ in all subjects is always compared with the first CS+- and the first CS-- trial, the second with the second, and so on. Thus, the paired Wilcoxon tests were performed across subjects in individual trials corresponding to the same ordinal position in each condition. We corrected for multiple comparisons using cluster-based permutation statistics, shuffling the condition labels at every trial”.

2. The main result regarding theta power in the amygdala that the authors present is that theta power is decreased in CS++ trials during extinction compared to CS+- trials. They interpret this to reflect a safety signal, yet it is unclear in that case why there would be no difference between CS+ and CS- trials during acquisition. Is this really reflecting safety, or some change in expectation with the trials that occurs after the CS+- items have changed their contingencies. This is especially the case since the authors present results based on CS+ versus CS- trials during extinction, so this should also be true during acquisition. If instead they are focusing on how these contingencies change, they should specifically focus on the change in contingencies, they should also examine CS++ versus CS+- trials.

We thank the Reviewer for this question. First, we would like to clarify that in the contrasts presented in Figure 2A, we compared trials reflecting *current* valence during extinction, i.e., we aggregated CS+- and CS-- trials into an overall CS- category and contrasted them with CS++ trials. We reasoned that current valence would be the most prominent factor influencing AMY theta power responses during extinction and aimed to perform the most general and sensitive contrast first, including the maximum number of trials in each condition (see also our response to Reviewer 2 Criterion 1, *Data & Methodology*, Point 4). Also, following successful extinction of CS+- trials, they should be represented similar to CS-- trials.

As the results of the current valence contrast show higher theta power for safe cues, i.e., CS-, we concluded that theta increases in these trials reflect their safety. However, we acknowledge the reviewer's point that this conclusion might be a bit oversimplified, as we do not observe a theta power increase for CS- trials during acquisition. In the revised manuscript, we now describe this interpretation in a more nuanced manner (see below).

To test whether the observed theta power effects during extinction reflect safety or changes in expectations / contingencies, we separately compared CS++ trials with CS+- trials as suggested by the reviewer, and for completeness we also compared CS++ versus CS-- trials and CS+- vs CS-- trials. Results showed significantly lower theta power in CS++ trials compared to both CS+- ($p = 0.005$) and CS-- trials ($p = 0.026$), with effects occurring in overlapping time periods to those observed in the "current valence" contrast in Figure 2A (1.18-1.75s, i.e. towards the end of cue presentation). Notably, the significant differences between CS++ and both CS+- and CS-- trials suggest that contingency changes alone do not drive theta activity, as this would predict a specific difference between CS++ and CS+-, but not between CS++ and CS--. Consistent with this interpretation, we did not observe any significant differences between CS+- and CS-- trials ($p = 0.38$), a contrast in which contingencies change but current valence is the same.

We propose that a parsimonious interpretation of our findings is that the positive valence of the CS- during extinction, reflecting safety in both CS+- and CS-- trials, drives theta activity. Critically, this safety during extinction is not the same as during acquisition, as highlighted by the reviewer, as at least two factors are different: (1) Contingencies have changed for one of the cues (CS+-) and (2) contexts have changed. Thus, rather than reflect safety per se, our findings demonstrate that theta power reflects safety specifically during extinction, i.e., safety after contingencies and contexts have changed.

We have revised Figure 2 to include the additional contrasts presented in this response (CS++ vs. CS+-, CS++ vs. CS--, and CS+- vs. CS--). In addition, we have modified the discussion of the manuscript to highlight the role of AMY theta power in signaling safety specifically during extinction. In lines (473-483), we now state:

“Notably, the increase in theta power for CS- items observed during extinction, coupled with the absence of such increases during acquisition, suggests that theta power does not simply signal safety but rather safety within the specific context of extinction. At least two key differences distinguish acquisition from extinction in our paradigm: (1) Contingencies changed for one of the cues (CS+-), and (2) contexts changed from A to B. While both of these factors may have influenced theta responses, future research is needed to determine whether theta power during extinction signals safety specifically when both context and contingencies change in the context of a multi-cue paradigm such as ours, or whether it would also be observed in extinction learning paradigms in which contexts do not change (e.g., AAB designs) or in those including only one cue (CS+-).”

3. Similarly, in the analysis of item stability, the authors make the primary comparison between CS+ and CS- trials. They show that CS+ trials increase their stability during extinction. However, again, this raises the question as to why they do not see these differences during acquisition. Again, if the focus here is on the change in contingencies for the CS+- items, then the comparison of item stability should also include analyses of CS++ versus CS+- items rather than all CS- items.

As in our analysis of AMY theta power, we initially focused on the contrast between current valence in the item stability analysis, as this contrast provides the most general and statistically powerful comparison and reflects the result of successful extinction, which should result in similar representations of CS+- and CS-- items. Nevertheless, we agree with the reviewer on the importance of assessing the effect of contingency changes in the item stability analysis as well. We conducted the same additional analyses as for AMY theta power during extinction (see previous point): (1) CS++ vs. CS+-, (2) CS++ vs. CS--, and (3) CS+- vs. CS--. We performed these analyses separately for the AMY and TMP, focusing on the temporal windows where we observed significant effects in the original analyses (AMY: 1.25–1.5s; TMP: 0.65–1s).

We observed the following results (see Figure below):

- A. CS++ trials exhibited significantly higher item stability than CS+- trials during extinction in both AMY ($t(31) = 2.30, p = 0.028$) and TMP ($t(40) = 2.46, p = 0.018$).
- B. CS++ trials exhibited significantly higher item stability than CS-- trials during extinction in AMY ($t(31) = 2.39, p = 0.023$), while they showed a numerical but not statistically significant difference in TMP ($t(40) = 1.64, p = 0.11$).
- C. CS+- trials were not significantly different from CS-- trials during extinction in either AMY ($t(31) = 1.3, p = 0.2$) or TMP ($t(40) = 0.52, p = 0.61$).

Considered together with the results presented in our original manuscript (Figure 3), these results demonstrate that item stability was not sensitive to contingency changes but was specifically increased for items that were rated as threatening during extinction in the AMY. In the TMP, the results are qualitatively similar (though not identical): While CS++ trials were higher than CS+-, the contrast between CS++ and CS-- trials did not reach significance. However, we note that the more prominent differences in the TMP were observed between trials with different valence (i.e., between CS++ and CS+- and between CS++ and CS--, which followed the same direction), while CS+- and CS-- trials showed very similar levels of item stability (CS+-: 0.011 ± 0.027 , CS--: 0.014 ± 0.029).

Taken together, these results suggest that item stability in both AMY and TMP reflects the successful extinction of the previous valence representations of the CS+- items, because these previously threatening cues are in this phase processed similar to cues that were previously safe (CS--; see also our response to Reviewer 2 Criterion 1, *Data & Methodology*, Point 4). We modified Figure 3 of our manuscript to include these control analyses, and adapted the main text in the Results section accordingly (lines 246-261).

Item stability during extinction. Item stability during extinction for three different contrasts (CS++ vs CS+-, CS++ vs CS--, and CS+- versus CS--), during the time period where the significant differences in item stability for CS+ vs. CS- trials were observed in the AMY (1.25–1.5s, top row) and the TMP (0.65–1s, bottom row). These plots have been included in Figure 3C and 3E of the revised manuscript.

4. The authors examine coordination of item stability across brain regions by computing correlations between RSA similarity values. This is an interesting approach. Had the authors also considered simply examining standard metrics of connectivity between these brain regions during these times?

We thank the Reviewer for this suggestion. Conceptually, we were predominantly interested in investigating the representations of contexts and items during fear learning and extinction. For this reason, we employed metrics of representational coordination rather than standard connectivity analyses in order to investigate the propagation of representational signatures throughout the fear and extinction network. However, we acknowledge that connectivity metrics could offer complementary insights into the neural mechanisms of fear extinction.

Over the past decades, several connectivity metrics have been developed and applied. One of the first to be proposed was the Phase Locking Value (PLV) (Lachaux et al., 1999), which quantifies the consistency of phase angle differences between two signals. The PLV has been widely applied in cognitive neuroscience (Mormann et al., 2000; Pacheco Estefan et al., 2019; Varela et al., 2001), but this metric is highly susceptible to volume

conduction since it includes apparent connectivity at phase differences at zero lag. Volume conduction refers to the phenomenon where electrical activity from neuronal sources propagates almost instantaneously through brain tissues, leading to signal spread. This effect can artificially inflate metrics of connectivity in iEEG, as different electrodes might capture activity from the same neurophysiological source.

To circumvent the problem of volume conduction, several metrics based on phase lag were developed in the last years. In particular, phase-based connectivity measures were established that ignore zero phase-lag connectivity, including the phase-slope index (Nolte et al., 2008), the phase lag index (Stam et al., 2007), and the weighted phase-lag index (Vinck et al., 2011). The Phase Lag Index (PLI) quantifies asymmetries in phase lead/lag, ignoring synchronized activity at zero or 180° phase lag—which is often the result of volume conduction rather than true connectivity. The weighted PLI (wPLI) assigns greater weight to larger phase differences by considering the magnitude of the imaginary component of the cross-spectrum, effectively suppressing small phase differences that may result from volume conduction. By improving robustness against noise as compared to the PLI and minimizing false connectivity, wPLI has become a widely employed metric for mapping functional networks in EEG/iEEG research (Imperator et al., 2019; Vinck et al., 2011).

To complement our representational coordination analyses and provide a more comprehensive description of how fear extinction signals propagate throughout the fear and extinction network, we assessed the wPLI between pairs of ROIs. We specifically focused on the ROIs where we observed significant coordination in our item stability (Figure 3D) or context specificity (Figure 4D) analyses. These included AMY-TMP, AMY-HPC, AMY-IPFC, TMP-HPC, TMP-OFC, and TMP-IPFC. We computed wPLI across trials for each pair of electrodes in their correspondent ROI pairs, focusing on the theta (4–8 Hz) frequency band, as this was the frequency range where we observed the most prominent effects in the AMY in our data. Notably, phase-based connectivity in the theta range has been associated with long-range neural communication (Fell & Axmacher, 2011). We specifically selected the time periods where we observed significant coordination effects in the original analysis (see table below).

We computed wPLI for all electrode pairs within predefined ROI pairs, separately for each subject and each of our three cue types (CS++, CS+-, and CS--). In subjects with bi-hemispheric implants, we computed wPLI for channel pairs within the same hemisphere only. wPLI metrics obtained for each electrode pair were averaged across electrodes in each subject before performing statistical comparisons. Significant differences were assessed with a repeated measures one-way ANOVA.

Our results did not reveal any significant differences in wPLI during the time periods of significant coordination in any of the ROI pairs even at an uncorrected level (see table below).

ROIs	Time period	ANOVA results
AMY-TMP	0-1.75s	$F(2,24) = 2.22, p = 0.12$
AMY-HPC	0.5-1.75s	$F(2,20) = 1.08, p = 0.34$
AMY-IPFC	1-1.65s	$F(2,10) = 0.16, p = 0.85$
TMP-HPC	0-1.45	$F(2,22) = 0.09, p = 0.91$
TMP-OFC	0.15-1s	$F(2,7) = 0.72, p = 0.5$
TMP-IPFC	0-0.8s	$F(2,14) = 0.24, p = 0.78$

The absence of significant differences in our phase-based connectivity analysis suggests that the observed coordination in representational signals in our study is not linked to phase-based connectivity in the theta frequency range. However, we acknowledge that the space of possible connectivity analyses is vast, and our metric of representational coordination may align more closely with other forms of connectivity, such as power-based or causal connectivity, which could be investigated more comprehensively in the future.

We have included the new Supplementary Note 6 in the revised manuscript to incorporate these analyses.

5. Many of the conclusions are based on the links that are thought to be established between items and context, and in particular between fear memories and context. This builds on an extensive prior literature. However, here, the role of context is unclear. As far as I can tell, it looks like the context video presented during each item is randomly chosen (from one of four videos per experimental stage), and is not directly linked with a specific item. It is certainly possible that my reading is incorrect. But if this were the case, its not clear how context might serve a role in fear memory here, since context being randomly assigned becomes irrelevant. Instead, one would expect that a more direct analysis here would require that each item (CS+ or CS-) be linked with a specific context. Because of this, it is not clear how to interpret many of these data. For example, they show that there is greater context stability during the early stages of acquisition in HPC and during extinction only in PFC. Why only these time points and stages of the experiment, since context is presented on all trials. This makes it hard to interpret, especially since there is no direct association between context and each individual item or even contingency (CS+ versus CS-).

We thank the reviewer for raising this question, which allows us to clarify how context was operationalized in our paradigm, and how we evaluated the relationship of item and context representations. We defined context as different videos instances of a particular landscape category, which differed between the phases of our experiment. For example, videos of city scenes were presented during acquisition, videos of sea landscapes were presented during extinction, and videos of countryside and mountain landscapes were presented during the test (these videos were more diverse to avoid any possible bias in the likelihood of representational reinstatement from the previous stages). These categories were counterbalanced between participants, i.e. another participant would see videos of countryside landscapes during acquisition, videos of mountain landscapes during extinction, and videos of sea landscapes and city scenes during test. As the reviewer correctly points out, the individual videos in each phase were unrelated to the valence of the cues—e.g., one particular video of a city scene occurred with equal probability for the two different CS+ cues and the CS- cue during acquisition. We chose this specific design because it allowed us on the one hand to employ a classical “ABC” protocol—reflected by the different video categories between the three experimental phases—and on the other hand, to test a core idea of theories of extinction learning, namely that representations of extinction contexts are more specific than representations of acquisition contexts (Maren et al., 2013). In the following, we will elaborate on these design features.

Our approach is closely aligned with traditional paradigms investigating context-dependent extinction learning and follows a prototypical ABC design. In such designs, the acquisition context (A) is unrelated to the cue contingencies, since it occurs for both CS+ and CS- trials, as in our paradigm. Similarly, the extinction context (B) occurs for all cues independent of their contingencies. Importantly, while only the CS+ trials actually changed their contingencies during extinction in our paradigm, participants had to reassess the contingencies for all cues because they could not know beforehand which contingencies may change and which would remain stable. Thus, in conjunction the four videos of each phase have the same learning-related properties as the contexts in conventional ABC designs: they do not predict contingencies in each experimental phase, but they do signal the change of contingencies from acquisition to extinction and to test.

Unlike conventional ABC designs, we presented multiple different videos in each phase. This was done to test the hypothesis that the neural representations of contexts during extinction are more specific (i.e., less generalized) than the neural representations of contexts during acquisition: Since a key objective of our study was to test the hypothesis that extinction learning is highly context-dependent (Maren et al., 2013), we were interested in evaluating context specificity during the extinction phase of our experiment, compare it to context specificity during acquisition, and test the effect of context specificity on the renewal of fear. The assessment of context specificity via RSA required the

inclusion of several thematically related videos in each experimental phase. Context specificity was then quantified by contrasting the representational similarity of neural activity during identical vs. different contexts (Kriegeskorte et al., 2008).

Importantly, while our analysis of context specificity is not by itself related to the representation of the cues—as the contexts are orthogonal to the cues—we analyzed whether contexts were nevertheless relevant for the reinstatement of fear and extinction memories. We correlated context specificity in the IPFC during extinction with the reinstatement of safe versus threatening cues in TMP, separately for each cue type (CS++, CS+-, and CS--) and observed a very specific relationship for the CS+- trials, where IPFC context-specificity modulates the reinstatement of fear memory traces from acquisition in the TMP (Figure 6).

We acknowledge the relevance of further interpreting our findings of context specificity in the HPC and IPFC. These analyses revealed a clear dissociation between context representations in the IPFC and HPC in terms of their temporal dynamics within a trial and their occurrence across experimental phases: In the HPC, context-specific representations occurred only during acquisition, and only during the period when the video was presented without the cue (please note this effect is observed “early” within a trial, and not within the experimental phase of acquisition, since all acquisition trials were included). In contrast, in the IPFC, context specificity was observed only during extinction, and only when both the context and the cue were displayed (here again, all trials during extinction were included).

The effect in the PFC aligns with literature suggesting a role of this region in inhibiting fear expression during extinction, and more generally with the presumed high context specificity during extinction (Maren et al., 2013). The HPC results, on the other hand, are consistent with research which has linked this region to the rapid formation of associative memories (McClelland et al., 1995). We speculate that the selective involvement of the HPC during acquisition reflects its role in rapidly encoding novel memory traces, as opposed to the PFC which shows a slower learning of the context representations. In addition, the well-known involvement of the HPC in “scene construction” (Maguire & Mullally, 2013) and in the processing of viewpoint-related information (Rolls, 1999) might have played a role in its specific involvement during the periods where the videos were presented. However, we also acknowledge that these interpretations remain speculative at this point.

To highlight these aspects of our findings, we have extended the discussion of our manuscript (see lines 563-584), which now states:

“Our results highlight the crucial role of both HPC and IPFC in encoding contextual information during fear learning and extinction, consistent with findings in rodents (Bouton, 2004; Gilmartin et al., 2014; Maren et al., 2013). Notably, the IPFC was

selectively engaged in the representation of context-specific activity during extinction and not during acquisition, again consistent with its involvement in the suppression of fear memories via context-signaling. While we observed a context-specific representation during acquisition as well (in the HPC), this effect did not coincide with the time period of the cue presentation – suggesting that specific contexts are not neglected during acquisition but exert a less prominent influence on the memory traces of individual cues. While the IPFC effects align with the role of this region in inhibiting fear expression (Gilmartin et al., 2014; Maren et al., 2013), the HPC responses are consistent with the greater sensitivity of this region to views of landscapes and scenes (Maguire & Mullally, 2013; Rolls, 1999) and the well acknowledged role of the HPC in rapidly encoding new memory traces during acquisition (McClelland et al., 1995). We note that several studies have shown that fear acquisition is characterized by an overgeneralized and decontextualized fear responses, while extinction engages PFC-dependent mechanisms more reliant on context representation (for reviews, see Gilmartin et al., 2014; Maren et al., 2013). We previously found that overgeneralization of representations predicted subsequent intrusive memories (Kobelt et al., 2024), which are notorious for their decontextualized nature and the fact that they can be easily and involuntarily triggered by ubiquitous sensory cues.

In addition, we have extended the description of our paradigm in the methods section (see also response to Reviewer 2, Criterion 1, *Data & Methodology*, point 3), in order to specifically discuss our operationalization of context and the use of thematic context videos for the RSA analysis. In lines 759-766, we now write:

“The association of the contexts and the cues was completely randomized in every subject, so that contexts were not predictive of the CS. Please note that this approach differs from conventional fear and extinction learning paradigms. Our design was developed to leverage the power of RSA and iEEG to dynamically track the representation of contexts across key hubs of the extinction network. On the other hand, our paradigm aligns with conventional ABC paradigms as it contains three different context categories across experimental phases (See Supplementary Note 4)”.

6. When looking at the correlations of context specificity, they find a significant correlation between PFC context stability and context stability in the Amygdala, but this is at a delayed time (the cluster is clearly above the diagonal in this plot). How should one interpret this, especially since there is no evidence of context stability in the amygdala.

We assessed potential temporal offsets between regions in all our analyses of representational coordination, including both item stability and context specificity,

because latencies and durations of neural representations may differ across brain regions, making it more likely that coordination is temporally shifted rather than simultaneous (see King & Dehaene, 2014; Pacheco Estefan et al., 2019). Notably, the relative timing of coordination offers insight into the temporal order in which representations occur across these regions.

Regarding the coordination of context specificity between IPFC and AMY (Figure 4D), we indeed found the most prominent relationship at shifted time periods (i.e., the amount of context specificity during earlier time points in the IPFC corresponded to the amount of context specificity during later time points in the AMY). To improve the visualization of this effect, we have plotted the diagonal in the coordination map (see figure below). Trials that exhibited increased context specificity in the IPFC at around 500ms after cue onset showed increases in the AMY after around 1s—i.e., with a lag of around 500ms—while this lag decreases for later time periods in the IPFC.

We conducted an additional analysis to assess the relative timings more precisely, and for this purpose performed the t-tests of the group-level correlations against zero at an alpha level of 0.01 instead of 0.05 (highlighted with a dashed outline in the figure below). This analysis showed a maximal coordination between 0.85s and 1.4s in IPFC time, i.e., partially overlapping with the IPFC's significant context-specificity window (0.8 to 1.15s; see also Figure 4C). The context-specific effect in the IPFC is significant in this time period of maximal coordination ($t(21) = 2.69$; $p = 0.0137$). Thus, while the coordination effect at an alpha level of 0.05 is naturally broader and includes all significant context-specificity time periods in the IPFC, the fact that the IPFC context-specificity effect precedes the time period of strongest coordination suggests that context specificity in the IPFC is driving the effect in the AMY.

As pointed out by the Reviewer, context specificity in AMY during the time window of significant coordination (1s to 1.65s) was not statistically significant ($t(31) = 0.86$, $p = 0.39$). To our knowledge, the AMY has not been directly linked to the representation of context during fear extinction. We would thus assume that it only contributes to this process as part of a network of brain regions including HPC and IPFC. The absence of a main effect of context specificity in the AMY in our data indicates that in this region, the similarity between representations of identical contexts is not substantially higher than the similarity between representations of different contexts. We speculate that the lack of increased similarity in within-context comparisons could be driven by the phenomenon of repetition suppression, where neural responses to similar context videos are attenuated when the same video is shown repeatedly, making the representations of the same videos as different as representations of different videos.

We have expanded the Methods section in the revised manuscript (lines 1035-1052), providing a more detailed rationale of assessing on-diagonal and off-diagonal effects in our coordinated representations analysis.

“We computed a single-trial metric of item-stability by assessing the similarity of the neural representation of an item presented in a trial with the representation of the same item in all other trials during a particular experimental phase. In addition, we computed a single-trial metric of context specificity by subtracting the similarity of each context video with all repeated presentations of the same video and the similarity with all different videos in a particular experimental phase. These metrics were computed in each of our ROIs and then correlated between different ROIs across trials for each participant. Correlations were calculated in sliding time windows (500ms windows, overlapping by 95%) across all possible combinations of time points. This resulted in one correlation value for each subject in each time-bin x time-bin pair. We assessed potential temporal offsets because the latency and duration of representations may differ across brain regions, making it more likely that coordination is temporally shifted rather than simultaneous (see King & Dehaene, 2014; Pacheco Estefan et al., 2019). Notably, the relative timing of coordination offers insights into the temporal order of representations across brain regions during fear extinction. Statistical significance was assessed by contrasting these rho values against zero at the group level, and correction for multiple comparisons was performed using cluster-based permutation statistics (see below)”.

Additionally, we have incorporated a new paragraph in the Discussion to interpret the lagged correlation of trial-level context specificity between the IPFC and AMY. In lines 509-530, we now state:

“The high temporal resolution of iEEG recordings allowed us to shed light on the relative time courses of signals across regions of the extinction network. Our results reveal a temporal sequence in which an earlier representation of specific contexts in IPFC is followed by a later correlation of $IPFC_{CONTEXT}$ and $AMY_{CONTEXT}$ across trials, which, in turn, coincides with the safety-related theta power increase in AMY. This temporal order of representational and neurophysiological signals aligns with previous research showing that distributed areas of the extinction network, including the PFC, first coordinate the processing of contextual information, which subsequently influence the AMY (Holland & Bouton, 1999; Quinn et al., 2008).

Notably, the correlation of IPFC and AMY context-specificity overlapped with the period of IPFC context specificity (from 0.8 to 1.15s after cue onset). However, it also shows a delay between IPFC and AMY, with IPFC context specificity emerging ~1s earlier. This temporal lag seems to decrease during later trial periods. The main effect of IPFC context-specificity occurs immediately prior to the time period when the strongest coordination is observed with the AMY (at approximately 1.2s). Although the AMY did not show a main

effect of context specificity—possibly because the similarity of same-context representations was attenuated by repetition suppression—the observed temporal offset suggests that the effect observed in the IPFC drives the AMY representations. Our findings underscore the relevance of dynamic and distributed context representations throughout the extinction network during fear extinction. More specifically, they suggest that AMY safety signals and representations during extinction depend on the preceding representation of specific contexts in IPFC.”

Coordination of context specificity in IPFC and AMY. We correlated context specificity values between the IPFC and the AMY across trials. Significant regions surviving multiple comparisons corrections using cluster-based permutation statistics at an alpha level of 0.05 are outlined in black, and at an alpha level of 0.01 are outlined in black dashed lines. The diagonal of the temporal generalization map indicates matched time points across trials. The time period of significant IPFC context specificity (Figure 4C, middle) is marked in red over the diagonal. At an alpha level of 0.05, the coordination is significant with a p-value of 0.001 (indicated with *** on the plot).

7. For all analyses looking at correlations of item or context stability between brain regions, it is also not clear how these correlations may or may not be related to fear contingencies. They appear to perform these analyses by looking at all trials, but do these differ if we look at CS+ versus CS- trials, for example, or CS++ versus CS+-.

We initially opted for a more general analysis including all trials within a given experimental phase to maximize statistical power in our correlation analyses. This was particularly important given that not all regions exhibited a significant effect in the metrics studied (item stability and context specificity). Since these correlations were conducted across trials, separately analyzing differences across conditions substantially reduced the number of trials (total number of trials: 72, i.e. 24 in each condition). However, we agree with the reviewer that it is valuable to investigate whether the coordination of representational signals differs across conditions.

We computed item stability and context specificity separately for each condition, focusing on the ROIs where we observed significant effects in the analysis including all trials during extinction. We then assessed whether coordination differed between CS+ vs CS- trials, and between CS++ vs CS+- trials, as suggested by the reviewer. We performed the analysis across all time points in the temporal generalization map and assessed statistical significance via cluster-based permutation statistics, shuffling the condition labels 1,000 times. We only considered significant those time-by-time clusters whose summed t-values ranked above the 95th percentile of the null distribution.

In most analyses, we did not observe large clusters in the temporal generalization maps, as observed in the resulting p-values presented in the table below (corresponding to the rank of these clusters within the distribution of surrogate cluster). Thus, while our main results in the manuscript indicate that item stability differed between experimental conditions in the TMP and the AMY (Figure 3), the *coordination* of item stability across brain regions did not differ between conditions. We report these new results in the correspondent sections on item stability (lines 285-287) and context specificity (lines 330-331) of the revised manuscript, and in the new Supplementary Note 1.

Analysis	Trial	CS+ vs CS-	CS++ vs CS+-
	Region		
Item stability	TMP-AMY	N= 22, p = 0.69	N = 19, p = 0.69
	TMP-HPC	N= 23, p = 0.8	N = 21, p = 0.38
	TMP-IPFC	N= 17, p = 0.51	N = 16, p = 0.81
	TPM-OFC	N= 9, p = 0.59	N = 9, p = 0.93
	AMY-HPC	N= 24, p = 0.34	N = 23, p = 0.86
Context specificity	IPFC-AMY	N = 14, p = 0.91	N = 14, p = 0.95
	IPFC-TMP	N= 17, p = 0.21	N = 16, p = 0.64

Summary table presents the results of the contrasts between CS+ and CS- trials, as well as between CS++ and CS+- trials, in the analyses of the coordination of both item stability and context specificity between regions. Note that the analyses were conducted at all time points in the temporal generalization maps, and the reported p-values reflect the rank of the largest observed cluster in relation to a null distribution constructed by shuffling the condition labels. The number of participants (N) included in each analysis is also indicated.

8. The authors also find a negative correlation between amygdala theta power and item stability in the amygdala and hippocampus. But again, it is hard to understand how to interpret this without specifically knowing how such item and context stability change between conditions, and also how theta power changes between CS++ and CS+- conditions.

We have now contrasted AMY theta power for CS++ and CS+- items during extinction and observed significantly higher theta power for CS+- trials (see our response to point 2). Similarly, we compared item stability across conditions and found that it is higher for CS++ items as compared to CS+- items in both AMY and TMP (see our response to point 3).

To assess differences in AMY_{THETA} and item stability and context specificity correlations across conditions, we extracted a single-trial metric of AMY_{THETA} based on the cluster of significant differences between CS+ and CS- items observed during extinction (Figure 2; see as schematic depiction in Figure 5A, left). We separately z-scored this metric for CS+ and CS- trials, in order to avoid any spurious correlation of single-trial values driven by main condition differences. In addition, we averaged item stability and context specificity across time in the time period of AMY_{THETA} effects in each trial and each ROI. We correlated AMY theta power with item stability and context specificity across trials, separately in our three trial types (CS++, CS+- and CS--), and assessed condition differences using a one-way repeated measures ANOVA.

Our results did not reveal any significant difference between conditions in the AMY ($F(2, 44) = 1.15, p = 0.32$). Post-hoc tests revealed that none of the pairwise comparisons showed differences, even at an uncorrected level (all $p > 0.19$). Similarly, in the HPC, correlations did not differ between conditions ($F(2, 34) = 0.2, p = 0.82$) and pairwise correlations were not significantly different (all $p > 0.49$). Finally, the analysis of IPFC context specificity revealed no significant differences between conditions ($F(2, 22) = 0.91, p = 0.42$), and no significant differences in the pairwise post hoc tests (all $p > 0.2$).

These results demonstrate that the observed trial-level correlations between AMY_{THETA} and item stability in AMY and HPC did not differ across trial types. In addition, they show that the correlations between AMY_{THETA} and IPFC context specificity are not significantly different across conditions. Together, these findings suggest that AMY theta oscillations play a role in the representation of cues and contexts throughout the extinction network irrespective of their valence. We present these results in the new Supplementary Note 2.

9. Finally, the authors deploy a clever reinstatement analysis examining the similarity of items between acquisition and test and between extinction and test. They find a positive relation between context representations in the PFC and reinstatement in the TMP for CS+- trials. However, to make this analysis clearer, one would also like to know if there is a difference in reinstatement between CS++ and CS+- and CS-- trials at all. This seems like it would provide a more direct test regarding whether the item representations during acquisition and/or extinction are reinstated during test, with the hypothesis being that there will be significant differences in reinstatement between the different categories.

We thank the Reviewer for this suggestion. We compared the reinstatement values between our three trial types (CS++, CS+-, CS--), separately for acquisition-to-test and extinction-to-test reinstatement, in the brain regions where we observed significantly higher item-stability for the CS+ as compared to the CS- trials during extinction, i.e., the AMY and the TMP. In addition, we compared the differential reinstatement of memory traces (acquisition-to-test minus extinction-to-test) across these conditions. In a separate analysis, we investigated whether the amount of acquisition-to-test reinstatement differed from the amount of extinction-to-test reinstatement for any of the categories.

For the first analysis, we assessed possible condition effects on reinstatement during time windows of 500ms during acquisition/extinction and test, sliding in 50ms (90% overlap). Only matching time-windows were considered for this analysis because a temporal generalization analysis would not be possible for the differential reinstatement, which involves time periods during three phases (acquisition, extinction, and test). We averaged across trials and across time within each 500ms window for each participant and performed one-way repeated-measures ANOVAs at each time bin to assess statistical significance. In the AMY, we did not observe any significant differences between our three conditions in the acquisition-to-test reinstatement (all time bins: $F < 1.73$, $p > 0.18$), the extinction-to-test reinstatement (all time bins: $F < 2.8$, all $p > 0.06$) or in the differential reinstatement analysis (all time bins: $F < 2.54$, all $p > 0.08$), even before applying correction for multiple comparisons. Similarly, in the TMP, we found no significant condition differences, even at an uncorrected level, for acquisition-to-test reinstatement (all time bins: $F < 1.4$, $p > 0.26$), extinction-to-test reinstatement (all time bins: $F < 2.53$, $p > 0.089$) or in the differential reinstatement analysis (all time bins: $F < 2.42$, $p > 0.09$; see Figure below). These results suggest that reinstatement does not differ between conditions.

In the second analysis, we compared the magnitude of acquisition-to-test and extinction-to-test reinstatement separately for the CS++, the CS+- and the CS-- trials. We focused on the time periods where significant item-stability effects were observed in each region (AMY: 1.25-1.5s; TMP: 0.65-1s). In the TMP, none of the cue types showed significant differences between acquisition-to-test and extinction-to-test reinstatement (CS++: $t(25) = -1.11$; $p = 0.28$; CS+-: $t(25) = 0.647$; $p = 0.52$; CS--: $t(25) = 1.59$; $p = 0.12$). In the AMY, we observed no significant differences in CS++ ($t(30) = -0.092$; $p = 0.93$) or CS-- trials ($t(30) = -0.352$; $p = 0.728$). However, for the CS+- trials, we observed a trend toward higher levels of acquisition-to-test reinstatement compared to extinction-to-test reinstatement ($t(30) = 1.72$; $p = 0.096$).

These latter findings suggest that in the AMY, fear memories of CS+- items might show slightly higher levels of reinstatement than the subsequent extinction memories of these items, an effect we did not observe for CS++ or CS-- items. In general, however, levels of reinstatement were relatively similar between the three cue categories—suggesting

that reinstatement may be influenced by factors other than the contingency of items. Theories on the psychopathological mechanisms of anxiety disorders have proposed context specificity during extinction as a major factor influencing the balance between the renewal of fear memories vs. the retrieval of extinction memories (Beckers et al., 2023; Lebois et al., 2019; Wang et al., 2024). For this reason, we hypothesized that context specificity would specifically influence differential reinstatement in CS+- trials, whose contingencies change between acquisition and extinction.

We have added the results of these new analyses in the new Supplementary Note 3 and Supplementary Figure 1.

.

Supplementary Figure 1. Reinstatement of acquisition and extinction memory traces do not differ across trial types. Acquisition-to-test (left), extinction-to-test and differential reinstatement (acquisition minus extinction-to-test) did not differ between our three trial types (CS++, CS+-, CS--).

10. In several places in the text, the references to the specific figure and figure panels seem incorrect, e.g. pointing to the wrong figure panel.

Thanks for noticing. We have checked carefully and corrected all issues related to the referencing of figures and figure panels.

References

- Axmacher, N. (2023). *Intracranial EEG: A Guide for Cognitive Neuroscientists*. Springer International Publishing.
- Baas, J. M., Nugent, M., Lissek, S., Pine, D. S., & Grillon, C. (2004). Fear conditioning in virtual reality contexts: A new tool for the study of anxiety. *Biological Psychiatry*, *55*(11), 1056–1060. <https://doi.org/10.1016/j.biopsych.2004.02.024>
- Balooch, S. B., Neumann, D. L., & Boschen, M. J. (2012). Extinction treatment in multiple contexts attenuates ABC renewal in humans. *Behaviour Research and Therapy*, *50*(10), 604–609. <https://doi.org/10.1016/j.brat.2012.06.003>
- Bandarian Balooch, S., & Neumann, D. L. (2011). Effects of multiple contexts and context similarity on the renewal of extinguished conditioned behaviour in an ABA design with humans. *Learning and Motivation*, *42*(1), 53–63. <https://doi.org/10.1016/j.lmot.2010.08.008>
- Battaglia, S., Garofalo, S., & di Pellegrino, G. (2018). Context-dependent extinction of threat memories: Influences of healthy aging. *Scientific Reports*, *8*(1), 12592. <https://doi.org/10.1038/s41598-018-31000-9>
- Beckers, T., Hermans, D., Lange, I., Luyten, L., Scheveneels, S., & Vervliet, B. (2023). Understanding clinical fear and anxiety through the lens of human fear conditioning. *Nature Reviews Psychology*, *2*(4), 233–245. <https://doi.org/10.1038/s44159-023-00156-1>
- Bernabei, J. M., Li, A., Revell, A. Y., Smith, R. J., Gunnarsdottir, K. M., Ong, I. Z., Davis, K. A., Sinha, N., Sarma, S., & Litt, B. (2023). Quantitative approaches to guide epilepsy surgery from intracranial EEG. *Brain*, *146*(6), 2248–2258. <https://doi.org/10.1093/brain/awad007>
- Bierwirth, P., Antov, M. I., & Stockhorst, U. (2023). Oscillatory and non-oscillatory brain activity reflects fear expression in an immediate and delayed fear extinction task. *Psychophysiology*, *60*(8), e14283. <https://doi.org/10.1111/psyp.14283>
- Bierwirth, P., Sperl, M. F. J., Antov, M. I., & Stockhorst, U. (2021). Prefrontal Theta Oscillations Are Modulated by Estradiol Status During Fear Recall and Extinction Recall. *Biological Psychiatry: Cognitive Neuroscience and Neuroimaging*, *6*(11), 1071–1080. <https://doi.org/10.1016/j.bpsc.2021.02.011>
- Bocchio, M., Nabavi, S., & Capogna, M. (2017). Synaptic Plasticity, Engrams, and Network Oscillations in Amygdala Circuits for Storage and Retrieval of Emotional Memories. *Neuron*, *94*(4), 731–743. <https://doi.org/10.1016/j.neuron.2017.03.022>

- Boddez, Y., Baeyens, F., Luyten, L., Vansteenwegen, D., Hermans, D., & Beckers, T. (2013). Rating data are underrated: Validity of US expectancy in human fear conditioning. *Journal of Behavior Therapy and Experimental Psychiatry*, *44*(2), 201–206. <https://doi.org/10.1016/j.jbtep.2012.08.003>
- Bouton, M. E. (2004). Context and behavioral processes in extinction. *Learning & Memory (Cold Spring Harbor, N.Y.)*, *11*(5), 485–494. <https://doi.org/10.1101/lm.78804>
- Bouyeure, A., Pacheco, D., Fellner, M.-C., Jacob, G., Kobelt, M., Rose, J., & Axmacher, N. (2024). Distinct representational properties of cues and contexts shape fear learning and extinction. *bioRxiv*, 2024.12.16.628638. <https://doi.org/10.1101/2024.12.16.628638>
- Buzsáki, G. (2002). Theta Oscillations in the Hippocampus. *Neuron*, *33*(3), 325–340. [https://doi.org/10.1016/S0896-6273\(02\)00586-X](https://doi.org/10.1016/S0896-6273(02)00586-X)
- Cattani, A., Arnold, D. B., McCarthy, M., & Kopell, N. (2024). Basolateral amygdala oscillations enable fear learning in a biophysical model. *eLife*, *12*, RP89519. <https://doi.org/10.7554/eLife.89519>
- Chen, S., Tan, Z., Xia, W., Gomes, C. A., Zhang, X., Zhou, W., Liang, S., Axmacher, N., & Wang, L. (2021). Theta oscillations synchronize human medial prefrontal cortex and amygdala during fear learning. *Science Advances*, *7*(34), eabf4198. <https://doi.org/10.1126/sciadv.abf4198>
- Chen, Y., Aponik-Gremillion, L., Bartoli, E., Yoshor, D., Sheth, S. A., & Foster, B. L. (2021). Stability of ripple events during task engagement in human hippocampus. *Cell Reports*, *35*(13), 109304. <https://doi.org/10.1016/j.celrep.2021.109304>
- Cichy, R. M., Pantazis, D., & Oliva, A. (2014). Resolving human object recognition in space and time. *Nature Neuroscience*, *17*(3), 455–462. <https://doi.org/10.1038/nn.3635>
- Constantinou, E., Purves, K. L., McGregor, T., Lester, K. J., Barry, T. J., Treanor, M., Craske, M. G., & Eley, T. C. (2021). Measuring fear: Association among different measures of fear learning. *Journal of Behavior Therapy and Experimental Psychiatry*, *70*, 101618. <https://doi.org/10.1016/j.jbtep.2020.101618>
- Corcoran, K. A., & Maren, S. (2001). Hippocampal Inactivation Disrupts Contextual Retrieval of Fear Memory after Extinction. *The Journal of Neuroscience*, *21*(5), 1720. <https://doi.org/10.1523/JNEUROSCI.21-05-01720.2001>
- Corcoran, K. A., & Maren, S. (2004). Factors regulating the effects of hippocampal inactivation on renewal of conditional fear after extinction. *Learning & Memory*, *11*(5), 598–603.

Corneille, O., & Gawronski, B. (2024). Self-reports are better measurement instruments than implicit measures. *Nature Reviews Psychology*, 3(12), 835–846. <https://doi.org/10.1038/s44159-024-00376-z>

Costa, M., Pacheco, D., Gil-Nagel, A., Toledano, R., Imbach, L., Sarnthein, J., & Strange, B. A. (2024). Retrieval of human aversive memories involves reactivation of gamma activity patterns in the hippocampus that originate in the amygdala during encoding. *bioRxiv*, 2024.01.18.576178. <https://doi.org/10.1101/2024.01.18.576178>

Courtin, J., Chaudun, F., Rozeske, R. R., Karalis, N., Gonzalez-Campo, C., Wurtz, H., Abdi, A., Baufreton, J., Bienvenu, T. C. M., & Herry, C. (2014). Prefrontal parvalbumin interneurons shape neuronal activity to drive fear expression. *Nature*, 505(7481), 92–96. <https://doi.org/10.1038/nature12755>

Craske, M. G., Treanor, M., Conway, C. C., Zbozinek, T., & Vervliet, B. (2014). Maximizing exposure therapy: An inhibitory learning approach. *Behaviour Research and Therapy*, 58, 10–23. <https://doi.org/10.1016/j.brat.2014.04.006>

Dalal, S. S., Zumer, J. M., Guggisberg, A. G., Trumpis, M., Wong, D. D., Sekihara, K., & Nagarajan, S. S. (2011). MEG/EEG source reconstruction, statistical evaluation, and visualization with NUTMEG. *Computational Intelligence and Neuroscience*, 2011(1), 758973.

Davis, P., & Reijmers, L. G. (2018). The dynamic nature of fear engrams in the basolateral amygdala. *Memory Mechanisms in Health and Disease*, 141, 44–49. <https://doi.org/10.1016/j.brainresbull.2017.12.004>

Desikan, R. S., Ségonne, F., Fischl, B., Quinn, B. T., Dickerson, B. C., Blacker, D., Buckner, R. L., Dale, A. M., Maguire, R. P., Hyman, B. T., Albert, M. S., & Killiany, R. J. (2006). An automated labeling system for subdividing the human cerebral cortex on MRI scans into gyral based regions of interest. *NeuroImage*, 31(3), 968–980. <https://doi.org/10.1016/j.neuroimage.2006.01.021>

Dimakopoulos, V., Gotman, J., Stacey, W., von Ellenrieder, N., Jacobs, J., Papadelis, C., Cimbalnik, J., Worrell, G., Sperling, M. R., Zijlmans, M., Imbach, L., Frauscher, B., & Sarnthein, J. (2022). Protocol for multicentre comparison of interictal high-frequency oscillations as a predictor of seizure freedom. *Brain Communications*, 4(3), fcac151. <https://doi.org/10.1093/braincomms/fcac151>

Dunsmoor, J. E., Ahs, F., Zielinski, D. J., & LaBar, K. S. (2014). Extinction in multiple virtual reality contexts diminishes fear reinstatement in humans. *Extinction*, 113, 157–164. <https://doi.org/10.1016/j.nlm.2014.02.010>

Dunsmoor, J. E., Bandettini, P. A., & Knight, D. C. (2007). Impact of continuous versus intermittent CS-UCS pairing on human brain activation during Pavlovian fear conditioning. *Behavioral Neuroscience*, *121*(4), 635.

Dunsmoor, J. E., Martin, A., & LaBar, K. S. (2012). Role of conceptual knowledge in learning and retention of conditioned fear. *Biological Psychology*, *89*(2), 300–305. <https://doi.org/10.1016/j.biopsycho.2011.11.002>

Dunsmoor, J. E., & Murphy, G. L. (2014). Stimulus Typicality Determines How Broadly Fear Is Generalized. *Psychological Science*, *25*(9), 1816–1821. <https://doi.org/10.1177/0956797614535401>

Effting, M., & Kindt, M. (2007). Contextual control of human fear associations in a renewal paradigm. *Behaviour Research and Therapy*, *45*(9), 2002–2018. <https://doi.org/10.1016/j.brat.2007.02.011>

Fell, J., & Axmacher, N. (2011). The role of phase synchronization in memory processes. *Nature Reviews. Neuroscience*, *12*(2), 105–118. <https://doi.org/10.1038/nrn2979>

Fenton, G. E., Halliday, D. M., Mason, R., & Stevenson, C. W. (2014). Medial prefrontal cortex circuit function during retrieval and extinction of associative learning under anesthesia. *Neuroscience*, *265*, 204–216. <https://doi.org/10.1016/j.neuroscience.2014.01.028>

Fischl, B. (2012). FreeSurfer. *20 YEARS OF fMRI*, *62*(2), 774–781. <https://doi.org/10.1016/j.neuroimage.2012.01.021>

Fullana, M. A., Harrison, B. J., Soriano-Mas, C., Vervliet, B., Cardoner, N., Àvila-Parcet, A., & Radua, J. (2016). Neural signatures of human fear conditioning: An updated and extended meta-analysis of fMRI studies. *Molecular Psychiatry*, *21*(4), 500–508. <https://doi.org/10.1038/mp.2015.88>

Garcia, J., & Koelling, R. A. (1966). Relation of cue to consequence in avoidance learning. *Psychonomic Science*, *4*(3), 123–124. <https://doi.org/10.3758/BF03342209>

Garfinkel, S. N., Abelson, J. L., King, A. P., Sripada, R. K., Wang, X., Gaines, L. M., & Liberzon, I. (2014). Impaired Contextual Modulation of Memories in PTSD: An fMRI and Psychophysiological Study of Extinction Retention and Fear Renewal. *The Journal of Neuroscience*, *34*(40), 13435. <https://doi.org/10.1523/JNEUROSCI.4287-13.2014>

Gilmarin, M. R., Balderston, N. L., & Helmstetter, F. J. (2014). Prefrontal cortical regulation of fear learning. *Trends in Neurosciences*, *37*(8), 455–464. <https://doi.org/10.1016/j.tins.2014.05.004>

Glenn, C. R., Klein, D. N., Lissek, S., Britton, J. C., Pine, D. S., & Hajcak, G. (2012). The development of fear learning and generalization in 8–13 year-olds. *Developmental Psychobiology*, *54*(7), 675–684. <https://doi.org/10.1002/dev.20616>

Glenn, C. R., Lieberman, L., & Hajcak, G. (2012). Comparing electric shock and a fearful screaming face as unconditioned stimuli for fear learning. *International Journal of Psychophysiology*, *86*(3), 214–219. <https://doi.org/10.1016/j.ijpsycho.2012.09.006>

Glotzbach-Schoon, E., Tadda, R., Andreatta, M., Tröger, C., Ewald, H., Grillon, C., Pauli, P., & Mühlberger, A. (2013). Enhanced discrimination between threatening and safe contexts in high-anxious individuals. *Biological Psychology*, *93*(1), 159–166. <https://doi.org/10.1016/j.biopsycho.2013.01.011>

Hamm, A. O., Vaitl, D., & Lang, P. J. (1989). Fear conditioning, meaning, and belongingness: A selective association analysis. *Journal of Abnormal Psychology*, *98*(4), 395–406. <https://doi.org/10.1037/0021-843X.98.4.395>

Henin, S., Shankar, A., Borges, H., Flinker, A., Doyle, W., Friedman, D., Devinsky, O., Buzsáki, G., & Liu, A. (2021). Spatiotemporal dynamics between interictal epileptiform discharges and ripples during associative memory processing. *Brain*, *144*(5), 1590–1602. <https://doi.org/10.1093/brain/awab044>

Hermann, A., Stark, R., Milad, M., & Merz, C. (2016). Renewal of conditioned fear in a novel context is associated with hippocampal activation and connectivity. *Social Cognitive and Affective Neuroscience*, *11*(9), 1411–1421.

Herweg, N. A., Solomon, E. A., & Kahana, M. J. (2020). Theta Oscillations in Human Memory. *Trends in Cognitive Sciences*, *24*(3), 208–227. <https://doi.org/10.1016/j.tics.2019.12.006>

Imperatori, L. S., Betta, M., Cecchetti, L., Canales-Johnson, A., Ricciardi, E., Siclari, F., Pietrini, P., Chennu, S., & Bernardi, G. (2019). EEG functional connectivity metrics wPLI and wSMI account for distinct types of brain functional interactions. *Scientific Reports*, *9*(1), 8894. <https://doi.org/10.1038/s41598-019-45289-7>

Jacobs, J. (2014). Hippocampal theta oscillations are slower in humans than in rodents: Implications for models of spatial navigation and memory. *Philosophical Transactions of the Royal Society B: Biological Sciences*, *369*(1635), 20130304. <https://doi.org/10.1098/rstb.2013.0304>

Kalisch, R., Korenfeld, E., Stephan, K. E., Weiskopf, N., Seymour, B., & Dolan, R. J. (2006). Context-Dependent Human Extinction Memory Is Mediated by a Ventromedial Prefrontal and Hippocampal Network. *The Journal of Neuroscience*, *26*(37), 9503. <https://doi.org/10.1523/JNEUROSCI.2021-06.2006>

- Karalis, N., Dejean, C., Chaudun, F., Khoder, S., Rozeske, R. R., Wurtz, H., Bagur, S., Benchenane, K., Sirota, A., Courtin, J., & Herry, C. (2016). 4-Hz oscillations synchronize prefrontal–amygdala circuits during fear behavior. *Nature Neuroscience*, *19*(4), 605–612. <https://doi.org/10.1038/nn.4251>
- King, J.-R., & Dehaene, S. (2014). Characterizing the dynamics of mental representations: The temporal generalization method. *Trends in Cognitive Sciences*, *18*(4), 203–210.
- Kinner, V. L., Merz, C. J., Lissek, S., & Wolf, O. T. (2016). Cortisol disrupts the neural correlates of extinction recall. *NeuroImage*, *133*, 233–243. <https://doi.org/10.1016/j.neuroimage.2016.03.005>
- Knight, D. C., Nguyen, H. T., & Bandettini, P. A. (2003). Expression of conditional fear with and without awareness. *Proceedings of the National Academy of Sciences*, *100*(25), 15280–15283. <https://doi.org/10.1073/pnas.2535780100>
- Knight, D. C., Waters, N. S., & Bandettini, P. A. (2009). Neural substrates of explicit and implicit fear memory. *NeuroImage*, *45*(1), 208–214. <https://doi.org/10.1016/j.neuroimage.2008.11.015>
- Kobelt, M., Waldhauser, G., Rupietta, A., Heinen, R., Rau, E., Kessler, H., & Axmacher, N. (2024). The memory trace of an intrusive trauma-analog episode. *Current Biology*, *34*(8), 1657–1669.
- Kriegeskorte, N., Mur, M., & Bandettini, P. A. (2008). Representational similarity analysis-connecting the branches of systems neuroscience. *Frontiers in Systems Neuroscience*, *4*.
- Krisch, K. A., Bandarian-Balooch, S., & Neumann, D. L. (2018). Effects of extended extinction and multiple extinction contexts on ABA renewal. *Learning and Motivation*, *63*, 1–10. <https://doi.org/10.1016/j.lmot.2017.11.001>
- Lachaux, J.-P., Rodriguez, E., Martinerie, J., & Varela, F. J. (1999). Measuring phase synchrony in brain signals. *Human Brain Mapping*, *8*(4), 194–208. [https://doi.org/10.1002/\(SICI\)1097-0193\(1999\)8:4<194::AID-HBM4>3.0.CO;2-C](https://doi.org/10.1002/(SICI)1097-0193(1999)8:4<194::AID-HBM4>3.0.CO;2-C)
- Lebois, L. A., Seligowski, A. V., Wolff, J. D., Hill, S. B., & Ressler, K. J. (2019). Augmentation of extinction and inhibitory learning in anxiety and trauma-related disorders. *Annual Review of Clinical Psychology*, *15*(1), 257–284.
- Lehongre, K., Lambrecq, V., Whitmarsh, S., Frazzini, V., Cousyn, L., Soleil, D., Fernandez-Vidal, S., Mathon, B., Houot, M., Lemaréchal, J.-D., Clemenceau, S., Hasboun, D., Adam, C., & Navarro, V. (2022). Long-term deep intracerebral microelectrode recordings in patients with drug-resistant epilepsy: Proposed guidelines

based on 10-year experience. *NeuroImage*, 254, 119116.
<https://doi.org/10.1016/j.neuroimage.2022.119116>

Lesting, J., Daldrup, T., Narayanan, V., Himpe, C., Seidenbecher, T., & Pape, H.-C. (2013). Directional Theta Coherence in Prefrontal Cortical to Amygdalo-Hippocampal Pathways Signals Fear Extinction. *PLOS ONE*, 8(10), e77707.
<https://doi.org/10.1371/journal.pone.0077707>

Lesting, J., Narayanan, R. T., Kluge, C., Sangha, S., Seidenbecher, T., & Pape, H.-C. (2011). Patterns of Coupled Theta Activity in Amygdala-Hippocampal-Prefrontal Cortical Circuits during Fear Extinction. *PLOS ONE*, 6(6), e21714.
<https://doi.org/10.1371/journal.pone.0021714>

Likhtik, E., Stujenske, J. M., Topiwala, M., Harris, A. Z., & Gordon, J. A. (2014). Prefrontal entrainment of amygdala activity signals safety in learned fear and innate anxiety. *Nature Neuroscience*, 17(1), 106–113. <https://doi.org/10.1038/nn.3582>

Lissek, S., Biggs, A. L., Rabin, S. J., Cornwell, B. R., Alvarez, R. P., Pine, D. S., & Grillon, C. (2008). Generalization of conditioned fear-potentiated startle in humans: Experimental validation and clinical relevance. *Behaviour Research and Therapy*, 46(5), 678–687. <https://doi.org/10.1016/j.brat.2008.02.005>

Lissek, S., & Tegenthoff, M. (2024). Dissimilarities of neural representations of extinction trials are associated with extinction learning performance and renewal level. *Frontiers in Behavioral Neuroscience*, 18, 1307825.

Lonsdorf, T. B., Menz, M. M., Andreatta, M., Fullana, M. A., Golkar, A., Haaker, J., Heitland, I., Hermann, A., Kuhn, M., Kruse, O., Meir Drexler, S., Meulders, A., Nees, F., Pittig, A., Richter, J., Römer, S., Shiban, Y., Schmitz, A., Straube, B., ... Merz, C. J. (2017). Don't fear 'fear conditioning': Methodological considerations for the design and analysis of studies on human fear acquisition, extinction, and return of fear. *Neuroscience & Biobehavioral Reviews*, 77, 247–285.
<https://doi.org/10.1016/j.neubiorev.2017.02.026>

Lopour, B. A., Tavassoli, A., Fried, I., & Ringach, D. L. (2013). Coding of information in the phase of local field potentials within human medial temporal lobe. *Neuron*, 79(3), 594–606. <https://doi.org/10.1016/j.neuron.2013.06.001>

Lu, Y., Wang, C., Chen, C., & Xue, G. (2015). Spatiotemporal Neural Pattern Similarity Supports Episodic Memory. *Current Biology*, 25(6), 780–785.
<https://doi.org/10.1016/j.cub.2015.01.055>

Ma, Y., Kyuchukova, D., Jiao, F., Batsikadze, G., Nitsche, M. A., & Yavari, F. (2025). The impact of temporal distribution on fear extinction learning. *International Journal of Clinical and Health Psychology*, 25(1), 100536.

- Maguire, E. A., & Mullanly, S. L. (2013). The hippocampus: A manifesto for change. *Journal of Experimental Psychology: General*, *142*(4), 1180.
- Maren, S., Phan, K. L., & Liberzon, I. (2013). The contextual brain: Implications for fear conditioning, extinction and psychopathology. *Nature Reviews Neuroscience*, *14*(6), 417–428. <https://doi.org/10.1038/nrn3492>
- Maris, E., & Oostenveld, R. (2007). Nonparametric statistical testing of EEG- and MEG-data. *Journal of Neuroscience Methods*, *164*(1), 177–190. <https://doi.org/10.1016/j.jneumeth.2007.03.024>
- McClelland, J. L., McNaughton, B. L., & O'Reilly, R. C. (1995). Why there are complementary learning systems in the hippocampus and neocortex: Insights from the successes and failures of connectionist models of learning and memory. *Psychological Review*, *102*(3), 419–457. <https://doi.org/10.1037/0033-295X.102.3.419>
- Méndez-Bértolo, C., Moratti, S., Toledano, R., Lopez-Sosa, F., Martínez-Alvarez, R., Mah, Y. H., Vuilleumier, P., Gil-Nagel, A., & Strange, B. A. (2016). A fast pathway for fear in human amygdala. *Nature Neuroscience*, *19*(8), 1041–1049.
- Michelmann, S., Bowman, H., & Hanslmayr, S. (2016). The Temporal Signature of Memories: Identification of a General Mechanism for Dynamic Memory Replay in Humans. *PLOS Biology*, *14*(8), e1002528. <https://doi.org/10.1371/journal.pbio.1002528>
- Milad, M. R., Orr, S. P., Pitman, R. K., & Rauch, S. L. (2005). Context modulation of memory for fear extinction in humans. *Psychophysiology*, *42*(4), 456–464.
- Milad, M. R., & Quirk, G. J. (2012). Fear extinction as a model for translational neuroscience: Ten years of progress. *Annual Review of Psychology*, *63*(1), 129–151.
- Milad, M. R., Wright, C. I., Orr, S. P., Pitman, R. K., Quirk, G. J., & Rauch, S. L. (2007). Recall of Fear Extinction in Humans Activates the Ventromedial Prefrontal Cortex and Hippocampus in Concert. *Biological Psychiatry*, *62*(5), 446–454. <https://doi.org/10.1016/j.biopsych.2006.10.011>
- Miller, J. F., Neufang, M., Solway, A., Brandt, A., Trippel, M., Mader, I., Hefft, S., Merkow, M., Polyn, S. M., Jacobs, J., Kahana, M. J., & Schulze-Bonhage, A. (2013). Neural Activity in Human Hippocampal Formation Reveals the Spatial Context of Retrieved Memories. *Science*, *342*(6162), 1111–1114. <https://doi.org/10.1126/science.1244056>
- Mineka, S., & Öhman, A. (2002). Phobias and preparedness: The selective, automatic, and encapsulated nature of fear. *Biological Psychiatry*, *52*(10), 927–937. [https://doi.org/10.1016/S0006-3223\(02\)01669-4](https://doi.org/10.1016/S0006-3223(02)01669-4)
- Mormann, F., Lehnertz, K., David, P., & Elger, C. (2000). Mean phase coherence as a measure for phase synchronization and its application to the EEG of epilepsy patients.

Physica D: Nonlinear Phenomena, 144(3), 358–369. [https://doi.org/10.1016/S0167-2789\(00\)00087-7](https://doi.org/10.1016/S0167-2789(00)00087-7)

Mueller, E. M., Panitz, C., Hermann, C., & Pizzagalli, D. A. (2014). Prefrontal Oscillations during Recall of Conditioned and Extinguished Fear in Humans. *The Journal of Neuroscience*, 34(21), 7059. <https://doi.org/10.1523/JNEUROSCI.3427-13.2014>

Neumann, D. L., & Kitlertsirivatana, E. (2010). Exposure to a novel context after extinction causes a renewal of extinguished conditioned responses: Implications for the treatment of fear. *Behaviour Research and Therapy*, 48(6), 565–570. <https://doi.org/10.1016/j.brat.2010.03.002>

Neumann, D. L., Lipp, O. V., & Cory, S. E. (2007). Conducting extinction in multiple contexts does not necessarily attenuate the renewal of shock expectancy in a fear-conditioning procedure with humans. *Behaviour Research and Therapy*, 45(2), 385–394. <https://doi.org/10.1016/j.brat.2006.02.001>

Ney, L. J., O'Donohue, M. P., Lowe, B. G., & Lipp, O. V. (2022). Angry and fearful compared to happy or neutral faces as conditional stimuli in human fear conditioning: A systematic review and meta-analysis. *Neuroscience & Biobehavioral Reviews*, 139, 104756. <https://doi.org/10.1016/j.neubiorev.2022.104756>

Nguyen, R., Koukoutselos, K., Forro, T., & Ciochi, S. (2023). Fear extinction relies on ventral hippocampal safety codes shaped by the amygdala. *Science Advances*, 9(22), eadg4881. <https://doi.org/10.1126/sciadv.adg4881>

Nolte, G., Ziehe, A., Nikulin, V. V., Schlögl, A., Krämer, N., Brismar, T., & Müller, K.-R. (2008). Robustly estimating the flow direction of information in complex physical systems. *Physical Review Letters*, 100(23), 234101.

Oehr, C. R., Baumann, C., Fell, J., Lee, H., Kessler, H., Habel, U., Hanslmayr, S., & Axmacher, N. (2015). Human Hippocampal Dynamics during Response Conflict. *Current Biology*, 25(17), 2307–2313. <https://doi.org/10.1016/j.cub.2015.07.032>

Öhman, A., & Öst, L.-G. (1985). Animal and social phobias: Biological constraints on learned fear responses. *Theoretical Issues in Behavior Therapy*, 123–175.

Pacheco Estefan, D., Sánchez-Fibla, M., Duff, A., Principe, A., Rocamora, R., Zhang, H., Axmacher, N., & Verschure, P. F. M. J. (2019). Coordinated representational reinstatement in the human hippocampus and lateral temporal cortex during episodic memory retrieval. *Nature Communications*, 10(1), 1–13.

Pacheco Estefan, D., Zucca, R., Arsiwalla, X., Principe, A., Zhang, H., Rocamora, R., Axmacher, N., & Verschure, P. F. M. J. (2021). Volitional learning promotes theta phase

coding in the human hippocampus. *Proceedings of the National Academy of Sciences*, *118*(10).

Pacheco-Estefan, D., Fellner, M.-C., Kunz, L., Zhang, H., Reinacher, P., Roy, C., Brandt, A., Schulze-Bonhage, A., Yang, L., Wang, S., Liu, J., Xue, G., & Axmacher, N. (2024). Maintenance and transformation of representational formats during working memory prioritization. *Nature Communications*, *15*(1), 8234. <https://doi.org/10.1038/s41467-024-52541-w>

Paré, D., & Collins, D. R. (2000). Neuronal Correlates of Fear in the Lateral Amygdala: Multiple Extracellular Recordings in Conscious Cats. *The Journal of Neuroscience*, *20*(7), 2701. <https://doi.org/10.1523/JNEUROSCI.20-07-02701.2000>

Parvizi, J., & Kastner, S. (2018). Promises and limitations of human intracranial electroencephalography. *Nature Neuroscience*, *21*(4), 474–483.

Pieper, S., Halle, M., & Kikinis, R. (2004). *3D Slicer*. 632–635.

Pirazzini, G., Starita, F., Ricci, G., Garofalo, S., di Pellegrino, G., Magosso, E., & Ursino, M. (2023). Changes in brain rhythms and connectivity tracking fear acquisition and reversal. *Brain Structure and Function*, *228*(5), 1259–1281. <https://doi.org/10.1007/s00429-023-02646-7>

Popescu, A. T., Popa, D., & Paré, D. (2009). Coherent gamma oscillations couple the amygdala and striatum during learning. *Nature Neuroscience*, *12*(6), 801–807. <https://doi.org/10.1038/nn.2305>

Rahman, M. M., Shukla, A., & Chattarji, S. (2018). Extinction recall of fear memories formed before stress is not affected despite higher theta activity in the amygdala. *eLife*, *7*, e35450. <https://doi.org/10.7554/eLife.35450>

Rau, E. M. B., Fellner, M.-C., Heinen, R., Zhang, H., Yin, Q., Vahidi, P., Kobelt, M., Asano, E., Kim-McManus, O., Sattar, S., Lin, J. J., Auguste, K. I., Chang, E. F., King-Stephens, D., Weber, P. B., Laxer, K. D., Knight, R. T., Johnson, E. L., Ofen, N., & Axmacher, N. (2025). Reinstatement and transformation of memory traces for recognition. *Science Advances*, *11*(8), eadp9336. <https://doi.org/10.1126/sciadv.adp9336>

Rivière, D., Geffroy, D., Denghien, I., Souedet, N., & Cointepas, Y. (2011). *Anatomist: A python framework for interactive 3D visualization of neuroimaging data*. 3–4.

Rolls, E. T. (1999). Spatial view cells and the representation of place in the primate hippocampus. *Hippocampus*, *9*(4), 467–480.

Saint Amour di Chanaz, L., Pérez-Bellido, A., Wu, X., Lozano-Soldevilla, D., Pacheco-Estefan, D., Lehongre, K., Conde-Blanco, E., Roldan, P., Adam, C., Lambrecq, V.,

- Frazzini, V., Donaire, A., Carreño, M., Navarro, V., Valero-Cabré, A., & Fuentemilla, L. (2023). Gamma amplitude is coupled to opposed hippocampal theta-phase states during the encoding and retrieval of episodic memories in humans. *Current Biology*, *33*(9), 1836-1843.e6. <https://doi.org/10.1016/j.cub.2023.03.073>
- Schmitz, A., & Grillon, C. (2012). Assessing fear and anxiety in humans using the threat of predictable and unpredictable aversive events (the NPU-threat test). *Nature Protocols*, *7*(3), 527–532.
- Seidenbecher, T., Laxmi, T. R., Stork, O., & Pape, H.-C. (2003). Amygdalar and Hippocampal Theta Rhythm Synchronization During Fear Memory Retrieval. *Science*, *301*(5634), 846–850. <https://doi.org/10.1126/science.1085818>
- Shiban, Y., Pauli, P., & Mühlberger, A. (2013). Effect of multiple context exposure on renewal in spider phobia. *Behaviour Research and Therapy*, *51*(2), 68–74. <https://doi.org/10.1016/j.brat.2012.10.007>
- Sperl, M. F. J., Panitz, C., Rosso, I. M., Dillon, D. G., Kumar, P., Hermann, A., Whitton, A. E., Hermann, C., Pizzagalli, D. A., & Mueller, E. M. (2019). Fear Extinction Recall Modulates Human Frontomedial Theta and Amygdala Activity. *Cerebral Cortex*, *29*(2), 701–715. <https://doi.org/10.1093/cercor/bhx353>
- Stam, C. J., Nolte, G., & Daffertshofer, A. (2007). Phase lag index: Assessment of functional connectivity from multi channel EEG and MEG with diminished bias from common sources. *Human Brain Mapping*, *28*(11), 1178–1193.
- Staresina, B. P., Bergmann, T. O., Bonnefond, M., van der Meij, R., Jensen, O., Deuker, L., Elger, C. E., Axmacher, N., & Fell, J. (2015). Hierarchical nesting of slow oscillations, spindles and ripples in the human hippocampus during sleep. *Nature Neuroscience*, *18*(11), 1679–1686. <https://doi.org/10.1038/nn.4119>
- Stujenske, J. M., Likhtik, E., Topiwala, M. A., & Gordon, J. A. (2014). Fear and Safety Engage Competing Patterns of Theta-Gamma Coupling in the Basolateral Amygdala. *Neuron*, *83*(4), 919–933. <https://doi.org/10.1016/j.neuron.2014.07.026>
- Taschereau-Dumouchel, V., Kawato, M., & Lau, H. (2020). Multivoxel pattern analysis reveals dissociations between subjective fear and its physiological correlates. *Molecular Psychiatry*, *25*(10), 2342–2354. <https://doi.org/10.1038/s41380-019-0520-3>
- Taub, A. H., Perets, R., Kahana, E., & Paz, R. (2018). Oscillations synchronize amygdala-to-prefrontal primate circuits during aversive learning. *Neuron*, *97*(2), 291–298.
- Tonegawa, S., Liu, X., Ramirez, S., & Redondo, R. (2015). Memory Engram Cells Have Come of Age. *Neuron*, *87*(5), 918–931. <https://doi.org/10.1016/j.neuron.2015.08.002>

- Vansteenwegen, D., Vervliet, B., Iberico, C., Baeyens, F., Van den Bergh, O., & Hermans, D. (2007). The repeated confrontation with videotapes of spiders in multiple contexts attenuates renewal of fear in spider-anxious students. *Behaviour Research and Therapy*, *45*(6), 1169–1179. <https://doi.org/10.1016/j.brat.2006.08.023>
- Varela, F., Lachaux, J.-P., Rodriguez, E., & Martinerie, J. (2001). The brainweb: Phase synchronization and large-scale integration. *Nature Reviews Neuroscience*, *2*(4), 229–239.
- Vinck, M., Oostenveld, R., van Wingerden, M., Battaglia, F., & Pennartz, C. M. A. (2011). An improved index of phase-synchronization for electrophysiological data in the presence of volume-conduction, noise and sample-size bias. *NeuroImage*, *55*(4), 1548–1565. <https://doi.org/10.1016/j.neuroimage.2011.01.055>
- Visser, R. M., Scholte, H. S., Beemsterboer, T., & Kindt, M. (2013). Neural pattern similarity predicts long-term fear memory. *Nature Neuroscience*, *16*(4), 388–390. <https://doi.org/10.1038/nn.3345>
- Wang, Y., Olsson, S., Lipp, O. V., & Ney, L. J. (2024). Renewal in human fear conditioning: A systematic review and meta-analysis. *Neuroscience & Biobehavioral Reviews*, *159*, 105606. <https://doi.org/10.1016/j.neubiorev.2024.105606>
- Wehrli, J. M., Xia, Y., Gerster, S., & Bach, D. R. (2022). Measuring human trace fear conditioning. *Psychophysiology*, *59*(12), e14119. <https://doi.org/10.1111/psyp.14119>
- Xia, Y., Wehrli, J., Gerster, S., Kroes, M., Houtekamer, M., & Bach, D. R. (2023). Measuring human context fear conditioning and retention after consolidation. *Learning & Memory*, *30*(7), 139–150.
- Xie, W., Bainbridge, W. A., Inati, S. K., Baker, C. I., & Zaghoul, K. A. (2020). Memorability of words in arbitrary verbal associations modulates memory retrieval in the anterior temporal lobe. *Nature Human Behaviour*, *4*(9), 937–948. <https://doi.org/10.1038/s41562-020-0901-2>
- Xue, G., Dong, Q., Chen, C., Lu, Z., Mumford, J. A., & Poldrack, R. A. (2010). Greater Neural Pattern Similarity Across Repetitions Is Associated with Better Memory. *Science*, *330*(6000), 97–101. <https://doi.org/10.1126/science.1193125>
- Zaki, Y., Mau, W., Cincotta, C., Monasterio, A., Odom, E., Doucette, E., Grella, S. L., Merfeld, E., Shpokayte, M., & Ramirez, S. (2022). Hippocampus and amygdala fear memory engrams re-emerge after contextual fear relapse. *Neuropsychopharmacology*, *47*(11), 1992–2001. <https://doi.org/10.1038/s41386-022-01407-0>

We thank all three reviewers for their favorable evaluation of our manuscript. Below, we address the issues regarding code availability raised by Reviewer 2.

Reviewer #1:

Remarks to the Author:

I'd like to thank the authors for their thorough and thoughtful responses to all of my concerns. I have no further issues, and I am happy to recommend this excellent paper for publication. Congratulations on this outstanding work.

We thank the reviewer for their positive assessment of our manuscript.

Reviewer #2:

Remarks to the Author:

Reviewer Comments on the Author's Rebuttal Letter

Criterion 1 – Data & Methodology

Comment on Response to Rebuttal 1 of Criterion 1.

The authors have adequately addressed the comment regarding the cognitive profiles of the epilepsy patients included in this study. The addition of Supplementary Table 1, containing detailed information on IQ, MQ, and seizure onset zones (SOZ) for the participants, improves the clarity and interpretability of the findings. Given that complete neuropsychological data were not available for all subjects due to clinical constraints, the authors have acknowledged this limitation and provided relevant available data. The authors clarify that the participants' average IQ fell within the normal range, while also noting that the Memory Quotient (MQ) scores were below average. They have discussed how these cognitive characteristics might influence their findings and appropriately included this as a limitation in their discussion.

The limitation in the experimental design—specifically the limited number of visual cues and consistent instructions across trials—is reasonably defended, since it could mitigate the impact of potential cognitive impairments on task performance. Furthermore, the authors cite studies showing epilepsy patients can handle similarly complex tasks, supporting the study's context.

Overall, the authors' response satisfactorily addresses the initial concerns about cognitive factors.

Comment on Response to Rebuttal 2 of Criterion 1.

The authors have provided a clear and comprehensive response regarding the use of different co-registration methods across study sites. Their explanation—that these differences reflect institutional practices rather than technical limitations—is reasonable and aligns with common practices in multi-center iEEG research. The revision of the Methods section to highlight the shared core steps of electrode localization, including

the use of standard tools such as FreeSurfer, SPM, and 3D Slicer, enhances transparency.

The inclusion of detailed descriptions of both pipelines—the EPILOC toolbox employed in Paris and the custom approach used in Guangzhou—helps establish that both follow validated and widely accepted procedures. The cross-validation analysis, in which the Paris team re-processed a subset of Guangzhou data and arrived at consistent electrode labels, provides additional empirical support for the reliability of their co-registration methods.

Moreover, the authors' clarification that most analyses were based on broader anatomical regions helps alleviate concerns about minor localization discrepancies. Overall, the authors' response adequately addresses the concern and improves confidence in the methodological consistency and robustness of the study.

Comment on Response to Rebuttal 3 of Criterion 1.

The authors have provided a detailed and well-structured justification of their experimental design that addresses the concerns raised. They clarify that while the paradigm incorporates specific elements from prior studies—such as the 50% partial reinforcement rate used by Dunsmoor et al. (2014)—it was not directly based on that work. This distinction is now clearly stated in the revised manuscript.

The authors outline how each component of the paradigm is informed by existing literature, drawing on a broad range of studies from both human and animal fear conditioning research. The rationale for using a multi-cue structure, the ABC contextual framework, naturalistic stimuli and context videos, and the inclusion of a cognitive narrative is systematically presented and supported by relevant references. The use of multiple context videos per phase facilitates representational similarity analysis (RSA) and aligns with the methodological goals of the study.

The response also addresses practical considerations related to the clinical setting, such as the absence of individualized calibration for stimulus aversiveness and the timing of testing sessions. These decisions are explained transparently and are consistent with approaches taken in related studies. The mention of a recent fMRI study by the same group using a similar design, while not central to the current work, supports the feasibility and applicability of the paradigm.

Overall, the response clearly explained theoretical foundation of the study design. Particularly notable is that they included a comprehensive Supplementary Note.

Comment on Response to Rebuttal 4 of Criterion 1.

The authors have clearly explained the categorization of CS- trials during the extinction phase. They acknowledge that the original figures grouped both CS+- and CS-- trials under the CS- label and have revised the Methods section and corresponding figures (Figures 2 and 3) to reflect this distinction more accurately.

The updated figures, which include separate analyses for CS++, CS+-, and CS-- trials, improve the interpretability and specificity of the results. The observation that amygdala theta power differs between CS++ and both CS+- and CS-- trials—but not between CS+- and CS-- trials—supports the interpretation that neural responses during extinction are more strongly influenced by current stimulus valence than by prior reinforcement history. This pattern is also evident in the item stability analyses for both the amygdala and the temporal pole, where the most pronounced differences occur between CS++ and the two CS- conditions.

The interpretation that extinction leads to previously threatening (CS+-) cues being represented more similarly to consistently safe (CS--) cues is supported by the data. In addition, the discussion of residual fear memory reinstatement effects, as reflected in context representation dynamics (Figure 6), provides additional perspective on the neural processes underlying extinction.

Overall, the authors' clarification and additional analyses adequately address the concern.

Comment on Response to Rebuttal 5 of Criterion 1.

For my concern regarding the reported theta power levels for CS++ trials during extinction, they appropriately note that the early effects did not survive correction for multiple comparisons when applying a cluster-based permutation approach. This statistical rationale—emphasizing the increased likelihood of false positives given the large number of time-frequency bins analyzed—is consistent with current standards in time-frequency analysis.

The authors have also revised the Discussion section to explicitly clarify this point, thereby reducing the risk of misinterpretation. Their conclusion that the later significant effects (1.18–1.75 s) represent the only reliable indicators of reduced theta power in response to threatening stimuli is supported by the corrected analysis.

Overall, this response clarifies the statistical interpretation of the findings.

Criterion 3 - Appropriate use of statistics and treatment of uncertainties

Comment on Response to Rebuttal 1 of Criterion 3.

For the statistical concerns raised, they clearly explain the rationale for setting the minimum number of trials to eight, referencing Oehrns et al. (2015) to support the view that fewer than five trials are generally insufficient for reliable hippocampal theta analyses in iEEG research. This reference offers a relevant justification for the chosen threshold. In addition, control analyses that vary the minimum number of trials (5, 8, and 10) indicate that the primary findings are robust across different inclusion criteria, supporting the validity of the reported effects.

Regarding the behavioral data, the authors include a follow-up analysis of the test phase using one-way ANOVAs and post hoc comparisons. These analyses indicate that

CS++ items continued to be rated as more threatening than CS+- and CS-- items, consistent with partial fear renewal. The clarification that no progressive learning effects during the test phase survived correction for multiple comparisons helps to more accurately frame the behavioral findings.

The authors also address concerns about statistical power and sample size, noting the advantages of their within-subjects design and reporting effect size estimates (partial eta squared) that suggest the sample size was sufficient to detect key effects.

Finally, the authors examine the normality assumptions underlying their statistical tests. In addition to reporting the limitations of parametric analyses in certain trial-level behavioral measures, they provide results from non-parametric Wilcoxon and Friedman tests. The replication of core findings using these methods supports the robustness of the analyses.

Overall, the response provides a clear and methodologically grounded justification of the statistical approach and supports the reliability of the study's conclusions.

Criterion 6 - Suggested improvements

Comment on Response to Rebuttal 1 of Criterion 6.

The authors have provided a detailed response to the previous comment concerning the role of faster-cycle neural oscillations in threat and safety responses. The additional paragraph in the Discussion section (lines 490–508) situates the current findings within a broader neurophysiological framework by considering gamma oscillations. The authors outline relevant theoretical perspectives and empirical evidence, including mechanisms such as theta-gamma coupling and spike-field coherence, that may underlie the functional role of gamma activity.

They also acknowledge the absence of significant gamma effects in their own dataset and suggest potential directions for future research, such as analyses of phase-amplitude coupling and interactions with hippocampal representational patterns. This addition provides a more comprehensive context for interpreting the current results and clarifies the potential involvement of high-frequency oscillations in signaling processes related to threat and safety.

Comment on Response to Rebuttal 2 of Criterion 6.

I have no comments.

Comment on Response to Rebuttal 3 of Criterion 6.

I have no comments.

Comment on Response to Rebuttal 4 of Criterion 6.

The authors have provided a detailed response to the concern regarding potential

differences in electrode configurations between the Paris and Guangzhou cohorts. They outline how variables such as electrode type, contact spacing, and referencing schemes may influence the volume of neural tissue sampled, and describe the steps taken to account for these factors in their analyses.

Several strategies were used to assess and mitigate potential bias, including restricting analyses to gray matter contacts, averaging within subjects before conducting group-level comparisons, and evaluating sampling density using Delaunay triangulation. The analysis indicated no significant differences in electrode density between the two cohorts, supporting the comparability of the datasets.

Furthermore, the authors report that the key finding—amygdala theta power differences—was replicated independently in each cohort. This analysis, presented in Supplementary Figure 2, provides additional support for the robustness of the results across sites. The inclusion of this information, along with discussion of relevant considerations in multi-center iEEG research, contributes to the overall methodological transparency of the study.

These clarifications collectively address the concern and support the validity of the reported findings across different clinical settings.

Comment on Response to Rebuttal 5 of Criterion 6.

The authors have provided a clear response to the concern regarding the spatial specificity of electrode placement within large regions of interest (ROIs) such as the temporal pole (TMP), lateral prefrontal cortex (IPFC), hippocampus (HPC), and amygdala (AMY). The rationale for aggregating subregions—based on anatomical constraints, existing literature, and the need to maintain sufficient statistical power—is consistent with established practices in iEEG research.

The explanation of how subregions were grouped is supported by visual documentation (Supplementary Figure 3) and a comprehensive list of electrode coordinates (Supplementary Table 2), which contribute to transparency. The inclusion of Supplementary Video 1 further illustrates electrode placement in the AMY and HPC. Although the study did not aim to differentiate subnuclei or subfields within the AMY or HPC, the authors provide a clear justification for this decision and acknowledge the methodological trade-offs associated with their representational analysis approach. Overall, the response addresses the concern regarding the spatial extent of the TMP and IPFC regions by providing a rationale for the chosen level of aggregation and supporting it with visual and tabular documentation. While the study did not resolve electrode placement at the level of hippocampal subfields or amygdala nuclei, the authors offer a justification for this approach and acknowledge the associated limitations, thereby clarifying the level of anatomical specificity achieved in the current analysis.

Comment on Response to Rebuttal 6 of Criterion 6.

The authors have provided a detailed clarification regarding their approach to excluding seizure-related activity from the iEEG data. The description of the automatic artifact detection algorithm indicates that the procedure incorporates multiple features—such as amplitude, gradient, and high-frequency activity—based on established methods (Staresina et al., 2015). The use of z-scoring and conservative thresholding suggests that the approach is designed to be sensitive to epileptiform events.

In addition to automated detection, the authors confirm that all identified segments underwent visual inspection of both raw signals and time-frequency spectrograms. This combination of automated and manual review contributes to the reliability of the artifact rejection process. The additional step of excluding data 1,000 ms before and after each artifact point further reduces the risk of contamination by residual epileptic activity. These clarifications in the revised manuscript address the initial ambiguity and provide a more complete account of how seizure activity was excluded, thereby increasing confidence in the reliability of the preprocessing methods.

Remarks on code availability:

Reviewer comments regarding code reproducibility and syntax errors:

I prioritized reviewing the scripts listed in the provided README.md, specifically:

- behavior.m (for reproducing behavioral results, Fig. 1)
- power_analysis.m (for reproducing theta power results, Fig. 2)
- contrast_based_RSA.m (for reproducing item stability and specificity, Figs. 3-4)
- trial_level_analysis.m (for coordination analyses across ROIs, Figs. 3-5)

We thank the reviewer for their positive evaluation of our manuscript. Below, we provide a point-by-point response to the issues raised regarding the code availability.

The following issues were identified during this review:

1. Critical reproducibility issue – Missing dependency: load_paths_EXT.m

All provided analysis scripts begin with:

```
paths = load_paths_EXT;
```

However, this essential function (load_paths_EXT.m) is missing from the repository and explicitly excluded via .gitignore. There are also no instructions or templates provided for creating or configuring this file. Consequently, the scripts cannot currently be executed.

Required action:

- Include a default load_paths_EXT.m file, or
- Provide a clearly documented template for users, or
- Add detailed instructions in the README for configuring this file.

[We have now provided a template for this file in the Github repository \(https://github.com/dpachec/extinction/tree/main/additional_functions/\)](https://github.com/dpachec/extinction/tree/main/additional_functions/)

load_paths_EXT_template.m), which creates a Matlab struct where organized information about the paths can be included. Placeholder directories for raw and preprocessed files, behavioral and iEEG data are provided. We also included placeholder paths for directories that store the results of the analyses. These paths should be replaced with corresponding local paths to run the code.

2. Missing data files (*_trlinfo.mat)

The behavior.m script explicitly requires subject-specific trial information files (e.g., c_sub01_trlinfo.mat). These files were not provided.

Required action:

- Supply the missing *_trlinfo.mat files, or
- Provide instructions and scripts for generating these files from raw data.

We have included all _trlinfo.mat files in our OSF repository (<https://osf.io/xfprt/trlinfo>). These files can also be generated using the log2trlinfo.m script included in the Github repository

(https://github.com/dpachec/extinction/tree/main/additional_functions/log2trialinfo.m).

We have included instructions on how to generate these files in the readme.md file of the Github repository (<https://github.com/dpachec/extinction/README.md>). The log2trlinfo.m script essentially parses the .log files generated in the Presentation software into numerical variables for further processing in Matlab.

3. Missing file and required structure (allS_AMY_C.mat)

The scripts depend on allS_AMY_C.mat located in paths.results.power, containing an ALLEEG cell array with EEG channel locations (EEG.chanlocs). Without this file, the analyses cannot be executed.

Required action:

- Provide the allS_AMY_C.mat file or clear instructions on generating it.

We have included the file allS_AMY_C.mat in the OSF repository

(<https://osf.io/xfprt/power>). This file stores the results of the oscillatory power analysis conducted in the Amygdala in every subject. We also included instructions on how to generate this file in the script create_POW.m (line 73), which performs time-frequency decomposition of the epoched time-series of raw amplitude data (stored in <https://osf.io/xfprt/traces>).

4. Missing additional data files

The correlation analysis requires additional files from the paths.results.trial_based directory:

- AMY_POW_1-44Hz_TR.mat

- trlSTA_HPC_CE_1-44_1_0_500-50.mat

These files must contain the specific structures (allPOWAMY, itstaTRALL, or ctxTRALL) used for trial-wise analyses.

Required action:

- Provide the listed .mat files or detailed instructions and scripts to generate them.

We have included these files in the OSF repository (https://osf.io/xfprt/trial_based).

Below, we provide instructions on how to generate them.

The file AMY_POW_1-44Hz_TR.mat, can be generated using the following script: `extract_power_in_cluster_EXT.m`, included in the Github repository. This file stores the results of the trial level estimates of AMY power during the time-frequency period of significant differences between CS+ and CS- trials during extinction (Time frequency cluster shown in Figure 2A, bottom right). The ids of the time-frequency bins constituting the cluster are stored in the file `_44_clustinfo_AMY_THETA_px32.mat`, which can be found in the OSF repository ([https://osf.io/xfprt/clusters/ 44_clustinfo_AMY_THETA_px32.mat](https://osf.io/xfprt/clusters/44_clustinfo_AMY_THETA_px32.mat)).

The file `trlSTA_HPC_CE_1-44_1_0_500-50.mat` can be generated using the following script: `extract_tr_lev_stab_cxt_spec.m` (lines 1-91, see comments within the script for instructions on how to generate the file). To run this script, indicate the path to the temporally resolved neural RSMs in the `load_paths_EXT.m` script (neural RSM files were also uploaded to the OSF repository at https://osf.io/xfprt/neural_RSMs). To generate the neural RSMs, use the script `create_neural_RSMs.m` (see instructions within the script).

Additionally, I reviewed the `connectivity.m` script separately and found the following syntax error:

```
amyDF = eegfilt (amyD,1;;, f2u(1), f2u(end));
```

The syntax `1;;`, is invalid in MATLAB. The corrected syntax should be:

```
amyDF = eegfilt(amyD, 1, f2u(1), f2u(end));
```

Please also verify the accuracy of using 1 as the sampling rate parameter (`srate`), as EEG analyses typically require precise sampling rate information.

Required action:

- Correct the syntax error.
- Confirm and correct the sampling rate parameter if necessary.

We thank the reviewer for noticing this error. Indeed, the second parameter of the `eegfilt.m` function is the sampling rate. We note that the code block where the syntax error was found was not used for the final analyses presented in the manuscript (this

code would not execute due to the syntax error), but constituted a preliminary version of the analysis. We have therefore removed this section of the script from the repository.

We confirm that all connectivity results (presented in the Supplementary Note 6) were obtained using the appropriate sampling rate parameter (i.e., 1000Hz). This can be observed in the `compute_connectivity_EXT.m` function, which is called in the first block of the `connectivity.m` script (line 113). In this script, the sampling rate is correctly specified as follows:

```
amyDF = eegfilt (amyD,1000, foi(1), foi(end));
```

Reviewer #3:

Remarks to the Author:

The authors have provided a revision of their manuscript, and have comprehensively addressed all of the points raised in the initial review. This is an interesting study, and could be of interest to a number of researchers across different fields.

We appreciate the reviewer's favorable evaluation of our manuscript.